# Description and Validation of the Simple, Efficient, Dynamic, Global, Ecological Simulator (SEDGES v.1.0)

Pablo Paiewonsky[1] and Oliver Elison Timm[1]

[1]Department of Atmospheric and Environmental Sciences, State University of New York at Albany, 1400 Washington Ave., Albany, NY 12222

*Correspondence to:* Pablo Paiewonsky (ppaiewonsky@albany.edu)

**Abstract.** In this paper, we present a simple dynamic global vegetation model whose primary intended use is auxiliary to the land-atmosphere coupling scheme of a climate model, particularly one of intermediate complexity. The model simulates and provides important ecological-only variables but also some hydrological and surface energy variables that are typically either simulated by land surface schemes or else used as boundary data input for these schemes. The model formulations and their derivations are presented here, in detail. The model includes some realistic and useful features for its level of complexity, including a photosynthetic dependency on light, full coupling of photosynthesis and transpiration through an interactive canopy resistance, and a soil organic carbon dependence for bare soil albedo. We evaluate the model's performance by running it as part of a simple land surface scheme that is driven by reanalysis data. The evaluation against observational data includes net primary productivity, leaf area index, surface albedo, and diagnosed variables relevant for the closure of the hydrological cycle. In this set up, we find that the model gives an adequate to good simulation of basic large-scale ecological and hydrological variables. Of the variables analyzed in this paper, gross primary productivity is particularly well simulated. The results also reveal the current limitations of the model. The most significant deficiency is the excessive simulation of evapotranspiration in mid- to high northern latitudes during their winter to spring transition. The model has relative advantage in situations that require some combination of computational efficiency, model transparency and tractability, and the simulation of the large scale vegetation and land surface characteristics under non-present day conditions.

## 1 Introduction

Simulation of the land surface is a critical component of models of the Earth climate system. In such models, heat, energy, and moisture transfer between the land and atmosphere or ocean are calculated with information from the hydrological and vegetation components of these models. SEDGES simulates the gross properties of vegetation, as well as some large scale land surface characteristics, which are important for the simulation of energy and moisture exchanges with the atmosphere. These simulated variables are to be used by a land surface scheme in which SEDGES is embedded in a climate or Earth system model. Models of similar complexity as SEDGES exist that simulate the vegetative if not also the hydrological and energy-transferring aspects of the land surface include VECODE (Brovkin et al., 1997, 2002) and ENTS (Williamson et al., 2006). The purpose of these kinds of models is to efficiently and reasonably simulate dynamic land surface characteristics and behavior and provide

these to the atmospheric components of Earth system models of intermediate complexity. In such a framework, the need for sophisticated simulations of the land surface is obviated by the simplifications present in the other model components (which reduce the benefit of more realistic representations of land surface processes as compared to when the other model components are complex and realistic), by a desire to more easily understand the processes underlying experimental results, and by the computational burden that goes along with the increased complexity.

SEDGES is a new model that is based on the original SimBA model (Kleidon, 2006b), which is coupled to the Planet Simulator (PlaSim) general circulation model (GCM) (Lunkeit et al., 2007) and even more strongly based on a later version of SimBA that the lead author of this paper developed (Lunkeit et al., 2011), and which is also coupled to PlaSim. Neither version of SimBA has been thoroughly evaluated. However, when coupled to PlaSim, the earlier version's net primary productivity was shown to be broadly reasonable in (Kleidon, 2006b), and it has been used successfully in a number of studies (e.g., Kleidon, 2006a, b; Bowring et al., 2014). The SimBA model is based on a light and water-use-efficiency model for crops (Monteith et al., 1989) that was later adapted and expanded to forest canopies (Dewar, 1997). This approach to vegetative productivity is maintained in SEDGES and helps form the core of the model. Other aspects of SEDGES's structure that are particular to SEDGES (i.e., are uncommon in or absent from other models) are carried over from the original SimBA, including the determination of soil water holding capacity and moist-soil leaf cover fraction by the amount of vegetation biomass (Fig. A3).

SEDGES builds upon SimBA by improving most of its parameterizations. Some of the updated and expanded model features are taken or adapted from currently existing models. Although SEDGES makes many improvements upon the original SimBA, it is still a simple model; i.e., it is not of "intermediate" complexity (c.f. Willeit and Ganopolski, 2016). Compared to the original SimBA, SEDGES (and the land surface framework that it presupposes) has four major increases in complexity: separation of evapotranspiration into soil and vegetative components, inclusion of aerodynamic conductance in the formulation for carbon uptake by vegetation, full coupling of photosynthesis and transpiration through interactive canopy conductance, and soil organic carbon-dependent soil albedo. Choosing the appropriate level of complexity of represented processes within a land surface or dynamic global vegetation model often depends on the context in which the model is to be used and involves subjective weighting of tradeoffs between increased accuracy and realism on the one hand and, on the other hand, robustness to the chosen values of poorly constrained parameters and model reliability in a wide range of situations (Prentice et al., 2015).

This paper presents the SEDGES model's structure, equations, and ability to simulate ecological and hydrological variables when used in conjunction with a simple soil hydrological scheme and parameterization for aerodynamic conductance and forced by reanalysis data.

## 2 Model Description: SEDGES

### 2.1 Overview of SEDGES

SEDGES is a simple, dynamic global vegetation model (DGVM). Within the context of the history of land surface modeling, the SEDGES framework (defined as SEDGES and the type of land surface model that it presupposes that it forms a part of) combines aspects of 1st and 3rd generation models (Sellers et al., 1997; Pitman, 2003; Prentice et al., 2015), which include,

respectively, the use of a simple single-layer "bucket" model for soil hydrology and the full coupling of photosynthesis and transpiration through interactive canopy conductance.

The intended main use of SEDGES is for it to be embedded within a land surface scheme and to thus help the land surface scheme simulate large-scale properties and behavior of the surface in which vegetation and soil play a role. These large-scale properties and behavior include interactions between the land surface and the atmosphere through simulated $CO_2$, moisture, energy, heat, and momentum exchanges. SEDGES either directly simulates or else provides the greater land surface scheme with the necessary variables. SEDGES also simulates ecological variables (e.g., biomass, soil organic carbon, forest cover fraction, leaf cover fraction) that are not directly used outside it by either the rest of the land surface scheme or by the atmosphere, but these ecological variables almost always have downstream impact on the simulations of the aforementioned land-atmosphere exchanges. SEDGES has been designed, in particular, for coupling with the Planet Simulator, and thus presupposes that the land surface scheme that it forms a part of be similar to that of PlaSim in its hydrology and scheme for surface evaporation. This section presents, in detail, the SEDGES model formulations for its output ecological and land surface variables, including their derivations.

Overall, sub-grid scale heterogeneity is treated similarly in SEDGES as it is in the SLAM-1T configuration of the CHASM model (Desborough, 1999) and in the original MOSES land surface scheme (Cox et al., 1999). Because of its need to couple with the Planet Simulator GCM, which uses a bulk aerodynamic formulation with a single tile (i.e. one surface type per grid cell), SEDGES handles sub-grid scale heterogeneity according to what is called the "average parameter" method (Giorgi and Avissar, 1997) or the "composite" approach (Li and Arora, 2012; Melton and Arora, 2014). In this approach, land surface characteristics are aggregated to provide a single, representative value at the scale of the entire grid cell. The framework in SEDGES for evapotranspiration (ET) is special, because it also qualifies as a simplified mosaic (or "mixed" (Li and Arora, 2012)) approach, such that only surface conductance differs between the surface types (or tiles). This mosaic approach is also used to extend a "big leaf" formulation for vegetative $CO_2$ uptake (appendix B of Raupach, 1998) to a mixed soil-vegetation surface, enabling us to isolate the fraction of total ET that is due to transpiration and thus formulate vegetative control over water loss and $CO_2$ uptake. The SEDGES framework neglects evaporation from intercepted canopy water and thus only distinguishes between two tiles when snow is absent: bare soil and vegetation. When there is snow cover, there are essentially three tiles: snow-free exposed bare soil, snow-covered surface, and snow-free exposed vegetation (see section 2.3.3 for more detail).

The core driving variable in SEDGES is gross primary productivity ($GPP$), which impacts the living biomass, which, in turn, heavily influences or determines the values of almost every other simulated land surface variable (listed in section 3). Forest cover and leaf cover fractions and (implicitly) rooting depth are parameterized to increase with biomass using respective relationships that are fixed. SEDGES (as well as SimBA) uses a simple single layer bucket for the soil hydrology and has essentially one plant functional type. Soil organic carbon is treated as a single homogeneous reservoir. Surface albedo depends on soil organic carbon content, snow depth, biomass, and leaf cover fraction. Surface roughness increases monotonically with

biomass. Gross primary productivity and canopy resistance[1] are coupled through the vegetation's effort to conjointly minimize water loss and satisfy photosynthetic demand for $CO_2$, as per the original Dewar (1997) model (see section 2.2.6). Table 1 lists all SEDGES variables, their units, and their relationship with the SEDGES vegetation model. How those variables are updated and their dependencies are shown in Figs. A3 and A4. Table 2 lists all parameters that are used in the paper, their values, and their units.

With its main use and design already indicated, SEDGES can also be forced offline with external data. The offline mode of SEDGES was used in combination with a simple soil hydrological scheme and parameterization for aerodynamic conductance and run using reanalysis climate data for the SEDGES model evaluation (see section 4).

In sections 2.2 and 2.3, we describe in detail the equations of the ecological and physical variables in the SEDGES model. Because SEDGES uses the original SimBA model (Kleidon, 2006b) (and its code) as a basis, we explicitly mention in the text when significant changes are introduced in SEDGES compared with SimBA. We provide extra detail on the original SimBA for the cases in which Kleidon (2006b) provides insufficient information for comparison with SEDGES. The broadest structural increases in complexity in SEDGES have already been identified in the introduction; our focuses have finer scope in these coming sections.

[Table 1 about here.]

[Table 2 about here.]

## 2.2 Equations for Ecological Variables

### 2.2.1 Vegetation Biomass

The most important SEDGES variable is biomass of live vegetation. This quantity is used to directly or indirectly derive almost all the land surface variables; these are presented in the coming sections. The prognostic equation for biomass is as follows:

$$\frac{\mathrm{d}C_{veg}}{\mathrm{d}t} = NPP - L, \tag{1}$$

where $C_{veg}$ is the carbon in living biomass $\mathrm{kgC\,m^{-2}}$, $L$ is the litterfall and equals $\frac{C_{veg}}{\tau_{veg}}$, $\tau_{veg}$ is the residence time of the vegetative carbon (see section 2.2.7) and equals 10 years, and $NPP$ (net primary productivity) is approximated as $0.5GPP$, where $GPP$ is gross primary productivity.

### 2.2.2 Net Primary Productivity and Gross Primary Productivity

SEDGES uses a constant $NPP/GPP = 0.5$ approximation. This approximation is supported by the conservative nature of the ratio of mitochondrial respiration to gross photosynthesis and (hence) of the ratio of net photosynthesis to gross photosynthesis

---

[1]Canopy resistance and conductance are aggregated or "bulk" versions of, respectively, stomatal resistance and conductance, such that the entire canopy can be treated like a big leaf (e.g., see Jarvis and McNaughton, 1986; Raupach and Finnigan, 1988)

over a wide variety of conditions, on time scales of weeks or more (see the brief review in Van Oijen et al. (2010)). Since each model time step in SEDGES is much shorter than this, it thus might seem incorrect to hold $NPP/GPP$ fixed for each time step. It is appropriate to do so, however, because the NPP to GPP ratio in SEDGES only impacts biomass changes, and the latter occur on very long time scales. Finally, meta-analyses on previous studies have found a robust (DeLucia et al., 2007; Litton et al., 2007) linear or proportional relationship between NPP and GPP, with slope or proportionality constant of around 0.5 (Gifford, 2003; DeLucia et al., 2007; Litton et al., 2007), albeit with considerable variation in $NPP/GPP$ across field sites (Amthor and Baldocchi, 2001; DeLucia et al., 2007; Litton et al., 2007).

$GPP$ is calculated as the minimum of a light-limited rate, $GPP_L$, and a water-limited rate, $GPP_W$ (Monteith et al., 1989; Dewar, 1997). That is, $GPP = min(GPP_L, GPP_W.)$

## 2.2.3 Light-limited Gross Primary Productivity

$GPP_L$ uses a light-use efficiency (LUE) formulation (e.g., Yuan et al., 2007) whose overall structure is very similar to that of the original version of SimBA (Kleidon, 2006b). The equation is as follows:

$$GPP_L = \epsilon_{max} \cdot f_1(CO_2) \cdot f_2(T_{sfc}) \cdot f_{APAR} \cdot SW\downarrow; \tag{2}$$

where

$\epsilon_{max}$ is a globally-constant maximum LUE parameter = $5.0 \times 10^{-10}$ kgC J$^{-1}$;

$f_1(CO_2)$ is a $CO_2$ fertilization function (described below);

$f_2(T_{sfc})$ is a temperature limitation function (described below);

$f_{APAR}$ is the fraction of photosynthetically active radiation (PAR) that is absorbed by green vegetation (see below); and

$SW\downarrow$ is the downward flux of shortwave radiation just above the canopy surface (in W m$^{-2}$). (Instead of this term, the original version of SimBA (Kleidon, 2006b) uses the net surface shortwave radiation flux.)

In Eq. (2), the first term on the right hand side, $\epsilon_{max}$, is the LUE with respect to the total short wave broadband radiation that is absorbed by the photosynthetic parts of the vegetation. This constant term is the maximum efficiency with which incident short wave radiation can be used to synthesize vegetative carbon at the reference (360ppmv) $CO_2$ level (also see explanation of $f_{APAR}$, below).

The second term in Eq. (2), $f_1(CO_2)$, increases productivity with increasing $CO_2$ and is taken directly from Eq. (5) of Franks et al. (2013). For $CO_2 > CO_{2comp}$, we have

$$f_1(CO_2) = CO_{2norm} \frac{c_a - CO_{2comp}}{c_a + 2CO_{2comp}}, \tag{3}$$

where

$c_a$ is the atmospheric $CO_2$ concentration (ppmv),

$CO_{2comp} = 40$ppmv is the light compensation point in the absence of dark respiration, and

$$CO_{2norm} = \frac{360 + 2CO_{2comp}}{360 - CO_{2comp}}.$$

Otherwise, $f_1(CO_2) = 0$. This fertilization function replaces an earlier "beta" factor approach used in SimBA (Lunkeit et al., 2011). The latter does not saturate and therefore yields unrealistically large fertilization at high $CO_2$ levels.

The third term in Eq. (2) is $f_2(T_{sfc})$. $f_2(T_{sfc})$ is a ramp function, reducing productivity linearly from a surface temperature of $T_{crit} \equiv 20\,°C$ to $0\,°C$, with $f_2(T_{sfc}) = 0$ for below-$0\,°C$ temperatures. The $20\,°C$ value is the critical temperature below which productivity drops. See appendix A for more discussion.

The fourth term in Eq. (2) is $f_{APAR}$. "$f_{APAR}$" refers to the fraction of photosynthetically active radiation (PAR) that is absorbed by photosynthesizing parts (i.e. green leaves) of plants. PAR is the portion ($\approx 50\,\%$) of incoming solar radiation that is in wavelengths usable for photosynthesis. We approximate $f_{APAR}$ as follows:

$$f_{APAR} = 1 - e^{-k_{veg} \cdot \Omega_c \cdot LAI}, \tag{4}$$

where

$k_{veg}$ is a light extinction coefficient (set to 1 for horizontal leaves),

$\Omega_c$ is the clumping index (or factor) and is set to 0.7, a near-mean value for natural land cover types (Pisek et al., 2010; He et al., 2012), and

$LAI$ is the leaf area index (see section 2.2.8).

Equation (4) uses a simple Beer-Lambert approach, extended to canopies with leaves that are non-randomly distributed in space (Nilson, 1971) (e.g., "clumped"). Here, we have used the common approximation (e.g., see Gower et al., 1999) that $f_{APAR}$ equals the fraction of intercepted PAR ($f_{IPAR}$). Other assumptions and simplifications in Eq. (4) include an azimuthally symmetric leaf distribution (Gower et al., 1999), leaf absorptivity of 1 with respect to PAR, and the neglect of the influence of non-photosynthesizing plant parts. A final and important assumption of horizontal leaves[2] eliminates zenith angle dependency of the light extinction coefficient (Campbell and Norman, 1998), which makes the canopy gap fraction the same from any angle. This, in turn, eliminates the dependency of $f_{IPAR}$ on diffusive/direct radiative partitioning and on zenith angle, thus reducing $f_{IPAR}$ (and $f_{APAR}$) to the simple Beer-Lambert expression of Eq. (4), with constant light extinction coefficient. Earlier versions of SimBA and the derivative model (Dewar, 1997) assume a constant value of 0.5 for $k_{veg}$.

The simplifications made in the last paragraph allow us to interchange $f_{APAR}$ and the leaf cover fraction, $f_{leaf}$. Thus, we can rewrite Eq. (4) as follows:

$$f_{leaf} = 1 - e^{-k_{veg} \cdot \Omega_c \cdot LAI}, \tag{5}$$

---

[2]While spherical leaf angle distribution is commonly assumed for unmeasured canopies, measurements of 58 temperate and boreal broadleaf tree species showed planophile (tending toward horizontal) leaf angle distributions in 30 and spherical distributions in just 5 (Pisek et al., 2013). Moreover, a modeling study (Hikosaka and Hirose, 1997) used game theoretical arguments to show how a horizontal leaf angle distribution is to be evolutionarily expected, in general (save for canopies with high LAI and high self-shading). Thus, globally applying a horizontal canopy leaf distribution is not unreasonable.

where $f_{leaf}$ is the areal fraction of view that is covered by photosynthesizing plant parts when looking directly down on the land surface. That is, $f_{leaf}$ = 1- the gap fraction from the nadir.

### 2.2.4 Water-limited Gross Primary Productivity

Most land plants take up the $CO_2$ they need for photosynthesis through tiny pores in their leaves called "stomata". Water is also lost (transpired) through these same openings. Water loss in excess of water uptake can lead to cavitation, hydraulic system collapse (e.g., see discussion in Sperry et al., 2002), and/or permanent reduction in photosynthetic capacity (Lawlor and Cornic, 2002). To prevent and mitigate such occurrences in both the present and future, most land plants adjust their stomatal openings in such a way as to balance the short and long term costs of transpiration with the photosynthetic gain from increased $CO_2$ intake (Cowan and Farquhar, 1977; Medlyn et al., 2011; Buckley and Schymanski, 2014; Prentice et al., 2014). Thus, water limitation in most land plants is closely tied to $CO_2$ limitation on photosynthesis. For this reason, this $CO_2$-limited rate in SEDGES is referred to as the "water-limited rate of GPP". This rate is given as follows:

$$GPP_W = \frac{(co2conv)(1 - \frac{c_i}{c_a})c_a(f_{leaf})\rho}{1.6r_c + r_a}; \tag{6}$$

where

$r_c$ is the canopy resistance (see section 2.2.6);

$r_a$ is the aerodynamic resistance;

$\rho$ ($\equiv \frac{p_{sfc}}{R_d T_{sfc}}$) is the surface air density, where $p_{sfc}$ is the surface pressure and $R_d$ is the gas constant for dry air;

$co2conv = 4.15 \times 10^{-7}$ kgC kgair$^{-1}$ ppmv$^{-1}$;

$c_i$ represents a "bulk leaf" intercellular concentration of $CO_2$ (ppmv); and

$\frac{c_i}{c_a}$ is set to 0.80. This value was lowered from the $\approx 0.86$ value that was apparently used in Raupach (1998) because such a high value is less representative of typical daytime $\frac{c_i}{c_a}$ values (e.g., see Prentice et al., 2014). SimBA uses a value of 0.7. The ramifications of using a fixed value of $\frac{c_i}{c_a}$ are discussed in section 6.

Equation (6) is essentially the bulk aerodynamic formula for $CO_2$ in appendix B of Raupach (1998), except that we have multiplied their right hand side by $f_{leaf}$ to generalize from their big leaf approach to the simplified mosaic framework (as mentioned in section 2.1), and we have multiplied by the ratio of the molar masses of carbon and $CO_2$ to convert the flux into carbon units. In addition, we neglect the contribution of leaf mitochondrial respiration for simplicity. (It can be shown that including daytime leaf mitochondrial respiration in the formulation of $GPP_W$ multiplies its current form by a factor of $\frac{1}{1-X}$, where $X$ is the ratio of leaf mitochondrial respiration under light to gross photosynthesis.)

The above bulk aerodynamic formulation for $GPP_W$ in SEDGES is more realistic than the diffusive scheme used in earlier versions of SimBA and in the Dewar (1997) model. The latter scheme, at the canopy level, assumes that $r_c >> r_a$ (which is noted, too, by Medlyn et al., 2017). This condition is likely to be satisfied for aerodynamically rough canopies and when using daily-averaged environmental conditions, such as in Dewar (1997). However, early diagnostic tests of SEDGES coupled with PlaSim showed that the condition was often not satisfied, thus motivating our implementation of the bulk aerodynamic

formulation for $CO_2$ uptake. Implementing the new formulation led to higher daytime values of $r_c$ for moist areas (which is shown by Medlyn et al., 2017, to follow from theory). These values were more in line with reported maximum surface conductances for various vegetation types (Kelliher et al., 1995). Use of the bulk aerodynamic formulation for $GPP_W$ also unifies vegetation-atmosphere exchanges of $CO_2$ and water under the same physical framework, which enables the process of transpiration to be properly tied into carbon uptake through interactive canopy conductance (see section 2.2.6).

### 2.2.5 Evapotranspiration and Transpiration

As was mentioned earlier in section 2.1, the SEDGES framework for evapotranspiration (ET) is very similar to the approaches taken by the SLAM-1T configuration of the CHASM model (Desborough, 1999) and by the original MOSES land surface scheme (Cox et al., 1999). These approaches all expand on the SimBA framework (appendix C) by separating evapotranspiration into soil and vegetative components and (in SEDGES and MOSES) by coupling photosynthesis and transpiration through interactive canopy conductance. We find that these changes greatly improve the simulation of ET (compare Fig. A2 with Fig. A16) when using the forcing set up described in section 4.

By design, SEDGES does not directly calculate the evapotranspiration; rather, it provides a surface wetness factor to be used by the bulk aerodynamic formulation of the land surface scheme that is coupled to the atmospheric model, which performs the actual calculation of ET. An exception to this exporting of ET calculation to outside of SEDGES is the explicit calculation by SEDGES of evaporation from exposed soil when there is snow cover.

Although the (total) evapotranspiration is calculated outside of SEDGES, the way it is simulated is intimately tied into many parameterizations within the actual SEDGES model, which presuppose a specific ET framework (as discussed in sections 2.1 and 3). The ET framework is thus presented in this section, along with the SEDGES parameterizations that help comprise it. For this simplified mosaic approach (section 2.1), we remind the reader that the aerodynamic conductance and saturation specific humidity of the air at the surface are homogeneous throughout the grid cell:[3]

$$ET = \frac{\rho}{\rho_w \, r_a} C_w \Delta q, \tag{7}$$

where $ET$ is the evapotranspiration in units of volumetric liquid water $\mathrm{m}^3 \, \mathrm{m}^{-2} \, \mathrm{s}{-}1$,

$\rho_w$ is the density of liquid water,

$\Delta q \equiv qsat_{sfc} - q$,

$qsat_{sfc}$ is the temperature- ($T_{sfc}$) and pressure- ($p_{sfc}$) dependent saturation specific humidity of the air at the surface,

$q$ is the specific humidity of the air at the lowest atmospheric model level, and

---

[3]In this paper, we define evapotranspiration (ET) and transpiration (T) as positive when there is a net water vapor flux from surface to atmosphere. In the actual SEDGES code, ET and T are defined as negative when the flux is from surface to atmosphere. In our exposition, we have reversed the sign convention to aid interpretation of the model equations.

$C_w$ is a surface "wetness" factor that reduces evapotranspiration from the potential rate by incorporating the effects of canopy resistance and soil resistance. $C_w$ ranges from 0 to 1, where "1" gives the potential evapotranspiration (PET) rate:

$$PET = \frac{\rho}{\rho_w \, r_a} \Delta q, \tag{8}$$

Although the above framework for ET (and PET) is required in order for SEDGES to operate as designed, there is no stipulation that the coupled land surface model calculate $qsat_{sfc}$ or $r_a$ in any preset way. That said, it is desirable for the calculation of $r_a$ to depend, in some way, on the surface roughness length that is output by SEDGES 2.3.3.

Under snow-free conditions, SEDGES derives the key ET variable, the surface wetness factor, as follows:

$$C_w = \frac{f_{leaf}}{1 + r_c g_a} + \frac{1 - f_{leaf}}{1 + r_{ss} g_a}; \tag{9}$$

where

$g_a$ is the aerodynamic conductance[4] $(= \frac{1}{r_a})$, and

$r_{ss}$ is the soil surface resistance, whose formulation is taken from the JULES model (Best et al., 2011), except for the use of the lower minimum soil resistance from van de Griend and Owe (1994). This reduction was made to compensate for the absence of evaporation from ponded surface water in SEDGES. Soil surface resistance is formulated as follows:

$$r_{ss} = min\left(r_{ss}max, \frac{r_{ss}min}{\beta_{ss}}\right), \tag{10}$$

where

$r_{ss}min$ is the minimum soil surface resistance of $10\ \mathrm{s\ m^{-1}}$ (van de Griend and Owe, 1994),

$r_{ss}max$ is the maximum soil surface resistance (a very large number), and

$\beta_{ss}$ limits evaporation from the soil surface and is considered a "water stress factor". This factor is formulated according to Best et al. (2011), except that SEDGES uses the entire soil hydrological layer instead of just the top of several layers. The water stress factor for the soil surface in SEDGES is as follows:

$$\beta_{ss} = W_{frac}{}^2, \tag{11}$$

where the soil wetness fraction, $W_{frac}$, is given as follows:

$$W_{frac} = \frac{W_{soil}}{W_{max}}, \tag{12}$$

---

[4]An important requirement for the aerodynamic conductance formulation is that the surface roughness for moisture be the same as that for momentum. See section 2.3.3 for more discussion.

where $W_{max}$ is the the biomass-dependent soil "bucket" depth and $W_{soil}$ is the water content (as depth) within the bucket. The SimBA model uses a general water stress factor that is a ramp function, reducing ET linearly from a soil wetness fraction of 0.25 to 0. In SimBA, $C_w$ is equal to this water stress factor. In this way, no distinction is made between soil moisture's impact on soil evaporation and its impact on transpiration.

Examining Eq. (9), we see that $C_w$ is comprised of a vegetation term and a bare soil term (since $1 - f_{leaf}$ is the bare soil fraction (neglecting non-photosynthesizing plant parts for simplicity)). The vegetation term involves loss of water only due to transpiration; canopy interception and evaporation are neglected for simplicity. Under snow-free conditions, ET is partitioned into transpiration and bare soil evaporation, respectively, by replacing $C_w$ in Eq. (7) with the vegetation term, $\frac{f_{leaf}}{1+r_c g_a}$ or with the bare soil term, $\frac{1-f_{leaf}}{1+r_{ss} g_a}$. Doing so with the vegetation term yields the following equation for transpiration:

$$T = \frac{\rho}{\rho_w} \frac{f_{leaf}}{r_a + r_c} \Delta q. \tag{13}$$

From the above formulations, one can see that we weight the contributions to evapotranspiration from transpiration and bare soil evaporation, respectively, by the fractional coverages of vegetation and bare soil.

     In the presence of snow cover, the surface is treated as a mosaic of different types, with each type sharing the same aerodynamic conductance, soil hydrology, and surface temperature, as discussed before (section 2.1). The parts of the grid cell that

are snow-covered evaporate at the potential rate. The portion of the grid cell considered to be snow-free surface is comprised of vegetation parts that protrude above the snow pack and are snow-free, the exposed bare soil, and the exposed, low-lying leaf cover. In these snow-free portions of the grid cell, transpiration is 0, and bare soil evaporation occurs (Eqs. 7, 9, and 10). The final equation for $C_w$ in the presence of snow in the grid cell is as follows:

$$C_w = \frac{(1-f_{for})(1-f_{leaf})(1-f_{snowflat})}{1+r_{ss} g_a} + (1-f_{for})(f_{snowflat}) + (f_{for})(f_{snowfor}), \tag{14}$$

where

$f_{for}$ is the forest cover fraction (Eq. 23),

$f_{snowflat}$ is the fraction of the flat portion of the grid cell that is covered by snow (see Eq. 27 and nearby text), and

$f_{snowfor} \equiv 0.12$ is the snow-covered fraction of the vegetation cover that protrudes above the snow pack (i.e. of the forest cover).[5]

From Eq. (14) and the relationship $ET = C_w PET$ (from Eqs. 7 and 8), one can deduce that bare soil evaporation, $E_{soil}$ when there is snow cover is given as follows:

$$E_{soil} = \frac{(1-f_{for})(1-f_{leaf})(1-f_{snowflat})}{1+r_{ss} g_a} PET. \tag{15}$$

---

[5]The value of 0.12 is a crude estimate that is obtained by assuming an albedo transition for forest vegetation from snow-free vegetation albedo to deep and cold snow albedo (0.80) that is linear with $f_{snowfor}$. Depending on whether one uses values of 0.30 and 0.12 for the values of snow-covered forest and snow-free vegetation, respectively, or values taken from Betts and Ball (1997), one ends up with $f_{snowfor}$ values ranging from $\approx 0.04$ to 0.26.

In SimBA, $C_w = 1$ when there is snow cover. The formulation was changed in SEDGES when it was discovered that it led to excessive ET when snow was present. The original MOSES and its progenitor UKMO scheme also set their equivalent $C_w$ to 1 when there is snow cover, which also leads to excessive ET under those conditions (Essery, 1998; Cox et al., 1999).

### 2.2.6 Canopy Resistance

Canopy resistance, $r_c$, is determined at each time step in accordance with the supply/demand principle in the original Dewar (1997) paper. That is, the plants attempt to adjust $r_c$ so that $CO_2$ uptake exactly matches the light-limited rate of canopy photosynthesis, $GPP_L$. ($GPP_L$ is the rate of gross carbon uptake in the absence of any $CO_2$ limitation.) However, $r_c$ must also be sufficiently high to prevent transpiration from exceeding the maximum rate suppliable by the roots (the "supply rate"). If the $r_c$ value that would be needed to satisfy the $CO_2$ demand would cause the transpiration to exceed the supply rate, then

water/$CO_2$ limitation occurs, and $r_c$ takes on the value that would yield transpiration at the supply rate. In the derivation of the equation for canopy resistance, the unconstrained adjustment of $r_c$ is the first step. The second step is the calculation of the minimum possible value of $r_c$ that satisfies the supply constraint on transpiration. If needed, the unconstrained $r_c$ is raised to this minimum value. $r_c$ is also constrained to not exceed a maximum possible value, $r_c max$. Details on the mechanics of the formulation are given in appendix B.

The original SimBA model does not explicitly consider canopy resistance. Although the incorporation of an interactive canopy resistance scheme within a coupled water/$CO_2$ exchange framework adds complexity to SEDGES, we feel that this is a critical feature for a vegetation model to have when it is used as part of a land surface model. In particular, significant climatic impacts of stomatal closure induced by increased $CO_2$ are well-established in the literature and include reductions in ET, increased runoff, and increased surface temperature (Sellers et al., 1996a; Betts et al., 1997; Levis et al., 2000; Betts et al.,

2007; Cao et al., 2010).

### 2.2.7 Soil Organic Carbon

The overall modeling framework for soil organic carbon is the same as that of both the original SimBA (Kleidon, 2006b) and the land surface component (ENTS) of the GENIE model (Williamson et al., 2006). In this simple framework, soil organic carbon is treated as a single homogeneous reservoir (in which litter is included) that has a single, temperature-dependent

residence time. Soil respiration (see below) depends only on temperature. The prognostic equation for soil organic carbon is given by

$$\frac{\mathrm{d}C_{soil}}{\mathrm{d}t} = L - R_{soil}, \tag{16}$$

where

$C_{soil}$ is the soil organic carbon (kgC m$^{-2}$),

$L$ is the litterfall (Eq. 1), and

$R_{soil}$ is the rate of soil respiration (kgC m$^{-2}$ s$^{-1}$).

Soil respiration is modeled according to the following equation:

$$R_{soil} = \begin{cases} \dfrac{c_8 C_{soil}}{\tau_{soil}\left(1+e^{\frac{c_9}{T_{soil}-254.85}}\right)}, & T_{soil} > 256.1\,\mathrm{K}, \\[2mm] 0, & T_{soil} \le 256.1\,\mathrm{K}, \end{cases} \tag{17}$$

where

$T_{soil}$ is the soil temperature at $\approx 0.20$ m depth (i.e., the middle depth of the top soil layer in PlaSim),

$\tau_{soil}$ is the residence time (42 years) of the soil organic carbon at 10 °C,

$c_8 \equiv 1 + e^{\frac{106}{283.15-254.85}}$, and

$c_9 = 106\,\mathrm{K}$.

The above formulation is based on that of ENTS (Williamson et al., 2006), except that we replace its temperature dependency (Lloyd and Taylor, 1994) with that from the RothC soil organic carbon model (Jenkinson et al., 1990; Clark et al., 2011), since

the latter's temperature function reduces a negative soil organic carbon bias in the high Arctic. In addition, we use $c_8$ as a normalizing constant to make the soil organic carbon residence time, $\frac{C_{soil}}{R_{soil}}$, at 10 °C the same as in the original SimBA formulation. Moreover, for strictly numerical purposes, 256.1K is used as the cut off temperature, below which soil respiration is set to 0. In SimBA, soil respiration is parameterized as follows: $\left(Q_{10}^{\frac{T_{sfc}-10}{10}}\right)\frac{C_{soil}}{\tau_{soil}}$. (Here, the $T_{sfc}$ is surface temperature in °C, and $Q_{10}$ is set to 2). In the current version of SEDGES, it is tacitly assumed, for simplicity, that all respired soil organic

carbon is emitted directly to the atmosphere.

### 2.2.8 Leaf Area Index and Leaf Cover Fraction

Leaf area index (LAI) is typically defined as "the one-sided leaf area per unit ground area", or else, for conifers, as the projected area of the needle leaves (Monson and Baldocchi, 2014, p. 246). In SEDGES (as well as in SimBA), LAI is based on a moist soil value that gets subsequently reduced for conditions of low soil moisture. We first describe the moist soil formulation.

For sufficiently moist soils, LAI is a simple function of biomass. In the version of SimBA that came out with version 15 of the Planet Simulator model (Lunkeit et al., 2007), this moist soil LAI is a linear function of forest cover fraction (which, in turn, depends directly on biomass). This parameterization is discarded in the subsequent SimBA in version 16 of Planet Simulator, in favor of a simpler LAI functional dependency on biomass (Lunkeit et al., 2011) to reduce the high number of multiple equilibria in the coupled climate-vegetation system that Dekker et al. (2010) found when using PlaSim version 15.

This new parameterization is maintained in SEDGES, but its three parameters have updated values.

The equation for LAI in the absence of soil moisture stress is a monotonically increasing function of biomass with decreasing slope:

$$LAI_m = LAI_{min} + \frac{2}{\pi}(LAI_{max} - LAI_{min})atan(c_6 C_{veg}), \tag{18}$$

where $LAI_{min}$ and $LAI_{max}$ are the minimum and maximum LAI's (respectively) under moist soil conditions. $LAI_{min}$ is set to 0.05 and represents a seeding source with negligible mass. $LAI_{max}$ is set to 7.

While winter-deciduous phenology is not included in SEDGES, the model simulates drought-deciduous phenology using a crude parameterization from SimBA that came originally from the ECHAM3 model (Klimarechenzentrum, 1993). $LAI_m$ is converted into leaf cover fraction, $f_{leaf\,m}$, by substituting it into Eq. (5), giving the following:

$$f_{leaf\,m} = 1 - e^{-k_{veg} \cdot \Omega_c \cdot LAI_m}, \tag{19}$$

where $f_{leaf\,m}$ is the leaf cover fraction under moist soil conditions.

Then, we set

$$f_{leaf} = \min\left(f_{leaf\,m}, f_{leaf\,dry}\right) \tag{20}$$

where

$$f_{leaf\,dry} = \begin{array}{l} 1, W_{frac} \geq W_{frac_{crit,lai}}, \\ \frac{W_{frac}}{W_{frac_{crit,lai}}}, 0 < W_{frac} < W_{frac_{crit,lai}}, \\ 0, W_{frac} = 0. \end{array} \tag{21}$$

Here, $f_{leaf\,dry}$ represents the maximum leaf cover permitted by the soil wetness fraction; $W_{frac_{crit,lai}}$ is the critical soil wetness fraction at which leaf cover begins to get restricted and is set to 0.05.

Although the final $LAI$ is merely a diagnostic, it has a one-to-one relationship with $f_{leaf}$ because the $f_{leaf}$ in Eq. (20) is converted back into LAI by inverting the $LAI$-$f_{leaf}$ relationship in Eq. (5) as follows:

$$LAI = \frac{-\ln(1 - f_{leaf})}{k_{veg} \cdot \Omega_c}. \tag{22}$$

### 2.2.9 Forest Cover Fraction

In SEDGES, forest cover fraction is defined as the fraction of the grid cell above ground that is covered by trees. However, woody shrubs are counted as partial trees. It has no seasonal phenology. Forest cover fraction helps to determine the amount of surface that is covered by snow for the calculation of evaporation (see Eq. 14). In SimBA, forest cover affects the albedo of snow-covered land; instead, SEDGES has a dependency on biomass (Eq. 26). As in the last version of SimBA (Lunkeit et al., 2011), forest cover is parameterized to commence at a biomass of 1 kgC m$^{-2}$. Such cover would represent woody shrubs. Only at a biomass threshold of 1.5 kgC m$^{-2}$ is the woody cover considered tall enough so as to protrude above the winter snow pack and lower the albedo (section 2.3.1). Forest cover fraction is formulated as follows:

$$f_{for} = \max(0, 1 - e^{-c_1(C_{veg} - c_2)}), \tag{23}$$

where $c_1$ is a shape parameter and $c_2$ is the biomass threshold for forest cover.

The derivation of the $c_1$ and $c_2$ parameter values involved translating NPP from outside data into SEDGES biomass. If one assumes long time scales and steady state conditions, then the translation is readily achieved by setting the left hand side of Eq. (1) to 0, and solving for $C_{veg}$ in terms of NPP. Doing so results in the following relationship: $NPP = \frac{C_{veg}}{\tau_{veg}}$. Using this relationship allows modeled annual NPP from Cramer et al. (1999) and McGuire et al. (1992) to be interpreted as SEDGES biomass equivalent. Then, comparing the real world spatial distribution of boreal forest, boreal woodland, and tundra with the NPP data in those sources, including the NPP ranges for boreal forest, boreal woodland, and tundra ecosystems in McGuire et al. (1992), yields a rough threshold value of $1 \, \text{kgC} \, \text{m}^{-2}$ for forest cover commencement. The shape parameter, $c_1$, was tuned so that forest cover fraction values in the Hagemann (2002) land cover dataset would match the NPP values in Cramer et al. (1999) for natural land cover types, while maintaining a fairly close likeness between the SEDGES-parameterized and reference Hagemann (2002) relationships between forest cover fraction and LAI. (This likeness was assessed by scatterplotting the forest cover fraction and maximum LAI for the moist ecosystems given in Hagemann (2002) along with SEDGES-formulated forest cover fractions and LAI's for a wide range of biomass values.) The original SimBA's (Kleidon, 2006b) dependency of forest cover fraction on biomass uses an S-shaped curve that is equivalent to $f_{for} = \frac{\max(0, atan(C_{veg} - 3) - atan(-3))}{0.5\pi - atan(-3)}$ and is similar to the formulation in the later SimBA version (Lunkeit et al., 2011). The two previous SimBA formulations give higher forest cover fractions than SEDGES except at some low biomass values. Neither SimBA parameterization yields a good match with the aforementioned reference data in the absence of using additional fitting parameters.

## 2.3 Equations for Remaining Land Surface Variables

While the equations for the ecological variables were presented in section 2.2, here, we provide the formulations for the land surface variables used by the PlaSim GCM as well as some purely diagnostic variables.

### 2.3.1 Surface Albedo

A surface albedo with respect to broadband short wave radiation is obtained by incorporating the effects of snow cover on the albedo for snow-free conditions. In obtaining the snow-free albedo ($\alpha_0$), we simplify by ignoring dependencies on solar zenith angle, diffuse-direct radiation partitioning, the spectral composition of the incident light, the effect of soil moisture on soil albedo, and any impact of leaf litter. Moreover, we neglect variations in leaf reflectivity that often occur between differing plant species and leaf development stages. The formulation of snow-free albedo in SEDGES is of the same form as that for SimBA, the ECHAM5 GCM (Rechid et al., 2009) and the ENTS land surface scheme (Williamson et al., 2006); it is as follows:

$$\alpha_0 = \alpha_{veg} f_{leaf} + \alpha_{soil}(1 - f_{leaf}), \tag{24}$$

where $\alpha_{veg}$ is the albedo of a completely leaf-covered surface, and $\alpha_{soil}$ is the albedo of the bare soil.

Soil albedo decreases linearly with soil organic carbon to a minimum value that is reached at 9 kgC m$^{-2}$ and stays constant with further carbon increase. The equation is as follows:

$$\alpha_{soil} = \begin{array}{l} \frac{\alpha_{peat}-\alpha_{sand}}{c_7}C_{soil} + \alpha_{sand}, C_{soil} \leq c_7, \\ \alpha_{peat}, C_{soil} > c_7, \end{array} \qquad (25)$$

where

$c_7 = 9$ kgC m$^{-2}$ and is the soil organic carbon level at which the soil albedo saturates,

$\alpha_{sand}$ is the soil albedo in the complete absence of soil organic carbon, and

$\alpha_{peat}$ is the albedo of soil when saturated with soil organic carbon. This soil albedo formulation is taken from ENTS (Williamson et al., 2006), except for the following modifications: the peat albedo has been increased from 0.11 to 0.12, the sand albedo has been increased from 0.30 to 0.32, and $c_7$ was decreased from 15 to 9 kgC m$^{-2}$. The first change was made for the sake of model simplicity (since the albedo of a fully-leaved surface is 0.12 in SEDGES); the increase in sand albedo was made to better match observed values in the Sahara and Arabian deserts (e.g., Knorr and Schnitzler, 2006). A saturation level for soil albedo as a function of soil column carbon content was estimated to be around 9 kgC m$^{-2}$ by visually comparing the soil/litter surface albedo map from Houldcroft et al. (2009) with a global gridded dataset of soil organic carbon content (Wieder et al., 2011). Using this estimate for saturation value in SEDGES reduces a positive albedo bias in the high arctic in summer as compared to using the standard ENTS value of 15 kgC m$^{-2}$.

The soil albedo formulation in SEDGES is an advent to SimBA, which has always used a fixed soil albedo of 0.30. The new dynamic scheme was adopted, in part, due to the important role played by ground albedo in Sahelian/Saharan vegetation-precipitation feedbacks, as seen through its effect on low frequency precipitation variability (Vamborg et al., 2014) and increased greening in the mid Holocene (Vamborg et al., 2011). The new scheme also gives more realistic snow-free albedo values in the high latitudes and in the hot deserts (Fig. A5). While the above formulation for snow-free albedo, $\alpha_0$ neglects, for simplicity, the radiative impact of non-photosynthesizing plant parts (e.g., stems and branches), this impact becomes important (although implicit) in the snow-covered formulation.

A new snow albedo scheme for SEDGES replaces the version from SimBA that linearly combines the albedos from the forested and flat portions of the grid cell according to the forest cover fraction. The SimBA formulation did not simulate well the sharp, real-world transition in snow-covered albedo from tundra to boreal forest (Loranty et al., 2014), which is why a new exponential decay scheme was developed.

In SEDGES, for snow depth > 0, the surface is treated as consisting of two components: 1) "flat" surface (consisting of exposed soil, prostrate vegetation, and dwarf shrubs) that can be partially or entirely buried by the snow cover; 2) "forest" that is covered by trees and shrubs of sufficient stature as to protrude above the snow pack and mask it from the sun via their leaves,

stems, and/or branches, regardless of accumulated snow depth on the ground. A non-linear combination of the albedo of the flat portion and the snow-covered forest albedo yields the final surface albedo ($\alpha$):

$$\alpha = \alpha_{snowfor} + (\alpha_{snowflat} - \alpha_{snowfor})e^{-c_4 \cdot max(0, C_{veg} - c_5)}, \tag{26}$$

where

$c_4$ is a shape parameter, and

$c_5$ is the biomass threshold (1.5 kgC m$^{-2}$) at which the woody vegetation is sufficiently tall as to begin to mask the snow.

     Albedo of the flat portion of the grid cell, $\alpha_{snowflat}$, is formulated according to the same temperature dependency (Roeckner et al., 2003) and snow cover fraction (Roesch and Roeckner, 2006) schemes for non-forested areas in the ECHAM5 model, except that we maintain the melting snow albedo of 0.4 from PlaSim and ECHAM4 (Roesch et al., 2001), and we neglect the

impact of sloping terrain. We thus have the following:

$$\alpha_{snowflat} = \alpha_0 + (\alpha_{deepsnowflat} - \alpha_0) \cdot 0.95 f_{snowflat}, \tag{27}$$

where

$swe$ is the snow depth in liquid water equivalent (m$^3$ m$^{-2}$),

$f_{snowflat} \equiv \tanh(100 swe)$ is the fraction of the flat portion of the grid cell that is covered by snow[6],

$\alpha_{deepsnowflat}$ is the albedo of a deep and pure snow pack and is temperature dependent. This dependency has the form of a ramp function, with a maximum albedo, $\alpha_{maxdeepsnowflat} \equiv 0.8$, for surface temperatures $\leq$ -5°C and a minimum albedo, $\alpha_{mindeepsnowflat} \equiv 0.4$, at melting point. These values are the same as in PlaSim and SimBA, except that they have maximum albedo attainment at -10°C.

     The albedo of the forest component of the grid cell when there is snow cover is as follows:

$$\alpha_{snowfor} \equiv min(\alpha_{snowflat}, \alpha_{maxsnowfor}). \tag{28}$$

Here, $\alpha_{maxsnowfor}$ is the maximum snow-covered albedo for forest and is set to 0.30, a mid-range white sky albedo value for the different forest types (Moody et al., 2007). Restricting $\alpha_{snowfor}$ to not exceed $\alpha_{snowflat}$ is done because the lower snow depth and warmer snow that reduce the albedo of the flat area also lower the albedo of the forested area to at least the level of the flat area.

---

[6]Roesch and Roeckner (2006) include the 0.95 multiplier in their definition of the snow-covered fraction of the non-forested part of the grid cell, but we do not. Note, too, that this scheme for non-forested snow cover fraction supplants the ECHAM4 parameterization $\left( f_{snowflat} = \frac{swe}{swe + 0.01} \right)$ used by SimBA, which was found to poorly estimate satellite-based observations compared to the adopted ECHAM5 parameterization (Roesch et al., 2001).

### 2.3.2 Soil Bucket Depth

We first introduced the soil bucket depth, $W_{max}$, in section 2.2.5 in Eq. (12). SEDGES uses a simplified version of the parameterization in the JeDi dynamic global vegetation model (Pavlick et al., 2013). The JeDi model uses a pipe representation of the rooting system (Shinozaki et al., 1964) in which a square root relationship emerges between $W_{max}$ and coarse root biomass as a result of an assumed constant density of fine roots in the rooting zone. To adapt this formulation to SEDGES, we further assume that coarse root biomass is a fixed fraction of total biomass and that the unit plant available water capacity is spatially constant. Doing so gives the following relationship:

$$W_{max} = \min\left(W_{maxmin}, c_{12}\sqrt{C_{veg}}\right) \tag{29}$$

where c12 is a tuning parameter set to $0.10 \text{ kgC}^{\frac{1}{2}}$. This value yields soil bucket depths that are a reasonably good fit with a plant available water dataset based on optimal rooting depths (Kleidon and Heimann, 1998; Hall et al., 2006; Kleidon, 2011). In that dataset, the soil bucket depths in the most sparsely-vegetated regions range from around 0.05m in the Canadian polar desert to < 0.003m in the hyper-arid Atacama and northeast Sahara deserts. SEDGES adopts a minimum soil bucket depth, $W_{maxmin}$, of 0.05m.

The versions of SimBA (Kleidon, 2006b; Lunkeit et al., 2007, 2011) compute $W_{max}$ using the following formulation: $W_{max} = Z \times W_{maxmax} + (1-Z)W_{maxmin}$, where Z is the forest cover fraction formulation in the original SimBA (see section 2.2.9), and $W_{maxmax}$ is some maximum value for $W_{max}$. This $W_{max}$ formulation was updated in SEDGES because the new formulation has the advantage of being derived from first principles (Pavlick et al., 2013).

### 2.3.3 Surface Roughness

SEDGES has been designed for use in land surface schemes in which the surface roughness lengths for momentum and water vapor are the same (e.g., schemes that use parameterizations from Louis, 1979; Louis et al., 1982). As such, SEDGES returns only a single value for surface roughness length, $z_0$, for a grid cell. The (total) surface roughness is comprised of a surface roughness due to orography and a surface roughness due to vegetation:

$$z_0 = \sqrt{z_{0veg}^2 + z_{0oro}^2}, \tag{30}$$

where $z_{0oro}$ is the orographic surface roughness, and $z_{0veg}$ is the roughness due to vegetation. $z_{0oro}$ is used by some GCM's (including Planet Simulator) to account for the enhancement of surface drag due to sub-grid scale topographic variation (e.g., see Beljaars et al., 2004). $z_{0oro}$ is set to zero by default, because small-scale orographic variation has little effect on land-to-atmosphere moisture transfer (Huntingford et al., 1998; Blyth, 1999).

In the SimBA formulation, surface roughness due to vegetation is parameterized assuming that only forest cover increases it appreciably and that prostrate vegetative cover does not. As such, sparsely-vegetated surfaces are assigned too high of a surface

roughness. SimBA has minimum and maximum values of $z_{0veg}$ of 0.05m and 2m, respectively (Kleidon, 2006b; Lunkeit et al., 2007, 2011). SEDGES replaces SimBA's linear dependency of $z_{0veg}$ on forest cover fraction with the following logistic curve that depends on biomass rather than forest cover fraction:

$$z_{0veg} = \frac{c_{17}}{1 + e^{-c_{16}(C_{veg} - c_{15})}} - z_{0const}, \tag{31}$$

where

$z_{0const} \equiv \frac{c_{17}}{1 + e^{-c_{16}(-c_{15})}} - z_{0min}$ is a constant (in m) that forces $z_{0veg}$ to equal $z_{0min}$ at zero biomass. The new formulation for $z_{0veg}$ is tuned by adjusting $c_{15}$ and $c_{16}$ so as to roughly match reference values in the literature for different land cover types (especially those in Hagemann, 2002). $z_{0min}$ is set to a bare soil value of 0.01 m (Oke, 1987). $c_{17}$ represents the approximate maximum value of $z_{0veg}$ and is assigned a value of 2.5 m, which is representative of tropical rainforests (Sellers et al., 1996b).

$z_{0veg}$ values that lie between these two extremes have been constrained by using the $NPP$-biomass relationship described in section 2.2.9 and by visually comparing reference $NPP$ and $GPP$ values (Cramer et al., 1999; Jung et al., 2011), biome/land cover distribution maps (Ramankutty and Foley, 1999; Olson et al., 2001), and the aforementioned literature values of $z_{0veg}$ for those land cover types.

## 3   How to Couple SEDGES

As stated in section 2.1, SEDGES is a DGVM and is to be embedded within a land surface scheme of a climate or Earth system model. As such, the input and output variables for SEDGES are not the same as those for the land surface scheme that SEDGES forms a part of, because SEDGES has its own set of exchanges with the rest of the land surface scheme in which it lies.

At minimum, the following time-varying input fields are required by SEDGES: aerodynamic conductance ($g_a$), surface

temperature ($T_{sfc}$), surface downwelling (broadband) short wave radiation ($SW\downarrow$), soil temperature at $\approx 0.20$ m ($T_{soil}$), (bulk aerodynamic) potential evapotranspiration ($PET$), soil moisture content ($W_{soil}$), surface pressure ($p_{surf}$), and snow depth water equivalent ($swe$). Most of these variables are either calculated by or made accessible to full-fledge land surface schemes.

Other variables that are used by SEDGES will usually be part of the land surface scheme and depend on additional input variables (not listed in the previous paragraph). This is also the case when using ERA-Interim reanalysis data to force SEDGES

(section 4). In addition, the following parameter values (see table 2) are not specified within the actual SEDGES code, which means that they must be declared and assigned values either in outside modules (e.g., in the non-SEDGES part of the land surface scheme) or by modifying the current SEDGES code: maximum light use efficiency ($\epsilon_{max}$), atmospheric $CO_2$ concentration ($c_a$), the gas constant for dry air ($R_d$), maximum transpiration rate ($trmax$), and residence times for vegetative and soil organic carbon at 10 °C ($\tau_{veg}$ and $\tau_{soil}$).

SEDGES returns the following output fields: live vegetative biomass ($C_{veg}$), soil organic carbon ($C_{soil}$), soil water holding capacity ($W_{max}$), surface wetness factor ($C_w$), transpiration ($T$) (as a diagnostic), evaporation from bare soil (only when snow

is present), canopy resistance ($r_c$), minimum canopy resistance ($r_c min$), surface albedo ($\alpha$), total surface roughness ($z_0$), forest cover fraction ($f_{for}$), leaf area index ($LAI$), leaf cover fraction ($f_{leaf}$), net primary productivity ($NPP$), light-limited gross primary productivity ($GPP_L$), temperature limitation function ($f_2(T_{sfc})$), water-limited gross primary productivity ($GPP_W$), litterfall ($L$), and soil respiration ($R_{soil}$).

In general, the land surface modeling framework that SEDGES presupposes must be reconciled with that of the land surface scheme of the Earth system or climate model that one is incorporating SEDGES into. In particular, in the absence of modification to either SEDGES or the original land surface scheme, the simulation of evapotranspiration according to the simplified mosaic and bulk aerodynamic formulations described in sections 2.1 and 2.2.5 is required. The requirement arises because the water and $CO_2$ exchanges predicted through this framework are presupposed by the canopy resistance equations. The simplest

and recommended scenario with regards to reconciling soil hydrology is that the land surface scheme use a single layer bucket model, because this is what SEDGES "sees" when calculating soil surface and canopy resistances. As an example, Eq. (32), used in our forcing of SEDGES with ERA-Interim data (section 4), gives the standard hydrological formulation for the simple single layer bucket model.

   In the likely scenario that the original land surface scheme of the Earth system or climate model does not perfectly match the

framework used by SEDGES, it should, in many cases, be easy to achieve consistency of frameworks by adapting SEDGES and/or the original land surface scheme. Such adaptation would be easiest for the case in which the land surface scheme uses a single or multiple tile/mosaic approach, because this approach could be converted to the simplified mosaic framework of SEDGES by, at each time step, replacing tile surface temperatures and soil wetness fractions with their respective grid cell weighted averages, and by computing an effective surface roughness for the grid cell (e.g., Mason, 1988) in the case of

significant areal water body coverage.

   Finally, many land surface schemes distinguish between a surface roughness length of water vapor and heat from a surface roughness length of momentum, which is done to correct the typical bulk aerodynamic formulation for its replacement of temperature at the height of the roughness length with surface temperature (e.g., see discussion and references in Chen et al., 1997a). Although SEDGES only computes a roughness length of momentum, there are many ways to convert this roughness

length to the roughness length for vapor/heat (e.g., see analyses and references in Chen et al., 2010).

   An advantage of SEDGES is that (at least in principle) it can be run at any horizontal and temporal resolution. However, we encourage users to run the model using a sub-daily time step so that the generally positive diurnal covariances of light, temperature, aerodynamic conductance, and surface-to-air specific humidity difference can be adequately captured. These covariances should increase ET, overall. On the other hand, If users choose a daily time step or longer, then they should also

anticipate an increase in vegetative productivity in water-limited regions due to the concomitant reduction in water stress. This reduction in water stress can be offset by decreasing $trmax$, the maximum transpiration parameter (appendix B).

## 4 Forcing SEDGES with ERA-Interim Reanalysis Data

In this section, we describe the process by which we force SEDGES using an external forcing dataset, with the goal of evaluating the capability of SEDGES as part of a land surface model. Repeated 1981-2010 6-hourly data from the ERA-Interim reanalysis (Dee et al., 2011) is used as the driver. The following required forcing variables (section 3) were obtained from the
reanalysis for direct input to SEDGES: surface (skin) temperature, surface downwelling short wave radiation, temperature of the second-from-top soil layer, surface pressure, and snow depth water equivalent. Short wave downwelling surface radiation is derived as follows: 3-hourly periods are isolated, and the two periods that straddle each given time (e.g., 12z straddled by the 09z-12z and 12z-15z periods) are averaged to give the flux at that time. A glacier mask is derived by considering any land point that has a daily mean snow depth always greater than zero to be glaciated. No vegetation occurs on glaciated grid points.
Finally, all ERA-Interim data are interpolated to T62 resolution using Climate Data Operators (CDO) before being read into SEDGES.

In order to evaluate the performance of SEDGES, it is necessary to simulate soil hydrology in an interactive way, rather than relying on soil moisture content from the external forcing dataset. (Early versions of SEDGES that were forced using reanalysis soil moisture resulted in unrealistic simulations of vegetation, especially in dry regions.) In order to do so, other land
surface modeling components are combined with SEDGES, such that these components incorporate hydrologically-relevant SEDGES output (namely, soil water holding capacity ($W_{max}$), surface wetness factor ($C_w$), bare soil evaporation (in presence of snow), and surface roughness ($z_0$)) and provide SEDGES with needed variables: aerodynamic conductance ($g_a$), potential evapotranspiration ($PET$), and soil moisture content ($W_{soil}$).

We simulate the aerodynamic conductance outside of SEDGES using these additional variables from the ERA-Interim
reanalysis from the lowest atmospheric model level: the $u$ and $v$ wind components, the specific humidity, and temperature. The aerodynamic conductance and (bulk aerodynamic) $PET$ are calculated according to the formulations of the Planet Simulator model (Lunkeit et al., 2011). Soil moisture content is simulated outside of SEDGES as a prognostic variable using a simple hydrological scheme and three additional ERA-Interim reanalysis variables: precipitation, snowfall, and snow melt. As we recommend above in section 3, we use a single layer bucket model, with bucket depth determined by SEDGES. The bucket
model is simple. As such, runoff is from the surface and only occurs when the soil moisture content exceeds the bucket depth. Infiltration of liquid water from rainfall and from snow melt is unrestricted. We use the discretized version of the following formulation to simulate soil moisture:

$$\frac{\Delta W_{soil}}{\Delta t} = P - S + M - ET_{soil}, \tag{32}$$

where $P$ is precipitation, $S$ is the rate of snowfall, and $M$ is the rate of snow melt (all in liquid water equivalent). Again, $W_{soil}$
cannot be greater than $W_{frac}$, with any excess going into runoff. Here, $ET_{soil}$ is the combined soil evaporation and transpiration (and thus not coming from the snow cover). Recall from section 2.2.5, that when there is snow cover, soil evaporation ($E_{soil}$) is calculated by SEDGES. Under conditions of sufficient snow cover (see below for more), transpiration is 0, and $ET_{soil}$ reduces from being the (total) $ET$ to being $E_{soil}$.

Soil hydrology in the framework we have constructed for driving SEDGES requires careful treatment in the presence of non-zero snow depths. Such care is especially needed because of the tile-based TESSEL land surface scheme that is used by ERA-Interim and the data interpolation to the coarse T62 grid. The underlying TESSEL tile scheme and the spatial interpolation imply that data for a given grid cell that is fed into SEDGES can be physically inconsistent in the SEDGES framework

(section 2.1). The inconsistency arises because a grid cell can have a non-zero snow depth and yet have a homogeneous (as required and seen by SEDGES) surface temperature that is well-above $0\,^\circ$C. This situation, in the absence of any modification to the SEDGES code, gives rise to common occurrences in which substantial transpiration is occurring from snow-covered leaves and in which snow-covered vegetation and ground are evaporating as if their surface temperatures were well above freezing. These occurrences are not only unphysical; they give rise to higher simulated ET from a given snowy, above-freezing

region than would occur over the same region if it were divided up into its original small, homogeneous but differing tiles.

A second issue that must be addressed when there is snow is the tendency for ERA-Interim to over-predict freezing rain at the expense of snowfall (Dutra et al., 2011). Because we treat rain as entering the soil without interception by the snow pack (as is done in the TESSEL land surface model (Dutra et al., 2010)), the freezing rain bias led our early simulations to have unrealistically large recharge of soil water reservoirs in winter in many locations, even though surface temperatures were well

below $0\,^\circ$C. Handling of the aforementioned snow-temperature inconsistency and the freezing-rain bias are discussed next.

In order to simply resolve the above issue of snow with above-freezing temperature, we separate possible conditions into two cases: 1) $swe > snowthresh$ and 2) $swe \leq snowthresh$, where we define $snowthresh$ as the $swe$ threshold above which the surface is treated as snow-covered with respect to evaporation from and physiology of vegetation and the threshold at or below which the surface is treated as snow-free. In case 1, we assume that transpiration is negligible (due to combined

cold temperature and physical coverage of leaves by snow) and thus introduce a slight modification to Eq. (2), by setting $f_2(T_{sfc})$ to 0 (its value at freezing), which, in turn, forces the productivity to 0. In case 2, we assume that evaporation from snow is negligible, and $f_2(T_{sfc})$ is as normal. Finally, we address the excessive partitioning of precipitation into liquid form at subfreezing temperatures by further assuming for case 1 that soil moisture cannot increase unless there is some snow melt during the same time step. Snow melt indicates the presence of above-freezing temperatures and, thus, the presence of liquid

water that does not freeze on contact with the surface and can thus infiltrate into the soil.

## 5  Model Evaluation

In this section we evaluate SEDGES as part of the simple hydrological scheme described in section 4, forced with ERA-Interim reanalysis data. We emphasize the results from the simulation forced with historical $CO_2$, using reference datasets of vegetative carbon, LAI, surface albedo, tree cover fraction, soil organic carbon, GPP, and evapotranspiration.

Four simulations are carried out: three equilibrium simulations and one transient simulation. In the three equilibrium simulations, atmospheric $CO_2$ levels of 280, 360, and 560ppm, respectively, are held fixed through the simulations. The vegetation is spun up from a vegetative- and soil organic carbon-free state. The spin up includes an acceleration of the carbon cycle for the first 260 years (only). From simulation year 261 to year 1500, the carbon cycle is run normally. The model is very nearly

at equilibrium by the last 30 years in all three runs, and these years are thus analyzed. The transient simulation begins with the end state of the 280ppm equilibrium simulation and is subsequently forced using observed, transient atmospheric $CO_2$ values from 1832 to 2010. The $CO_2$ data is taken from the Mauna Loa dataset (Keeling et al., 1976) for 1959 and on and from 20 year-smoothed ice core data (Etheridge et al., 1998) for the prior years. Unless stated explicitly, the results and analyses always refer to the transient $CO_2$ simulation. The model performs reasonably well, overall; GPP is simulated exceptionally well. In the evaluation process, we concentrate on the comparison of SEDGES with observation-based data products, but we sometimes supplement these comparisons with comparisons to land surface models and their performances, especially to the ENTS model (Williamson et al., 2006) because it is of similar complexity as SEDGES.

## 5.1 Evaluation of Productivity

GPP that is simulated by reanalysis-forced SEDGES is evaluated against two observation-based datasets: MTE (Jung et al., 2011) and CARBONES. In addition, we use the model intercomparisons of (Sitch et al., 2008), (Piao et al., 2013), Anav et al. (2015) and modeling-based results of Hemming et al. (2013) and Holden et al. (2013) to assess the relative performance of SEDGES.

For the 1982-2008 period, SEDGES simulates a global mean annual GPP of 126 $PgC\ yr^{-1}$ as compared to 120 $PgC\ yr^{-1}$ for the observation-based MTE dataset (Jung et al., 2011). Zonal means of GPP, taken over the glacier-free land mask obtained from the ERA-Interim data and for 1990-2009, are shown for SEDGES, MTE, and the CARBONES data assimilation dataset in Fig. A6. SEDGES captures the large scale patterns very well. Compared to the two reference datasets, SEDGES has positive productivity biases in most of the Southern Hemisphere and around 35 °N, and it has negative biases near the equator and in the high northern latitudes. Moving to the full spatial field, we can see in Fig. A7 that the annual mean GPP, for 1990-2009, in SEDGES and MTE compares very well, overall, with those of MTE and CARBONES. Regionally, as compared to the two reference datasets, SEDGES simulates too high GPP in western Argentina, non-desertic tropical Africa, around the Korean Peninsula, the southwestern and eastern Australian coasts, and too little GPP in almost all the equatorial tropics and Amazonia (see section 5.6 for discussion), the Pacific northwest of the United States, northwestern Europe, and in parts of north-central and far eastern Siberia. In Siberia, SEDGES simulates too low GPP in a dry region northeast of the Kolyma Mountains. This underestimate may be partially due to SEDGES's neglect of leaf mitochondrial respiration in the light, which has been found to be a large fraction (up to $\approx 0.3$) of gross photosynthesis in Arctic tundra plants (McLaughlin et al., 2014; Heskel et al., 2014). Indeed, a fraction of 0.3 would, under purely water-limited conditions, increase GPP by 43%.

The spatial correlations between SEDGES and the reference datasets are as high as can be expected (table 3). Spatial correlations between multi-year annual means of GPP from MTE and three offline land surface models (ORCHIDEE, JULES, and CLM4CN) range from $\approx 0.87$ to 0.95 (Anav et al., 2015), whereas it is 0.92 for SEDGES. The same correlations between the three models and CARBONES ranges from $\approx 0.83$ to 0.87 (Anav et al., 2015), whereas it is 0.86 for SEDGES.

[Table 3 about here.]

SEDGES simulates well the multi-year monthly means of GPP, as shown by comparison with those for the MTE dataset in Fig. A8 for different latitudinal bands. From this figure, we can see that SEDGES captures well the seasonal progressions of GPP. A noticeable departure of SEDGES from MTE, however, is a slight phase shift in the northern hemisphere extratropics, with a generally earlier spring productivity increase in SEDGES than in MTE. We suspect that the phase shift is due to the absence of both winter-deciduous leaf phenology and physiological constraints on speed of acclimation to temperature (discussed more in appendix A). Moving to the full spatial field, Fig. A9 shows that the correlations in the seasonal cycle between SEDGES and MTE are generally high in areas with strong seasonality of productivity such as the mid to high latitudes, India, and the dry tropics. Temporal correlations are weaker, but still generally positive, in semi-arid regions and in Amazonia. Correlations are often negative in regions with low absolute seasonal variation in GPP such as equatorial Africa, the northern hemispheric deserts, and southern Australia. SEDGES' problems in the dry regions may be due to the simple single-layer bucket soil hydrology that was used. The negative correlation in equatorial Africa is due to insufficient moisture stress in that region in ERA-Interim-forced SEDGES. The lack of moisture stress in this region causes dry season GPP to exceed that of the wet season due to the reduced cloud cover, as is seen in the wetter parts of the Amazon (Wu et al., 2016). However, the real African equatorial forest experiences a drop in GPP during the dry season relative to wet season (Guan et al., 2015). Thus, the real world behavior in equatorial Africa is anti-correlated with that in the model. In equatorial Africa, the lack of moisture stress in ERA-Interim-forced SEDGES is likely attributable to a pronounced positive precipitation bias in the ERA-Interim forcing data in that region (Lorenz and Kunstmann, 2012; Dolinar et al., 2016).

The interannual variability of global GPP for 1990-2009 in SEDGES is 1.79 $\mathrm{PgC\ yr^{-1}}$, whereas it is 2.50 $\mathrm{PgC\ yr^{-1}}$ for the CARBONES dataset. Anav et al. (2015) report values of 3.23, 4.4, and 2.87 $\mathrm{PgC\ yr^{-1}}$ for offline-driven land surface models, ORCHIDEE, JULES, and CLM4CN, respectively. The interannual variability of GPP at each grid point is shown in Fig. A10 for SEDGES and CARBONES. SEDGES successfully captures the spatial pattern of the CARBONES reference dataset, although its magnitudes are smaller, overall. These smaller magnitudes are probably attributable to a number of factors. Fig. A10 shows that SEDGES most severely underestimates interannual GPP variability in cropland regions (c.f. Ramankutty et al., 2008), which can differ greatly in form and function from the natural vegetation simulated by SEDGES. Secondly, the underestimate of mean annual GPP by SEDGES in some regions such as northern Asia (Fig. A7) likely contributes to the model's underestimate of GPP variability in those same regions. Finally, SEDGES applies the same temperature limitation function to productivity (appendix C) in all situations. In so doing, the model underestimates physiological constraints on the speed with which vegetation can adapt to new thermal regimes. Although in real life these constraints apply mostly at daily time scales, interannual variation in the frequency of extreme temperature events probably has greater impact on annual GPP in the real world than in SEDGES.

The trend in annual GPP for 1990-2009 is captured extremely well by SEDGES. The CARBONES reference dataset yields a global annual GPP trend of 0.086 $\mathrm{PgC\ yr^{-2}}$, whereas SEDGES gives a value of 0.080 $\mathrm{PgC\ yr^{-2}}$. The spatial pattern of the trend is seen in Fig. A11. SEDGES captures the spatial pattern of GPP trend in the CARBONES dataset better than ORCHIDEE, JULES, and CLM4CN, as reported by Anav et al. (2015). SEDGES generally outperforms the other models in Africa, Australia, the semi-arid areas of eastern Europe, and central and eastern Asia. Notable areas where SEDGES (but not

the other models) deviates from CARBONES include Amazon and parts of northwestern North American, in which SEDGES simulates more negative trends in GPP than the reference dataset. The negative GPP trend in SEDGES in the Amazon is very possibly due, at least in part, to its simplifications affecting $CO_2$ fertilization of productivity, as is explained next.

The SEDGES treatment of $CO_2$ fertilization is seen via Eqs. (3) and (6) and notably lacks a direct dependency on temperature. However, Long (1991), using the widely-used Farquhar model of photosynthesis (Farquhar et al., 1980), finds that, for a given $\frac{c_i}{c_a}$, the proportional increase in leaf photosynthesis with elevated $CO_2$ increases substantially with temperature under both Rubisco- and light-limited conditions. In line with that result, a modeling study at the scale of a global grid (Hickler et al., 2008) which uses a photosynthesis model based on that of the Farquhar one, finds the fertilization response of NPP to elevated $CO_2$ increases substantially, too, when moving from boreal forest to temperate forest to tropical forest. Hence, if SEDGES were to use a more complex parameterization of photosynthesis that is based on the Farquhar model, one would expect the Amazon region to show a more positive GPP trend in Fig. A11 and thus be closer to the CARBONES data.

In spite of the simplifications affecting $CO_2$ fertilization in SEDGES, its global productivity response to $CO_2$ forcing is comparable to that of other models of the land surface and vegetation. The results of transient experiments in which 10 such models are forced offline with repeated early 20th century climate and with increasing $CO_2$ values show relative increases of global NPP per $ppm$ $CO_2$ of 0.05% to 0.20% (with mean of 0.16%) for the 1980-2009 period (Piao et al., 2013); it is 0.102% for SEDGES. Similar transient simulations with 5 of the aforementioned models (but also including projected $CO_2$ levels) (Sitch et al., 2008) give global GPP increases[7] from 280 to 360$ppm$ $CO_2$ that range from approximately 14% to 19%; they give global GPP increases from 360 to 560$ppm$ $CO_2$ that range from approximately 23% to 36%. In comparison, the transient run of SEDGES gives an increase in global GPP from 280 to 360$ppm$ $CO_2$ of 20%, and an at-equilibrium global GPP increase from 360 to 560$ppm$ $CO_2$ of 30%. Finally, Hemming et al. (2013), approximating the purely physiological effect[8] of $CO_2$ on productivity using a GCM, finds an increase in equilibrium GPP of 75% when doubling $CO_2$ from near-pre-industrial values. A most likely increase in global NPP of 40% to 60% with $CO_2$ doubling from pre-industrial levels is found by Holden et al. (2013), using a combination of global atmospheric $CO_2$ rise and estimated land use changes since the pre-industrial. Doubling pre-industrial $CO_2$ (280 to 560$ppm$) in equilibrium simulations in SEDGES induces an NPP increase of 55%.

## 5.2 Evaluation of Vegetative Carbon

Vegetative carbon is simulated reasonably well by SEDGES. Unless stated otherwise, we compare the means of the last 30 years of the $CO_2$-varying run with an older reference dataset (Olson et al., 1985). The Olson dataset was chosen because its data sources reflect the potential natural vegetation better than the sources of the more recent NDP-017b dataset (Gibbs, 2006). SEDGES has an equilibrium global vegetative carbon of 530 $PgC$ under pre-industrial levels of $CO_2$, and a mean of 615 $PgC$

---

[7]visually estimated from their Fig. 9

[8]In addition to the direct impact of $CO_2$ on GPP, some vegetation feedback onto GPP via climate is included here; also, vegetation distributions and LAI are held fixed.

for 1981-2010 of the transient $CO_2$ run. In comparison, the Olson et al. (1985) dataset and the more recent Gibbs (2006) dataset give a global vegetative carbon of 451 PgC and 560 PgC, respectively (Jiang et al. (2015), citing Gibbs (2006)).

Figure A12 shows the distributions of mean vegetative carbon for SEDGES (1981-2010) and that for the Olson et al. (1985) dataset. The spatial pattern of Olson vegetative carbon is captured well by SEDGES (table 3: R = 0.57, $R^2$ = 0.33). The RMSE (root mean squared error) between SEDGES and the Olson datasets is 3.92 kgC m$^{-2}$. (We calculate RMSE in this study by taking the square root of an area-weighted mean of the squared differences between SEDGES and the reference at each grid point.) Among the discrepancies, two tendencies prevail: the almost-direct relationship between GPP biases and vegetative carbon biases, and an overall tendency to overestimate vegetative carbon in the tropics and to underestimate it in the mid to high latitudes. SEDGES has large positive vegetative carbon biases in most of the tropics (except the western Amazon), southeast North America, China, and northwestern Europe. One should note that many of these biases reflect land cover change from forests to croplands (Ramankutty and Foley, 1999). In addition, positive biases in the wet-and-dry and semi-arid tropics are partially attributable to the absence of a fire module in SEDGES; fire can have a large impact on vegetative cover in these regions (Bond et al., 2005). SEDGES has large negative biases in western Canada and central Siberia. The subarctic is generally negatively-biased. These vegetative carbon biases are due to a combination of the aforementioned GPP biases and the globally-uniform residence time for vegetative carbon in SEDGES (table 2). The SEDGES 10 year residence time for vegetative carbon matches the observation-based estimate ($\equiv \frac{C_{veg}}{NPP}$) given by Jiang et al. (2015).

Compared to SEDGES, ENTS (Williamson et al., 2006) simulates vegetative carbon slightly closer to the Olson reference, overall. ENTS has a strong negative biomass bias in eastern Australia, and it greatly overpredicts biomass in eastern Brazil and northeastern Canada; SEDGES predicts these regions fairly well. On the other hand, ENTS also predicts more biomass in central Siberia (i.e. has a weaker negative bias), simulates western Canada well, has a less severe negative biomass in far eastern Siberia and Alaska, and has a less severe positive biomass bias in the tropics.

## 5.3   Evaluation of soil organic carbon

At present, land surface models have great difficulty in simulating soil organic carbon well. Tian et al. (2015), using 10 offline-forced terrestrial biosphere models at 0.5° resolution, find highly diverging model estimates of soil organic carbon dynamics as well as systematic biases due to absent processes affecting high latitude soil organic carbon stocks. The models in Tian et al. (2015) have Pearson correlation coefficients under 0.4 with respect to the Harmonized World Soil Database (HWSD) reference dataset (Wieder et al., 2011). SEDGES performs better with a 0.58 correlation (at the coarser T62 resolution), although it should be kept in mind that model performance in simulating soil organic carbon has been found to improve dramatically when aggregating at large spatial scales (Todd-Brown et al., 2013). As listed in table 3, SEDGES has a 7.9 kgC m$^{-2}$ RMSE with respect to the HWSD data.

Figure A13 shows the full spatial distribution of soil organic carbon for SEDGES and the HWSD reference dataset. Compared to HWSD, SEDGES simulates less soil organic carbon in semi-arid and arid areas, in the more northern parts of the Arctic, and generally more soil organic carbon in the remaining areas. Compared to the ENTS model (Williamson et al., 2006), SEDGES gives a broadly similar simulation of soil organic carbon, as expected since their soil respiration formulations are

almost identical. Some notable differences between the two are that SEDGES generally simulates more soil organic carbon than ENTS; SEDGES predicts too much soil organic carbon in southeastern North America, southeastern Asia, and the tropics, whereas ENTS generally does not; in high northern latitudes, ENTS (only) simulates a pattern of very high values in far eastern Siberia, Alaska/Yukon, and northeastern Canada with lower values in between, but this pattern is not seen in the HWSD

dataset.

Taking the globe as a whole, SEDGES has a residence time for soil organic carbon of 25.7 years and a time-averaged soil organic carbon of 1516 PgC for 1981-2010. Respective estimates for these values based on HWSD data are 24 years for residence time (Todd-Brown et al., 2013) and range from 891 PgC to 1657 PgC for total terrestrial soil organic carbon storage (Tian et al., 2015). Köchy et al. (2015) estimate total terrestrial soil organic carbon at a higher value of $\approx 3000$ PgC because

they include estimates of carbon in the deeper soil layers that HWSD does not.

## 5.4   Evaluation of LAI

We use reprocessed Moderate Resolution Imaging Spectroradiometer (MODIS) data from 2001 to 2010 from Bejing Normal University (BNU) (Yuan et al., 2011) as the reference set. We interpolated this data to T62 resolution and derived multi-year monthly means from it. Outside of the tropics, we compare only the maximum of the multi-year monthly means of LAI

between SEDGES and the reference data because of the absence of cold deciduous phenology in SEDGES and because serious deficiencies with MODIS in capturing the seasonal cycle of LAI in boreal coniferous forests have been found (Serbin et al., 2013).

Figure A14 shows the maximum multi-year monthly mean LAI in SEDGES and the BNU MODIS-based reference, for the years, 2001 to 2010. SEDGES has strong positive biases in southeastern and extreme northern South America, in parts of

Africa, in parts of southern and eastern Asia, in northern coastal Australia, and in parts of Europe. Many of these LAI biases lie where positive GPP biases occur and/or where croplands have replaced natural forests. Strong negative biases occur in parts of northern Asia. Of these, the biases east and northeast of Lake Baikal occur in the absence of negative GPP biases. Overall, the spatial patterns of maximum LAI in SEDGES and the MODIS-based reference have a correlation of 0.793.

Figure A15 shows the temporal correlation in the tropics between the multi-year monthly means of SEDGES and the BNU

MODIS-based reference (i.e. the correlations of the two season cycles). In spite of the crude scheme for drought-deciduous phenology described in section 2.2.8, SEDGES does a reasonable job in simulating the seasonal cycle of LAI in much of the tropics. Negative correlations (indicative of poor model performance) occur near the equator, especially in the Amazon region, where LAI varies modestly, seasonally, in the real world (e.g., Wu et al., 2016). The wet parts of the Amazon generally experience high water availability, even in the dry season. As a result, LAI there is decoupled from soil moisture-induced

water stress (in contrast with the drier portions of the Amazon), and LAI, generally, is increasing during the dry season and decreasing during the wet season (Wu et al., 2016). SEDGES cannot capture this behavior.

## 5.5 Evaluation of Surface Albedo

Figure A5 shows monthly mean climatologies of SEDGES surface albedo and MODIS white sky albedo (NASA LP DAAC, b)[9] for the months of January, April, July, and October for the years 2001 to 2010. From these figures, we can see that SEDGES captures the overall spatial albedo pattern fairly well. In fact, spatial correlations between SEDGES and the reference range from a low of 0.77 in July and August to a high of 0.91 in April. The high correlation in boreal spring indicates relative strength in simulating the winter melt season at mid to high northern latitudes. The relative weakness of SEDGES in Northern Hemispheric summer is due to the absence of a strong snow signature on the land and relative weakness of the model in correctly capturing the second and third order determinants of (non-glacier) terrestrial albedo, since these determinants come to the fore during boreal summer. Three such determinants are now discussed.

The most significant model deficiency in albedo simulation that gains prominence during boreal summer is the simulation of bare soil surface albedos. Recall from section 2.3.1 that bare surface albedo in SEDGES is determined by soil organic carbon content and by the albedo of soil in the absence of soil organic carbon ($\alpha_{sand}$), which is assumed constant. In the real world, geological processes, the quantity and properties of dead biomass litter, and the albedo of the underlying bedrock greatly impact $\alpha_{sand}$ as well (Knorr and Schnitzler, 2006; Vamborg et al., 2011). The absence of these real world processes in SEDGES gives rise to an underestimate of albedo in the Arabian and the Saharan deserts and an overestimate of albedo in the snow-free season in most of the world's remaining deserts (Fig. A5). MODIS values of snow-free albedo (NASA LP DAAC, a) in the polar desert (not shown) vary generally from 0.12 to 0.22 in July. The negative soil organic carbon bias in SEDGES in the high Arctic (section 5.3) causes the snow-free albedo to be too high in polar deserts of the high Arctic.

A second major model deficiency in boreal summer albedo simulation stems from SEDGES's inability to distinguish between low albedo ($\approx 0.09$) evergreen coniferous forests and high albedo ($\approx 0.16$) mid or high latitude broadleaf deciduous forests. This deficiency is seen most clearly in July (Fig. A5) in eastern North America, in which SEDGES fails to capture the south-to-north decrease in albedo from the temperate to subarctic regions.

The third significant model deficiency in the simulation of albedo during boreal summer is a negative bias in areas that are grassland or savanna in the real world, particularly those found in Mongolia, the U.S. Great Plains, and tropical Africa. In the latter two regions, this negative albedo bias is associated with a positive bias in vegetative carbon (section 5.2) and LAI (Fig. A14).

Outside of boreal summer, positive biases in vegetative carbon in the mid to high latitudes also cause a lowering of SEDGES January albedo as compared to MODIS (Fig. A5). The most affected regions are in central Asia and the North American

---

[9]The original MODIS data were processed by spatially interpolating to fill in missing data, taking multi-year means (i.e. means across multiple years) of the 23 16-day periods for each year, spatially interpolating these into T62 resolution, temporally interpolating the 16-day periods linearly to obtain 24 new such periods such that each month was covered by two periods, taking the mean of the two periods for each month, masking out albedo values for each month and region that had polar night-caused missing values in its original pre-interpolated comprising data, and masking out water and land ice grid points in the ERA-Interim reanalysis T62 data that was used to forced SEDGES. White sky albedo from MODIS was used as the reference dataset because SEDGES's albedo formulation lacks dependency on the zenith angle of the direct beam.

prairies and High Plains.[10] In March (not shown), there is a positive albedo bias in northern Siberia (specifically, in transition areas between boreal forest and tundra) whose source cause is a negative GPP bias seen in Fig. A7. This positive bias carries over, albeit less strongly, into April (Fig. A5).

Finally, it must be noted that our albedo evaluations of SEDGES have been with respect to white sky albedo, and not actual albedo.

## 5.6 Evaluation of Evapotranspiration and Runoff

Evapotranspiration from SEDGES is compared to that of a multi-source compilation reference dataset, Mueller et al. (2013), for the period from 1989 to 2005. Simulated evapotranspiration by SEDGES qualitatively follows that of the reference dataset, successfully capturing the first order seasonal zonal mean pattern and annual pattern (Figs. A16 and A17). Some salient deficiencies exist, however.

Globally, the annual means in Figs. A16 and A17, show that ET is excessive, overall. Future versions of SEDGES will improve this bias, possibly through inclusion of a more realistic treatment of the ratio of intercellular to atmospheric $CO_2$ ($\frac{c_i}{c_a}$) instead of the currently fixed value of 0.80, which is probably too high, in general. The important role of $\frac{c_i}{c_a}$ in transpiration is discussed more below and in section 6. On the other hand, the percentage of global evapotranspiration coming from transpiration is 64% in SEDGES, which compares very well with a literature-based estimate of 61% (with 15% standard deviation), derived from 81 studies (Schlesinger and Jasechko, 2014).

Regionally, the most severe discrepancy from the Mueller et al. (2013) reference dataset is the excessive ET simulated by SEDGES in the mid- to high northern latitudes, which is most pronounced in the transition season from late winter to early spring (Fig. A16). This ET bias occurs because the new parameterization (see section 2.2.5) to reduce excessive sublimation from snow-covered surfaces in the original SimBA parameterization is only moderately effective. This deficiency is possibly the most crucial one to improve in future model versions, possibly by implementing a scheme to decrease surface roughness from snow-covered terrain (e.g., Cox et al., 1999). Another major contributor to the mid- to high northern latitude positive ET bias is the absence of winter-deciduous leaf phenology and the assumed instantaneous temperature acclimation in SEDGES (appendix A), which together increase spring GPP (section 5.1) and thus, concomitantly, increase transpiration. Additionally, Fig. A18, which shows July ET for SEDGES and the Mueller et al. (2013) dataset, suggests that snow-free bare soil evaporation from moist areas is excessive, as indicated by the relatively large positive ET biases in the high northern latitudes, which have a large contribution to total ET from soil evaporation due to their moist soils and low LAI (Fig. A14) (and hence low leaf cover fraction).

In addition to the northern mid- to high latitude positive bias, SEDGES suffers from a negative ET bias in the equatorial region (Fig. A16), which has at least three likely causes. A positive bias in annual cloud cover fraction in the ERA-Interim reanalysis data (Dolinar et al., 2016) results in a negative bias in the incident surface shortwave radiation field (Zhang et al., 2016) on the order of $\approx 5\%$ to 30% throughout most of the equatorial tropics for at least 6 months of the year. This bias may

---

[10]Even though these areas would be mostly grasslands rather than cropland if not for human land cover change (Ramankutty and Foley, 1999), the winter albedo would be similarly high in either case.

result in a relative reduction of GPP of similar magnitude in SEDGES, which would help to explain the negative GPP bias that SEDGES simulates here (seen in Figs. A6 and A7), while also explaining a reduction in transpiration of comparable (but lesser) magnitude. The latter would result from the increased canopy resistance (Eq. 13) that would ensue from the decreased demand for $CO_2$ uptake due to the lower light. Lack of an increase in light use efficiency under cloudy skies in SEDGES

further exacerbates the negative GPP bias caused by the surface shortwave reduction (see section 6 for more discussion). The second reason for low ET near the equator is the complete absence of canopy interception loss in SEDGES, since this source of surface evaporation has been found to comprise around 20% of total ET in tropical rainforests (Shuttleworth, 1988; Da Rocha et al., 2004; Czikowsky and Fitzjarrald, 2009; Miralles et al., 2011)[11]. Finally, the low vapor pressure deficits in the wet tropics do not activate a mechanism in SEDGES that, in other models, would increase transpiration. Less transpiration is simulated by

SEDGES in this region because SEDGES's simplifying assumption of a fixed $\frac{c_i}{c_a}$ promotes underestimation (overestimation) of transpiration when vapor pressure deficits are low (high), due to the increase of $\frac{c_i}{c_a}$ with decreasing vapor pressure deficit (Medlyn et al., 2011, 2012; Prentice et al., 2014). It then follows from Fick's law of diffusion (at the plant scale) that higher $\frac{c_i}{c_a}$ decreases the leaf-to-air $CO_2$ gradient, thus requiring that stomatal conductance increase to ensure sufficient $CO_2$ inflow for photosynthesis. The higher stomatal conductance increases transpiration. A similar process happens at the canopy scale.

Conspicuously high ET can be seen toward the end of the wet season in both the northern and southern edges of the tropics in Fig. A16. A very similar zonal mean temporal pattern is found in transpiration in the STEAM land surface model (Fig. 6 of Wang-Erlandsson et al., 2014). Because this model was evaluated using ERA-Interim forcing data, the high ET anomalies that we see in Fig. A16 may be attributable to this particular forcing; the ET anomalies in SEDGES may also be attributable to the lack of canopy interception loss, which declines in tandem with precipitation toward the end of the wet season in Wang-

Erlandsson et al. (2014), thus reducing the magnitude of the ET aberrations.

On a smaller regional scale, areas of complex terrain and topography (the Andes and the Himalayas) have very strong positive annual mean ET biases (Fig. A17). We believe that this is an interpolation artifact stemming from cold, snow-covered grid cells lying adjacent to warm, snow-free grid cells. With spatial interpolation of the ERA-Interim to the coarse T62 resolution, such cases often result from deep snow being distributed among the entire area and having a relatively warm temperature due to the

influence of the low-lying grid cells at the original high resolution. This set up results in strong evaporation.

With respect to other models of similar complexity, SEDGES simulates ET much better than the original SimBA (see appendix C). However, at least when looking at the climatological annual mean, ET is considerably better-simulated by ENTS (Williamson et al., 2006) than by SEDGES. In general, ENTS has a negative ET bias in dry regions and a positive bias in moist regions, especially in parts of the tropics. However, these deviations are generally no greater than the biases that SEDGES has.

Simulated runoff is compared to the composite gridded dataset from the University of New Hampshire (UNH) and World Meteorological Organization's Global Runoff Data Centre (GRDC) (Fekete et al., 2002). SEDGES simulates runoff well except for excesses in the equatorial tropics and in the mid latitudes of the Southern Hemisphere (Fig. A19), especially in equatorial Africa and in the Andes (Fig. A20). Excessive runoff in the equatorial tropics is due to the aforementioned negative ET bias,

---

[11]Here, we combine in-situ precipitation and ET data from Da Rocha et al. (2004) with a ground-observed interception loss-to-precipitation ratio of 0.116 from nearby (as reported by Czikowsky and Fitzjarrald, 2009) to get an interception loss-to-ET ratio of 0.22.

but it is also attributable to a general overestimate of precipitation in these latitudes by the ERA-Interim reanalysis, especially in equatorial Africa (Lorenz and Kunstmann, 2012; Dolinar et al., 2016). Global runoff for SEDGES, excluding areas of land ice, is about $44 \times 10^{15}$ kg yr$^{-1}$, which is slightly higher than the $38 \times 10^{15}$ kg yr$^{-1}$ for the UNH/GRDC dataset and the global (excluding Antarctica) runoff estimate of $37 \times 10^{15}$ kg yr$^{-1}$ by Dai and Trenberth (2002).

It should be noted that the parameterization for soil water holding capacity (section 2.3.2) gives generally much lower values (not shown) in the wet and dry tropics than in the reference dataset (Kleidon and Heimann, 1998; Hall et al., 2006; Kleidon, 2011). Such lower values of soil water holding capacity should lead one to expect a positive runoff bias in these areas in SEDGES. This is not the case, however, when we compare those regions of the wet and dry tropics where the ERA-Interim precipitation forcing has insignificant bias (and where biases in SEDGES runoff are thus unlikely to be attributable

to precipitation deficiencies in the forcing). The lack of runoff bias in these regions suggests that the parameterization of soil water holding capacity in SEDGES (Eq. 29) works well in tandem with its ET scheme and the simple soil bucket hydrology (Eq. 32).

Finally, while it might seem easy to attribute deficiencies in the SEDGES hydrology to our use of the single layer soil bucket, this type of model has been found to perform well under many conditions: wet soils, when precipitation events tend to be large

relative to the bucket depth, and under strong root compensation or hydraulic redistribution of soil moisture (Guswa et al., 2002; Guswa, 2005). Even when the aforementioned conditions do not hold, distortions stemming from use of the single layer bucket model are comparable to those coming from other aspects of SEDGES hydrology.

## 5.7 Evaluation of Forest Cover Fraction

The overall global pattern of tree cover fraction of the International Satellite Land Surface Climatology Project (ISLSCP II)

reference dataset (DeFries and Hansen, 2009) is captured reasonably well by SEDGES, as seen from Fig. A21. However, SEDGES has almost double the amount of arboreal cover. In Europe, China, and India, this is at least partially due to anthropogenic land cover change of natural forests and woodland to croplands (Ramankutty and Foley, 1999). Aside from this deforestation, the model's lack of competing plant functional types (especially boreal trees, warm region trees, and grasses) in conjunction with the constant residence time of 10 years for all vegetative carbon, leads to an overestimation of arboreal

vegetation by SEDGES in South America, Africa, and North America, as well as a slight underestimation in boreal forest regions. A noted limitation of the reference dataset is its underestimation of real world tree cover in areas with the highest cover fraction. Thus, SEDGES' overestimates in these regions may partially be due to this phenomenon, as well. Finally, a less inclusive definition for arboreal cover in the reference dataset than in SEDGES contributes to the cover overestimate in the latter. SEDGES assumes that forest is vegetated land that protrudes above the winter snow pack, whereas the ISLSCP II

dataset requires that trees be at least 5 m tall to count toward tree cover and thus excludes short trees and tall shrubs that do not get completely buried by snow.

## 5.8 Model Evaluation: Summary and Conclusions

The performance of the new SEDGES model has been evaluated for the present day through comparison with numerous reference land datasets. To do this, we have forced SEDGES offline with ERA-Interim reanalysis data, using our recommended simple single layer bucket soil hydrology to simulate terrestrial water storage, incorporating a simple scheme to handle a snow-temperature mismatch in the forcing data, and adopting the Planet Simulator's formulation for aerodynamic conductance. Note that the particular forcing data and hydrological implementations impact the analyzed output of the SEDGES simulations. In particular, the spin up using repeated climatology from 1981-2010 yields conditions that differ from a spin up using purely historical data. Not only do forcing and hydrology affect the model evaluation, but so do the reference datasets against which it is compared, since even they are not perfect representations of reality. Given all of these particularities, one must view the presented evaluation as a conditional, non-definitive, but yet informative guide to the strengths and the weakness of the model.

The following output variables of SEDGES have been examined: $GPP$, vegetative carbon, soil organic carbon, $LAI$, surface albedo, $ET$, runoff, and forest cover fraction. In comparison to the respective reference datasets, SEDGES simulates each variable at least reasonably well. Relative model strengths lie in the simulation of $GPP$; relative weaknesses in $ET$, snow-free albedo, and forest cover fraction. Given the simplicity of its formulation, the strength in $GPP$ is unexpected, particularly the ability of SEDGES to capture the spatial patterns of temporal trends in GPP. Simulated $ET$ is too high in the mid to high northern latitudes and is too low in the wet tropics. SEDGES captures albedo patterns fairly well outside of boreal summer, but subtler differences among the different desert types and among the non-desert areas are not well-captured during the snow-free season. Simulated forest cover fraction is generally too high.

## 6 Discussion and Conclusions

A new simplified model for the representation of dynamic ecological processes for use in conjunction with climate models has been developed. This new model was combined with a simple hydrological scheme and forced with reanalysis data. In evaluating its performance, we concentrated on the comparison with present-day observation-based data products, while also referring to land surface models (especially ENTS) and their performances for a relative comparison of the strength and weaknesses of SEDGES.

The quantitative comparison highlighted strengths and weaknesses of the model. A notable strength is that SEDGES's simulation of gross primary productivity is comparable to and sometimes better than that of state-of-the-art dynamic global vegetation models. Our evaluation has also shown that SEDGES performs well in a number of other metrics. Overall, the results show that SEDGES can be used to adequately simulate modern land surface characteristics, including input variables to land-atmosphere coupling schemes used in climate models, as well as key variables of the hydrological and terrestrial carbon cycles.

The most severe weaknesses of our offline-run SEDGES are the aforementioned strong positive evapotranspiration (ET) bias that it yields in the mid-to-high latitudes in winter and early spring and its over-prediction of arboreal cover. The latter results from the use of only one plant functional type, which carries over from the SimBA model, on which SEDGES is based.

Even an expansion to just two plant functional types (e.g., tree and herbaceous) and simulation of competition between them would entail a significant increase in model complexity. However, SEDGES's use of only one plant functional type, along with a fixed relationship between biomass and both wet soil $LAI$ and rooting depth hydrology, excessively constrains large scale vegetation structure by excluding the emergence of location-adapted landscape characteristics. As result of these issues, simulated vegetation in water-limited regions and wetness-induced changes in biomass should be treated as more heuristic than definitive. Returning to ET, although less severe, the positive ET bias is also seen in the mid-to-high latitudes during the snow-free season. In contrast, the equatorial tropics have a strong negative ET bias due to high cloud cover and concomitant low insolation, which increases canopy resistance and thus reduces transpiration (as seen in Eq. 13). This negative bias occurs for various reasons, including SEDGES's lack of distinction between sunlit and shaded leaves (see below), lack of evaporation from canopy-intercepted water in SEDGES (see section 5.6 for more), and a positive cloud cover and a negative surface insolation bias in the forcing dataset (see section 5.6 for more). The last of these may also be associated with further reductions in transpiration by way of a more stable and humid boundary layer, which would decrease the specific humidity difference between the surface and the lower atmosphere ($\Delta q$) and increase the aerodynamic resistance ($r_a$) in Eq. (13). Regardless, ET would be less inhibited by these cloudy and humid conditions if the ratio of intercellular to atmospheric $CO_2$, $\frac{c_i}{c_a}$, were given a more realistic treatment that is in line with recent theoretical development and measurements pertaining to optimized stomatal conductance (Medlyn et al., 2011; Prentice et al., 2014; Lin et al., 2015); such treatment is already occuring in some land surface models (e.g., De Kauwe et al., 2015; Kala et al., 2015; Willeit and Ganopolski, 2016). The new theory formalizes how and why, at the leaf level, $\frac{c_i}{c_a}$ increases/decreases with decreased/increased vapor pressure deficit (VPD) (which is approximately proportional to $\Delta q$). Because $CO_2$ uptake is proportional to $1 - \frac{c_i}{c_a}$ (Eq. 6), increased $\frac{c_i}{c_a}$ results in decreased canopy resistance (in order to maintain the same level of $CO_2$ uptake, i.e. match the light-limited rate of GPP), which results in decreased transpiration (Eq. 13) (unless transpiration is occurring at the maximum rate).

Implementation of variable $\frac{c_i}{c_a}$ was not made in SEDGES because the simple relationships between $\frac{c_i}{c_a}$ and VPD that are derived from optimization principles (Medlyn et al., 2011, 2012; Prentice et al., 2014) are incompatible with the SEDGES framework. The derivations for optimized $\frac{c_i}{c_a}$ both assume a Fick's Law of Diffusion transmission between the leaf and the outside air. In order to have consistency in moisture and $CO_2$ fluxes between the land surface and atmosphere, the land surface scheme also needs to use a diffusive exchange between outside air and leaves, or at the least, reasonably approximate such diffusive exchange. Diffusive exchange of moisture and $CO_2$ is not used in the SEDGES framework. Instead, transfer occurs from surface to atmosphere through a bulk aerodynamic formulation. This formulation only approximates the purely diffusive scheme when canopy resistance ($r_c$) greatly exceeds aerodynamic resistance ($r_a$). As is said in section 2.2.4, in early versions of SEDGES (coupled to Planet Simulator), it was found that this condition ($r_c >> r_a$) was frequently violated. That said, in light of the aforementioned ET biases, we intend to incorporate variable $\frac{c_i}{c_a}$ in future versions of SEDGES. Note that it is not enough to simply adjust the globally fixed $\frac{c_i}{c_a}$ in the current model to resolve the ET problems because doing so would improve the bias in the mid-to-high latitudes while worsening it in the equatorial tropics (or vice versa).

Use of a fixed value of $\frac{c_i}{c_a}$ is likely to play a more important role in transpiration (by way of affecting the water use efficiency) than in GPP under non-water-limiting conditions (i.e. when $GPP = GPP_L$, where $GPP_L$ is the light-limited

rate of GPP (section 2.2.3)). This is because, when $r_c >> r_a$ (i.e. using the diffusive approximation for heuristic purposes), transpiration is proportional to $\Delta q\ GPP_L/(1 - \frac{c_i}{c_a})$. As such, changes in $\frac{c_i}{c_a}$ (whose values are typically closer to 1 than 0 for $C_3$ plants) have greater relative impact on transpiration than they do in the equations for either RuBP regeneration-limited or Rubisco-limited photosynthesis (at least for non-extreme conditions) in the Farquhar model (Farquhar et al., 1980).

Furthermore, explicit dependence on $\frac{c_i}{c_a}$ is not critical when modeling GPP. In remote sensing applications, light-use efficiency models without explicit $c_i$ dependence have been widely successful in modeling GPP for many years (e.g., Yuan et al., 2007, and references within). An intermodel comparison between the aforementioned light-use efficiency models and the explicitly $c_i$-dependent enzyme-kinetic-based approaches (Schaefer et al., 2012) shows that including explicit $c_i$ variation (and hence $\frac{c_i}{c_a}$ variation) does not apparently improve the modeling of GPP. On the other hand, for arid regions in which water supply is often

limiting, the decrease in $\frac{c_i}{c_a}$ and concomitant increase in water-use efficiency (GPP per amount of transpired water) affects GPP somewhat strongly.

As the SEDGES model was developed, many of the formulations were calibrated, both independently and dependently of model simulations. As noted by Foley et al. (2013), model calibration should ideally be done separately from model evaluation, but this happens only rarely in practice. In this paper, some of the datasets that were most heavily relied on for calibration were

also used for model evaluation. These include the MODIS albedo dataset (NASA LP DAAC, b), the MTE GPP dataset (Jung et al., 2011) (multi-year annual mean), the (Mueller et al., 2013) ET dataset (multi-year annual mean), and the BNU MODIS-based LAI dataset (Yuan et al., 2011). The least well-constrained parameters and relationships that SEDGES is sensitive to are the maximum light use efficiency parameter ($\epsilon_{max}$), the soil organic carbon value at which soil albedo saturation occurs ($c_7$), the maximum transpiration rate ($trmax$), the $CO_2$ fertilization function, the low temperature limitation function, and $c_4$ and

$c_5$ in the relationship between biomass and snow-covered albedo. The model is only moderately sensitive to the $c_{12}$ parameter, which governs the relationship between biomass and the soil bucket depth. The "representative" value for ($\frac{c_i}{c_a}$) is also somewhat unconstrained (e.g., see Prentice et al., 2014).

Our offline simulations involving SEDGES and the subsequent comparisons with standard datasets should of course not be considered a complete model evaluation. SEDGES undoubtedly has limitations that cannot be seen from the simulations used

in this study. However, from the results of this study and from the formulations that are used in SEDGES, we can anticipate at least some of the situations in which the current model might perform less realistically and/or suggest how to modify SEDGES for these cases.

In section 2.2.2, for the calculation of the light-limited rate of GPP (Eq. 2), the maximum light use efficiency (LUE) parameter ($\epsilon_{max}$) is the same regardless of the diffuse or direct nature of the absorbed solar radiation. However, this lack of distinction

is questionable, as shown by two observation-based studies on the relationship between the diffuse fraction of SW radiation at top-of-canopy and LUE that control for the negative correlation between VPD (vapor pressure deficit) and diffuse SW fraction in their results (Alton et al., 2007; Williams et al., 2014). When going from conditions of predominantly direct solar radiation to predominantly diffuse solar radiation, Alton et al. (2007) finds an observed 6% to 33% increase in LUE in three forests, whereas Williams et al. (2014) finds an $\approx 17\%$ increase in LUE in shrub tundra. The increase in LUE is apparently due to a

more even distribution of photosynthetically active radiation (PAR) among the leaves, which reduces light saturation among the

sunlit leaves. The distinction between sunlit and shade leaves is missing in our model's single big leaf approach to canopy radiation, which tacitly assumes a spatially-averaged light profile at each level of the canopy (de Pury and Farquhar, 1997; Monson and Baldocchi, 2014, p.355). Not including sunlit/shade leaf distinction implies that, in the absence of water limitation, our model underpredicts GPP at low sun angles and under cloudy conditions (and overpredicts it for opposite conditions). This pattern is, in fact, reflected in the regional negative GPP biases that are mentioned in section 5.1, with negative productivity biases occurring in regions with reduced downwelling surface SW radiation for their latitude (in the reanalysis), especially in the equatorial tropics. Recent studies are showing that including sunlit/shade leaf distinction reduces errors in the simulation of GPP (Yuan et al., 2014; Wu et al., 2015) (but c.f. Schaefer et al., 2012). Thus, in a future version of SEDGES, we hope to incorporate the sunlit/shade leaf distinction.

Although gross photosynthesis is somewhat robust against changes in elevation (i.e. atmospheric pressure), there is an appreciable decline at high altitudes and high temperature (Terashima et al., 1995) (according to the widely-used Farquhar model (Farquhar et al., 1980)), which should affect places like the Tibetan Plateau and the South American Altiplano. In these regions, the calculations by Terashima et al. (1995) suggest that SEDGES would overestimate the rate of its light-limited gross primary productivity (Eq. 2) by $\approx 20\%$, and by potentially more for areas in these regions that have low biomass (since these would have a stronger feedback between leaf cover fraction and productivity).

Also in section 2.2.2, we stated that SEDGES approximates net primary productivity as half of gross primary productivity ($NPP/GPP = 0.5$), while noting that this approximation is only valid on the timescales of weeks or longer. On shorter time scales, the ratio may deviate substantially from 0.5, due to the strong short-term dependency of autotrophic respiration on temperature (e.g., Monson and Baldocchi, 2014), and, as such, carbon uptake by the land surface may not be well-simulated on these short time scales.

With respect to simulating terrestrial carbon pool changes, SEDGES's use of a single soil organic carbon reservoir and its fixed residence time for vegetative carbon may also need to be considered when drawing inferences from model results in transient simulations (e.g., see Friend et al., 2014).

These last issues with regards to the carbon cycle are part of a group of model deficiencies in simulating some ecological dynamics. Of these simulation deficiencies, the most severe are probably of the phenological changes (GPP, NPP, transpiration, and especially LAI) associated with green-up in the Northern Hemisphere. More generally, however, it is likely that simplifying assumptions made by SEDGES (namely, constant NPP/GPP as well as its universal temperature limitation function (appendix C)) lead the model to overestimate the capacity of vegetation to adapt to rapidly-changing conditions, especially on daily time scales. As such, SEDGES may underestimate the less positive impacts of extreme weather events on vegetation.

In paleoclimate simulations, periods before the Paleogene may require modification of SEDGES due to differences in atmosphere oxygen concentration and to the effect of evolution on large scale plant characteristics and behavior. SEDGES tacitly assumes an atmospheric oxygen concentration near its present-day value of $\approx 21\%$ in both the maximum light use efficiency parameter ($\epsilon_{max}$) and in the light compensation point in the absence of dark respiration ($CO_{2comp}$). However, $CO_{2comp}$ is proportional to the $O_2$ concentration in the atmosphere (e.g., p.102 of Monson and Baldocchi (2014)), which, in turn, affects the $CO_2$ fertilization effect in SEDGES (Eq. 3). Higher/lower $O_2$ would lead to greater/lesser fertilization. Next, for a given

$CO_2$ concentration, higher $O_2$ increases photorespiration and decreases gross photosynthesis, both experimentally and according to the aforementioned Farquhar model (Beerling et al., 1998) (although the effect was also found to be less pronounced for evolutionary older taxa). These results suggest that $\epsilon_{max}$ should be lowered/raised in SEDGES for higher/lower-than-present atmospheric $O_2$ levels. (In addition, although optimization theory by Prentice et al. (2014) and experimental measurements

show that the ratio of intercellular to atmosphere $CO_2$ $\left(\frac{c_i}{c_a}\right)$ increases/decreases with higher/lower atmospheric $O_2$, the increase is found to be only $\approx 0.06$ when going from the current (21%) to the extreme high (35%) $O_2$ level over the last 400 million years (Beerling et al., 1998). This $\frac{c_i}{c_a}$ difference is small in comparison with other uncertainties and inaccuracies in the model and can thus be probably neglected.)

Apart from the effects of changing atmospheric $O_2$ concentration, vegetation characteristics important to the large scale

have evolved and changed along with the Earth, and this fact has been recognized by previous researchers (e.g., by excluding plant functional types that have not yet evolved in paleoclimate simulations (e.g., Horton et al., 2010; Zhou et al., 2012)). We recommend making changes in line with a recent review of climate-relevant changes in plant physiology on geological time scales (Boyce and Lee, 2017). From the early Paleozoic to the early Devonian, the landscape was dominated by non-vascular vegetation that lacked stomatal control over water loss, substantial roots, and tree-like above-ground structures. In order to

simulate a landscape with this kind of vegetation, canopy resistance ($r_c$) should be set to near 0, the soil water holding capacity ($W_{max}$) should be set to its minimum value ($W_{maxmin}$) (regardless of biomass), and the biomass residence time $\tau_{veg}$ should be decreased as needed to keep the highest biomass from going above $1.0 \; \mathrm{kgC \; m^{-2}}$. Starting with the landscape dominance by vascular plants in the Carboniferous, modern-like landscapes of forests, deep-rooted vegetation, and stomatal control over water loss were widespread, and the current version SEDGES should be able to adequately simulate the vegetation of these

geological times (provided that the aforementioned effects of differing $O_2$ are included). The middle and late Devonian saw the development and expansion of vascular plants and would require a more in-depth analysis on how to properly simulate with SEDGES, because it represents a transition between the two aforementioned time periods.

It is important to realize that this paper has focused on SEDGES forced *offline*, i.e. not fully-integrated into a climate or Earth system model. While a stand-alone evaluation against present-day observations or in comparison with other models is

not sufficient to make firm statements on how SEDGES would perform in a coupled mode, it is beyond the scope of this paper to evaluate SEDGES within a coupled climate or Earth system model. Nevertheless, the performance of SEDGES in a coupled system will need careful evaluation. First results with the coupled PlaSim-SEDGES model are discussed in Paiewonsky (2017). In general, it is to be expected that coupling may exacerbate or reduce biases present in the offline mode of SEDGES that has been evaluated in this paper. For example, the aforementioned ET biases that SEDGES has (section 5.6) are likely

to be diminished through a long-known negative feedback (e.g., Sato et al., 1989): excessive ET moistens and stabilizes the boundary layer which, respectively, reduces the specific humidity difference between the surface and the lower atmosphere (lowers the $\Delta q$ and increases the $r_a$ in Eq. 7), thus reducing ET, and thus feeding back negatively onto the original positive ET bias. On the other hand, there are two notable positive feedbacks that may occur in coupled mode. First, the positive snow-free surface albedo bias in the high Arctic in SEDGES (e.g., see July in Fig. A5) may feed back positively in these regions and

worsen the initial bias by cooling the surface, thus reducing productivity, thus reducing soil organic carbon, and thus increasing

the surface albedo. Second, through another albedo mechanism in boreal spring (the forest-tundra snow albedo feedback (e.g., Bonan et al., 1992; Foley et al., 1994)), the high albedo bias in northern Siberia'a forest-tundra woodland zones would be expected to lead to additional cooling in this region, thus further reducing productivity, thus decreasing biomass, and thus increasing the snow-covered surface albedo, resulting in an exacerbation of the initial positive albedo bias in this region.

For its level of complexity, SEDGES and the land surface framework which it presupposes have the advantages of a flexible time step, canopy control over transpiration, transpiration that is fully coupled to photosynthesis, and vegetative productivity that depends on light and not just temperature and moisture. The short time step option permits SEDGES to be incorporated into models that resolve the diurnal cycle. Because of these strengths, we expect SEDGES to have advantages over vegetation models of similar complexity in simulating vegetation, primary productivity, and transpiration in past geological warm periods

in regions that receive little sunlight (e.g., the Eocene Arctic (Maxbauer et al. (2014) and references within)); and in simulating cloud-vegetation feedbacks in all eras. The SEDGES framework uses photosynthetic-driven stomatal control over evapotranspiration, which is critical when studying hydrological changes on land under altered atmospheric $CO_2$ concentrations, including anthropogenically-induced warming scenarios (Betts et al., 2007).

    The SEDGES framework has the advantages of being easier to understand, process-wise, as compared to more complex

vegetation and hydrological schemes. Increased vegetation model complexity can obfuscate model behavior, and it need not improve performance. In an intermodel comparison of the performance of 24 vegetation models at 39 eddy covariance flux tower sites, Schaefer et al. (2012) finds insignificant effects of nitrogen cycle inclusion and of having a light use efficiency or enzyme kinetic approach to productivity on a model's capacity to simulate GPP. Moreover, even when present-day biome distributions are well-simulated by complex vegetation models, model behaviors can diverge drastically from each other under

non-present conditions due to differing physiological assumptions (e.g., climatic limitations within plant functional types) that are not well-grounded and that may very well change under novel climatic, atmospheric $CO_2$, and nutrient conditions (Fisher et al., 2015).

    In conclusion, we feel that SEDGES provides a new viable and computationally efficient alternative to currently-implemented terrestrial vegetation/ecological models, in particular inside Earth System Models of Intermediate Complexity and for research

on the global scale interactions between the physical climate system and the terrestrial biosphere.

## 7   Code availability

Both the SEDGES model code and the code we used to drive SEDGES with external data are available from the following digital repository:

https://zenodo.org/badge/latestdoi/88959747

## 8   Data availability

ERA-Interim reanalysis data (Dee et al., 2011) available from http://apps.ecmwf.int/datasets/data/interim-full-daily/. The atmospheric $CO_2$ data are available from ftp://aftp.cmdl.noaa.gov/products/trends/co2/co2_annmean_mlo.txt for Mauna Loa record and from http://cdiac.ornl.gov/ftp/trends/co2/lawdome.smoothed.yr20 for the ice core data. We used the

EnsembleGPP_GL.nc file of the MTE GPP dataset (Jung et al., 2011), which is available from https://www.bgc-jena.mpg.de/geodb/projects/Home.php. The CARBONES GPP dataset was obtained from http://www.carbones.eu/wcmqs/. The vegetative carbon dataset (Olson et al., 1985) is available from http://cdiac.ornl.gov/epubs/ndp/ndp017/ndp017_1985.html. The HWSD soil organic carbon dataset HWSD) reference dataset (Wieder et al., 2011) is available from https://daac.ornl.gov/SOILS/guides/HWSD.html. The LAI data (Yuan et al., 2011) is

available from http://globalchange.bnu.edu.cn/research/lai/. The MODIS white sky albedo data (NASA LP DAAC, b) and MODIS snow-free albedo data (NASA LP DAAC, a) are available from https://reverb.echo.nasa.gov/reverb/. The reference ET dataset (Mueller et al., 2013) can, upon registration, be download from http://www.iac.ethz.ch/group/land-climate-dynamics/research/landflux-eval.html. The UNH/GRDC runoff dataset (Fekete et al., 2002) can be obtained from

http://www.grdc.sr.unh.edu/html/Data/index.html. The ISLSCP II tree cover dataset (DeFries and Hansen, 2009) is available from https://daac.ornl.gov/cgi-bin/dsviewer.pl?ds_id=931. The rooting zone plant available soil water storage capacity dataset (Kleidon and Heimann, 1998; Hall et al., 2006; Kleidon, 2011) is available from the following URL: https://daac.ornl.gov/cgi-bin/dsviewer.pl?ds_id=1006. Accessing these last two datasets requires registering and signing in.

## Appendix A:  Notes on the Temperature Limitation Function

In this section, we discuss the rationale behind our particular temperature limitation multiplier, $f_2(T_{sfc})$, on the light-limited rate of gross primary productivity (Eq. 2) in section 2.2.3.

$f_2(T_{sfc})$ increases linearly from 0 at $T_{sfc}$ of 0 °C to 1 at $T_{sfc}$ of 20 °C and then plateaus at 1. The linear increase is chosen for its simplicity and because it is analogous to the often-used "growing degree days" (GDD) metric (with 0 °C base) in agriculture and ecology (Kauppi and Posch, 1985; Prentice et al., 1992; Kaplan, 2001) (save for the shorter time scale).

SEDGES' critical temperature of 20 °C at which productivity is no longer limited lies close to the middle of the range of optimum temperatures for C3 plants (5 °C to 39 °C) that was found in a survey of 212 temperature response curves of photosynthesis in the literature for different species and growth temperatures (Way and Yamori, 2014; Yamori et al., 2014). While the critical temperature for the Kleidon (2006b) SimBA model is 5 °C, it otherwise uses the same ramp function formulation as SEDGES.

Temperature limitation on photosynthesis has been well studied (e.g., see reviews by Sage and Kubien, 2007; Yamori et al., 2014), but many uncertainties remain. In land plants, photosynthesis is significantly limited by both high and low temperatures. Individual plants of different species and from different environments have differing optimal temperatures for photosynthesis, such that away from the optimal temperature, photosynthetic rate decreases (Yamori et al., 2014). Around the temperature

optimum, there exists a range of temperatures for which photosynthesis is nearly as high as at the optimum (Berry and Bjork-man, 1980; Yamori et al., 2014). Both the temperature dependency and the maximum rate of photosynthesis (realized at the temperature optimum) can shift as a plant acclimates to a different set of environmental conditions (Berry and Bjorkman, 1980; Way and Yamori, 2014; Yamori et al., 2014).

In most plants, temperature limitation on photosynthesis depends on the species, current light levels, the environmental conditions under which the plant grew, and the internal levels of $CO_2$ in the leaves (Berry and Bjorkman, 1980; Sage and Kubien, 2007; Way and Yamori, 2014; Yamori et al., 2014). In addition, the current temperature limitation depends on the extent of acclimation to recent environmental conditions (e.g., Dietze, 2014)

The framework in which we use $f_2(T_{sfc})$ assumes perfect acclimation and adaptation of the vegetation to the current weather
and light conditions. As such, the SEDGES temperature limitation assumes the presence of the most productive set of plants that could ideally grow under those conditions and also allows for their instantaneous adaptation and acclimation to those conditions. In other words, for a given surface temperature ($T_{sfc}$), $f_2(T_{sfc})$ represents the universal physiological constraints on productivity exerted by that (isolated) given temperature.

For high temperatures, a conscious decision was made to not include a decline in productivity in SEDGES. Although it is
well-established that high temperatures (i.e. temperatures above the optimum) limit productivity for individual plants, some of that limitation that has been found empirically may have been due to an indirect effect of temperature on increasing leaf-to-air vapor pressure deficit (for constant ambient relative humidity), which causes optimum temperature to be lower than when vapor pressure deficit is held fixed (Lin et al., 2012). An even more important point is that high temperature limitation reflects more the inability of plants that are adapted to a given environment to perform equally well under all conditions that they might face
in that environment and less an intrinsic barrier for plant life at that location to adapt to a given high temperature regime (via some combination of species migrations, natural selection, evolution, and/or acclimation). For example, although the Yamori et al. (2014) study found C3 plants to generally have temperature optima between 10 °C and 35 °C, light-saturated and $CO_2$-saturated photosynthesis for the C3 hot desert plant, *Rhazya stricta* shows no significant decline with leaf temperature even up to 44 °C (Lawson et al., 2014). Moreover, its (ordinary) daily photosynthetic rates are nearly as high as those of common C3
agricultural crops, in spite of living under natural conditions of much higher vapor pressure deficit and presumably much drier soil (Lawson et al., 2014), which reduce productivity. Similar results as for *Rhazya stricta* have been found for the C3 desert shrub, *Larrea divaricata* (Mooney et al., 1978). Thus, the large scale ability of vascular plant life to adapt to high temperatures through the increased prevalence of heat-tolerant species and phenotypes should be properly accounted for when modeling vegetation and the carbon cycle in past warm climates (such as the Cretaceous) as well as in future climates.

**Appendix B: Numerical parameterization of canopy resistance and maximum transpiration rate**

Canopy resistance is the greater of an unconstrained canopy resistance ($r_{cu}$), which is determined by the light-limited rate of canopy photosynthesis ($GPP_L$), and a canopy resistance set by the maximum rate at which the rooting zone can supply water for transpiration. The equation for the unconstrained canopy resistance is essentially derived by incorporating the canopy

resistance ($r_c$) value from the previous time step, using the mismatch between $GPP_W$ and $GPP_L$ in the last time step, and accounting for updates to the variables that affect $GPP_W$ and $GPP_L$. Although this formulation of $r_{cu}$ depends on values from the previous time step, this dependency is arbitrary. An expression for $r_{cu}$ that depends only on values in the current time step could be derived and used. (Such a $r_{cu}$ would depend on all the variables that comprise $GPP_W$ and $GPP_L$, and these variables would be evaluated at the current time step.)

Our starting equation for deriving the unconstrained canopy resistance uses the formulations of $GPP_L$ and $GPP_W$ given in Eqs. (2), (4), (5), and (6) and is as follows:

$$
\begin{aligned}
&\equiv GPP_{Lt+1} - GPP_{Wt+1} \\
&\approx GPP_L \frac{f_{leaf}}{f_{leaf_{t-1}}} \frac{f_2(T_{sfc})}{f_2(T_{sfc_{t-1}})} \frac{SW\downarrow}{SW\downarrow_{t-1}} - GPP_W \frac{f_{leaf}}{f_{leaf_{t-1}}} \frac{1.6 r_{ct-1} + r_{at-1}}{1.6 r_{cu} + r_a}.
\end{aligned}
\tag{B1}
$$

Here, the equivalence statement reflects the first stomatal goal of parsimoniously meeting the light-driven demand for $CO_2$. The future values of $GPP_W$ and $GPP_L$ are approximated (for simplicity) by neglecting changes in slowly varying variables (e.g., $\rho$). Equation (B1) represents the standard case for deriving $r_c$. Special cases are discussed below (e.g., when $f_{leaf_{t-1}}$, $f_2(T_{sfc_{t-1}})$, or $SW\downarrow_{t-1}$ are zero.) Solving Eq. (B1) for $r_{cu}$ yields

$$
r_{cu} = \left( \frac{GPP_W}{GPP_L} \frac{f_2(T_{sfc_{t-1}})}{f_2(T_{sfc})} \frac{SW\downarrow_{t-1}}{SW\downarrow} (1.6 r_{ct-1} + r_{at-1}) - r_a \right) \bigg/ 1.6.
\tag{B2}
$$

Equation (B2) has provided us with $r_{cu}$, the canopy resistance that would occur in the complete absence of physiological limitations on plant water loss. In the real world, plants must restrict how much their stomata open when water cannot be extracted from the soil (or from internal plant storage) and transported to the leaf stomata fast enough to balance the loss to the atmosphere through the stomata. Doing so helps to keep water potentials within the plant from falling to levels that would cause "runaway" cavitation (Tyree and Sperry, 1988) or "hydraulic failure" (McDowell et al., 2008). Closing stomata increases stomatal resistances across the leaves, which, at the canopy scale, increases the canopy resistance. In this way, canopy resistance can be constrained by the supply rate of water from the soil to the leaves.

To simulate the maximum supply rate of water for transpiration, SEDGES adapts the simple model of Federer (1982). The original Federer maximum supply rate is directly proportional to the soil wetness fraction multiplied by a fixed constant parameter. This parameter is the *absolute* maximum transpiration value, which we denote here by "$trmax$". In SEDGES, we extend the original Federer formulation from the limited case of a fully-vegetated surface to the case of a mixed surface comprised of green leaves and exposed soil. This is achieved by simply multiplying the original expression by the leaf cover fraction, $f_{leaf}$. Thus, the supply rate for transpiration, $S_{tr}$, is as follows in SEDGES:

$$
S_{tr} = f_{leaf} \cdot \beta_{tr} \cdot trmax,
\tag{B3}
$$

where

$\beta_{tr}$ is a water stress factor that affects the roots' ability to supply water for transpiration and equals the soil wetness fraction, $W_{frac}$, and

$trmax$ has a value of $2.78 \times 10^{-7}$ m s$^{-1}$, which is taken from the BETHY vegetation model (Knorr, 2000) and is unfortunately not well-constrained (Knorr, 2000; Knauer et al., 2015). This maximum transpiration value is achieved only for a fully leaved

($f_{leaf} = 1$), fully wet soil.

The above extension of the original Federer formulation for water supply rate requires some justification. A reader that is familiar with the LPJ model might notice that SEDGES uses a similar formulation as LPJ (Gerten et al., 2004); that is, both models multiply the original simple Federer expression ($S_{tr} = \beta_{tr} \cdot trmax$) by a fractional term on the right hand side: $f_{veg}$ for LPJ and $f_{leaf}$ for SEDGES. However, because the definitions of $f_{veg}$ in LPJ and $f_{leaf}$ in SEDGES differ, it is

inappropriate to simply borrow the LPJ formulation and apply it in SEDGES. In LPJ, $f_{veg}$ is the vegetative cover fraction, which is (approximately) the portion of a grid cell in which vegetation resides. Although this portion appears to change some with daily phenology (Gerten et al., 2004), the basic idea in LPJ seems to be that the vegetated portion of the grid cell is (essentially) completely vegetated, i.e. having roots, stems, and leaves. As such, it makes sense in LPJ to multiply the original Federer supply rate formulation by $f_{veg}$ because this is the fraction of the grid cell that functions as in the Federer model. In

contrast, SEDGES assumes that vegetation resides in the entire grid cell. In other words, SEDGES assumes that vegetation is spread throughout the grid cell uniformly, which means that plant roots and above-ground parts are also distributed uniformly at the large scale. Thus, the vegetative fraction in SEDGES, $f_{veg}$, is assumed to be 1. In contrast, in SEDGES, $f_{leaf}$ is created by the covering of bare soil by green leaves when looking down from above the canopy. As such, it is not obvious why the simple supply rate formulation in Federer (1982) is multiplied by $f_{leaf}$ in SEDGES. The answer lies in a close examination of

the $trmax$ parameter of the original model.

The aforementioned simple supply rate model of Federer (1982) is tested, evaluated, and calibrated in the same paper against a more sophisticated "Type I" (Guswa et al., 2002) water uptake model that is forced using site-specific atmospheric observational data. Doing so reveals that the maximum transpiration rate (i.e. the $trmax$ parameter) in the simple model depends on the following input parameters for the Type I model (in decreasing order of strength): rooting density, root internal

resistance, depth of the rooting zone, and vegetation height/surface roughness. Although the first three of these dependencies are found to be substantial, those dependencies are not included in the simple model, which takes the $trmax$ parameter to be a constant. In addition, even the more sophisticated Type I water uptake model neglects soil drying-induced embolism's increase of root xylem resistance (e.g., Linton et al., 1998; Domec et al., 2004), which some authors (Domec et al., 2009; Javaux et al., 2013) feel plays a very significant or even dominant role in the whole plant conductivity of water. Significant

increase in rooting zone xylem resistance decreases the maximum transpiration rate within the framework of the Type I model given in Federer (1982).

The missing dependencies in the original simple Federer formulation for maximum supply rate of transpiration presented in the last paragraph are addressed (though somewhat crudely) in SEDGES by multiplying the original formulation by the leaf cover fraction, $f_{leaf}$. Here, multiplication by $f_{leaf}$ becomes a proxy for the effect of decreasing root embolism with increasing

soil wetness fraction in dry soils because $f_{leaf}$ increases with soil wetness fraction for very low values of soil wetness fraction

(see Eq. 21). At low biomass values ($\leq 0.25$ kgC m$^{-2}$), $f_{leaf}$ is a proxy for rooting density because $f_{leaf}$ scales with biomass (through Eqs. 5 and 18) and so does rooting density (implicitly). At higher biomass values ($> 0.25$ kgC m$^{-2}$), $f_{leaf}$ serves as a proxy for rooting depth since both scale with biomass.

The water supply rate in the preceding discussion, as noted above, directly impacts the minimum canopy resistance, for which we will now derive an expression. Starting from Eqs. (7) and (13), it follows that

$$\frac{T}{ET} = \frac{f_{leaf}}{(1 + r_c g_a) C_w}. \tag{B4}$$

Starting from Eq. (B4), we define $r_c min$ as the value of canopy resistance that matches the supply rate of transpiration in the current time step. That is, we have

$$f_{leaf} \cdot \beta_{tr} \cdot trmax \equiv \frac{f_{leaf}}{1 + r_c min \ g_a} \frac{ET}{C_w}, \tag{B5}$$

where we note that $\frac{ET}{C_w}$ is the potential evapotranspiration ($PET$) and $g_a = \frac{1}{r_a}$. Solving Eq. (B5) for $r_c min$ yields

$$r_c min = \left( \frac{PET}{\beta_{tr} \cdot trmax} - 1 \right) r_a. \tag{B6}$$

The final canopy resistance is restricted as follows:

$$r_c = min(r_c max, max(r_c min, r_{cu}, r_c min_{min})), \tag{B7}$$

where

$r_c min_{min}$ is the absolute minimum possible value for $r_c$ and is set to 0 for simplicity, and

$r_c max$ is the maximum possible value for $r_c$ and is set to an extremely large value, purely for the sake of model simplicity and elegance.

In deriving above Eqs. (B2), (B5), and (B6), we have excluded all cases in which we would have had to divide by variables with a zero value. These cases are handled as follows: Firstly, SEDGES checks to see if the light limited rate of $GPP$ is zero in the next time step or if the soil is completely dry. If so, then canopy resistance is set to the maximum possible value. In the former case, there is no carbon benefit to keeping stomata open, and in the latter case, the supply rate is zero. If, however, the first check comes back negative for both conditions, then SEDGES checks for the case of parched soil getting rewetted. If this is the case, canopy resistance is set to the minimum value, $r_c min$. If, however, the second check also comes back negative, then a final check is made to see if the light-limited rate of $GPP$ is zero in the current time step. Under this last scenario, either the sun has risen or the surface temperature has risen above freezing in between the last and current time steps. Here, an expression for $r_{cu}$ can be derived by substituting into Eq. (B1) a time-specifying version of Eq. (2): $GPP_{Lt+1} = \epsilon_{max} \cdot f_1(CO_2) \cdot f_2(T_{sfc}) \cdot f_{leaf} \cdot SW\downarrow$. We then solve for the unconstrained canopy resistance and get the following:

$$r_{cu}* = \left( \frac{GPP_W}{\epsilon_{max} \cdot f_1(CO_2) \cdot f_2(T_{sfc}) \cdot f_{leaf_{t-1}} \cdot SW\downarrow} (1.6 r_{ct-1} + r_{at-1}) - r_a \right) \Big/ 1.6. \tag{B8}$$

where the "*" is used to denote our special case of $GPP_{Lt+1} > 0$, $GPP_L = 0$, $W_{soil} > 0$, and $f_{leaf_{t-1}} > 0$. After its computation in Eq. (B8), $r_{cu}*$ is restricted the same as $r_{cu}$ is in Eq. (B7). In the case that the first, second, and final checks are all negative, then we have the standard scenario described by Eqs. (B2), (B5), and (B6).

## Appendix C:  Output of the original SimBA

As a supplementary experiment, we followed the same procedure as we did in forcing SEDGES (section 4) in running the original SimBA model (Kleidon, 2006b) (that found in version 15 of Planet Simulator) offline, except that the only equilibrium simulations used pre-industrial $CO_2$ and were run for 900 years without carbon cycle acceleration (because it was not needed). The results for GPP and ET are shown in figures A1 and A2. Excessively high ET in figure A2 results from the Manabe-style parameterization for evaporation from the land surface (Manabe, 1969), which is at the potential rate as soil wetness is reduced,

until a given threshold soil wetness fraction (0.25 for SimBA) is reached, after which the surface wetness factor, $C_w$ (Eq. 7) is reduced linearly with the soil wetness fraction until they both reach 0. This scheme for land surface evaporation was used with the first generation land surface models (e.g., Pitman, 2003) and is widely known to yield excessive evapotranspiration from the land surface (e.g., see the intermodel comparisons in Chen et al., 1997b; Desborough, 1999) and extremely dry soils (Chen et al., 1997b). Such extremely dry soils are also found in the offline-driven SimBA. This, in tandem with a higher soil

wetness fraction threshold for commencement of leaf fall (0.25 versus 0.05 in SEDGES), causes GPP to be unrealistically low throughout the world (Fig. A1).

In line with the findings by Dekker et al. (2010), in this offline version of SimBA, we found multiple steady states when the model was run using pre-industrial levels of $CO_2$: equilibrium biomass was slightly higher in some regions when the model was initialized with tropical forest-like biomass of 12 $kgC\,m^{-2}$ as opposed to a bare desert initialization. Only the

forest-initialized run was subsequently continued up through 2010.

[Figure 1 about here.]

[Figure 2 about here.]

*Competing interests.* The authors declare that they have no conflict of interest.

*Acknowledgements.* Climate Data Operators (CDO) was obtained from http://www.mpimet.mpg.de/cdo and was used to process much of the data. Version 6.2.1 of NCL (NCAR Command Language) was used to generate the plots (https://doi.org/10.5065/D6WD3XH5). Pablo Paiewonsky thanks Daniela Dalmonech for alerting us to severe deficiencies in the simulated evapotranspiration in an early version of
5    SEDGES that ultimately led us to revise and much improve the model. Pablo Paiewonsky's work on the model's development was financially supported, in part, by the NSF-funded Alliance for Graduate Education and Professoriate (AGEP) program and by a Carson Carr Graduate Diversity Fellowship from the Office of Diversity, Equity, and Inclusion of the State University of New York.

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

[Figure 3 about here.]

[Figure 4 about here.]

[Figure 5 about here.]

[Figure 6 about here.]

5 [Figure 7 about here.]

[Figure 8 about here.]

[Figure 9 about here.]

[Figure 10 about here.]

[Figure 11 about here.]

10 [Figure 12 about here.]

[Figure 13 about here.]

[Figure 14 about here.]

[Figure 15 about here.]

[Figure 16 about here.]

15 [Figure 17 about here.]

[Figure 18 about here.]

[Figure 19 about here.]

[Figure 20 about here.]

[Figure 21 about here.]

## List of Figures

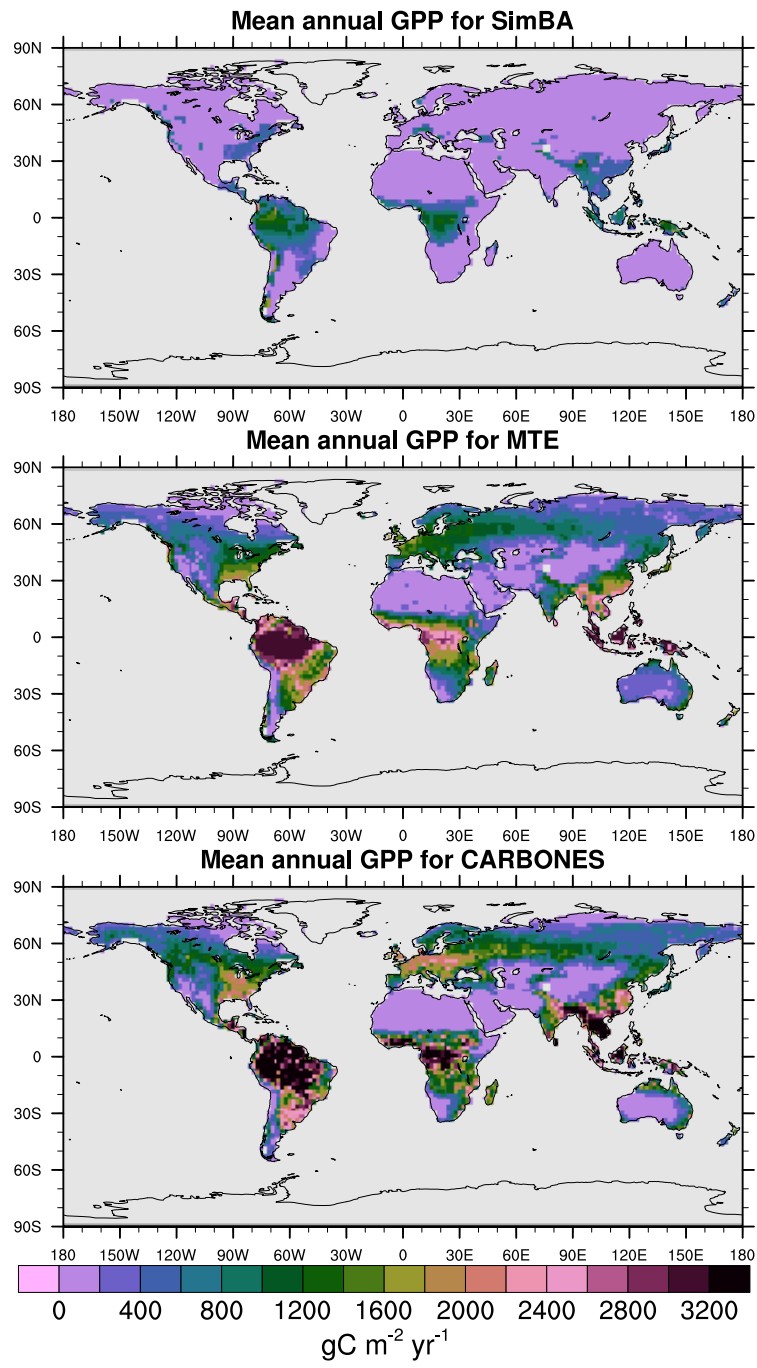

**Figure A1.** Multi-year annual mean of gross primary productivity (GPP) for the original SimBA model (Kleidon, 2006b) and the two reference datasets, MTE and CARBONES, for 1990-2009 over non-glaciated land.

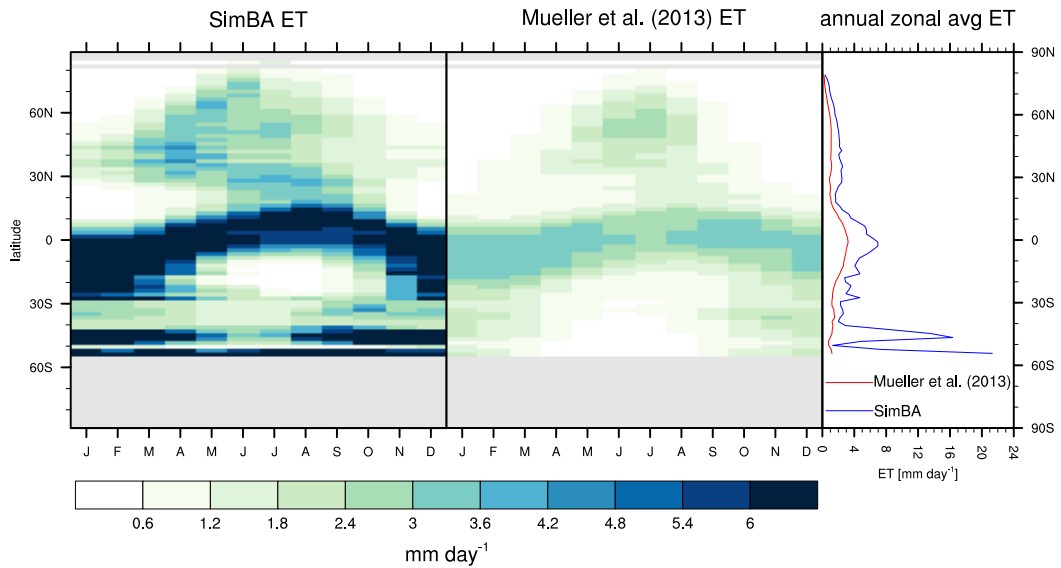

**Figure A2.** Zonal multi-year monthly means and zonal multi-year annual means of evapotranspiration (ET) for the original SimBA model (Kleidon, 2006b) and the Mueller et al. (2013) reference dataset for the 1989-2005 period over non-glaciated land.

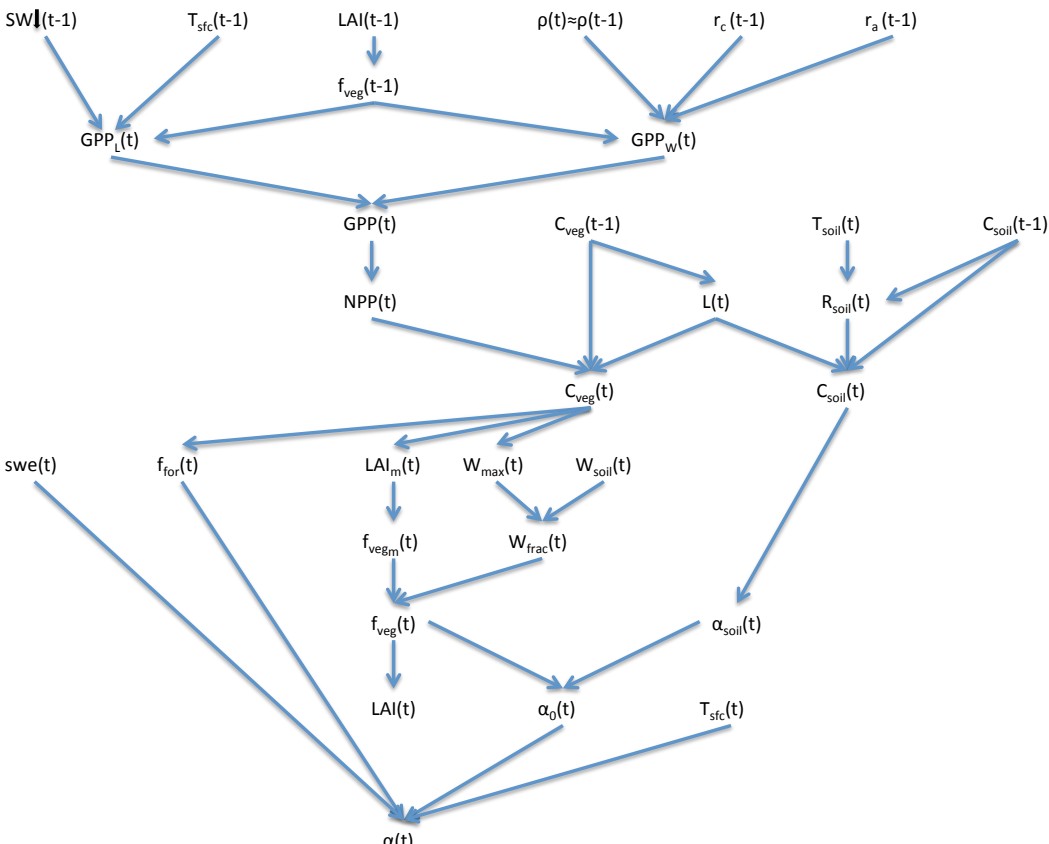

**Figure A3.** Variable dependencies and updating in SEDGES

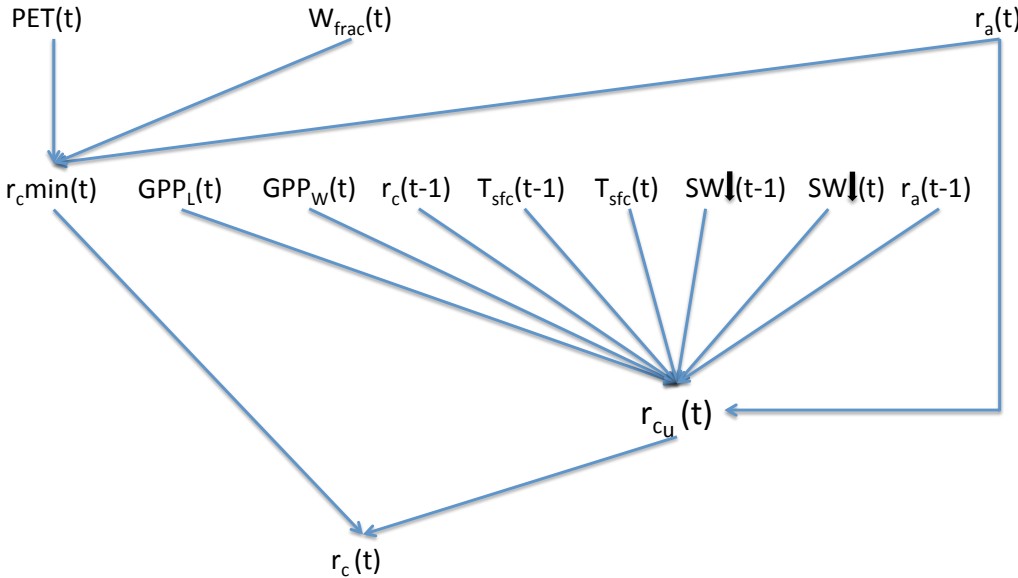

**Figure A4.** Canopy resistance dependencies and updating in SEDGES

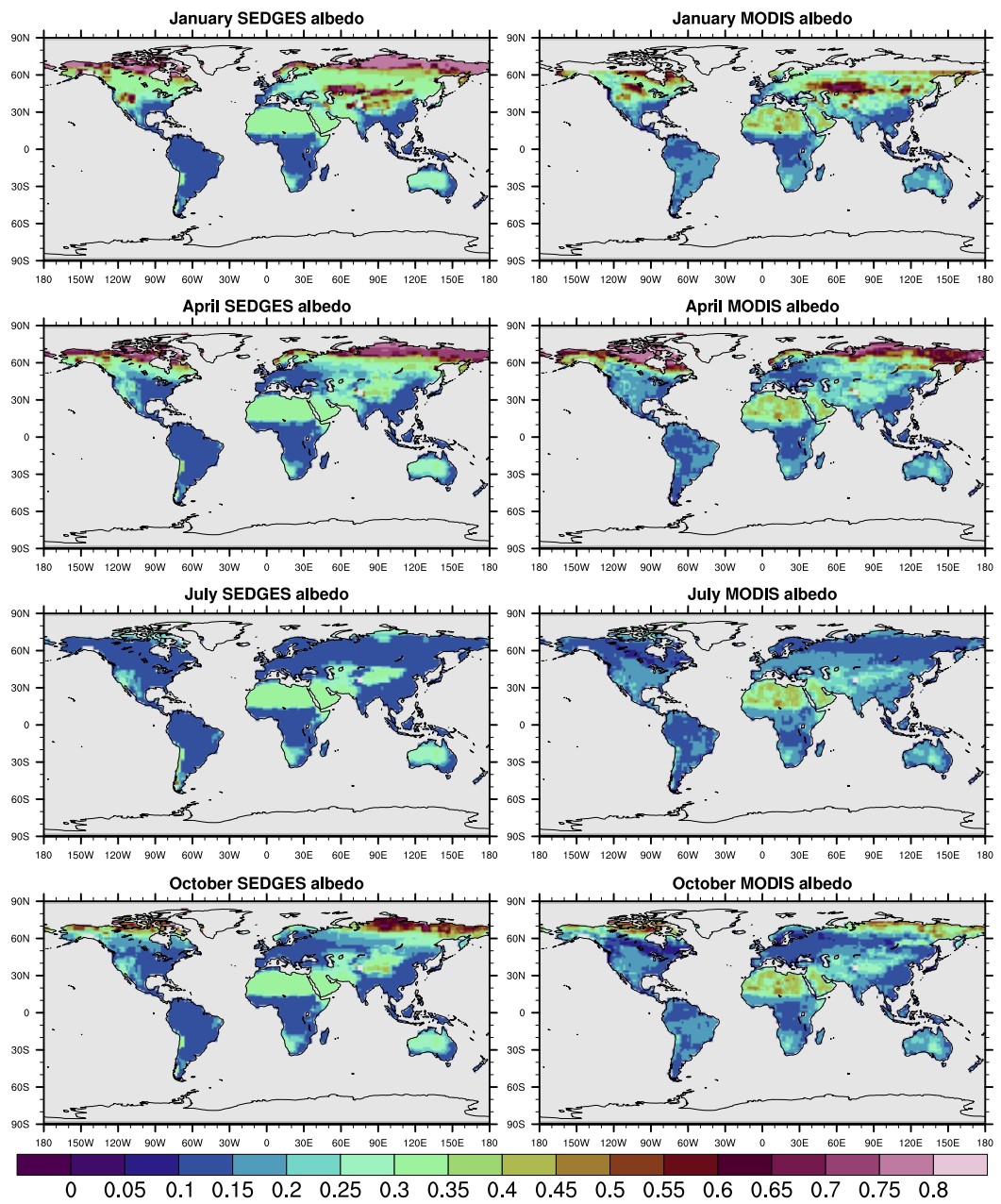

**Figure A5.** Left: SEDGES monthly mean climatologies of surface albedo for 2001-2010. Right: MODIS monthly mean climatologies of white sky albedo for 2001-2010 (NASA LP DAAC, b).

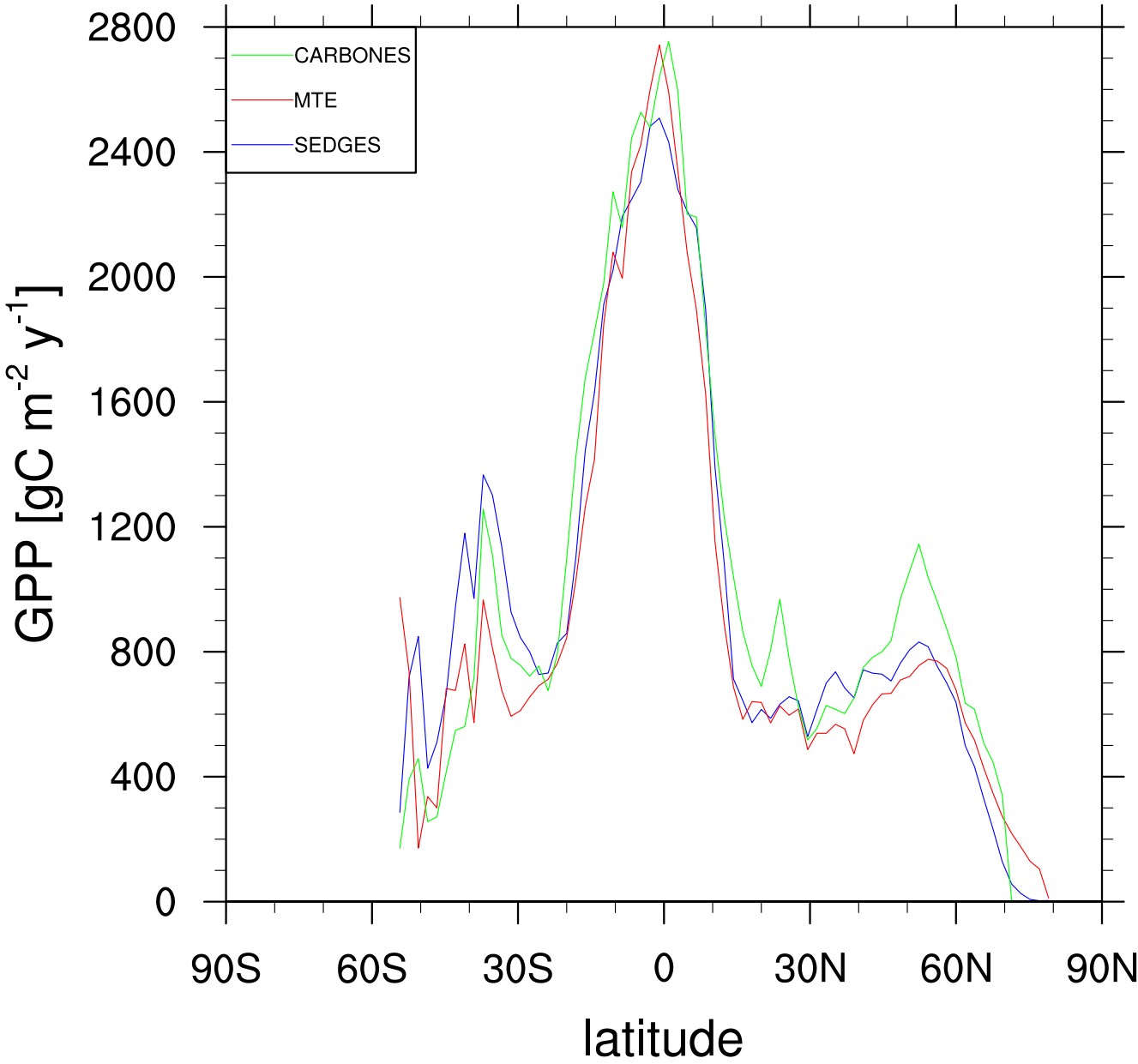

**Figure A6.** Zonal multi-year annual mean of gross primary productivity (GPP) for SEDGES and the two reference datasets, MTE and CARBONES, for 1990-2009 over non-glaciated land.

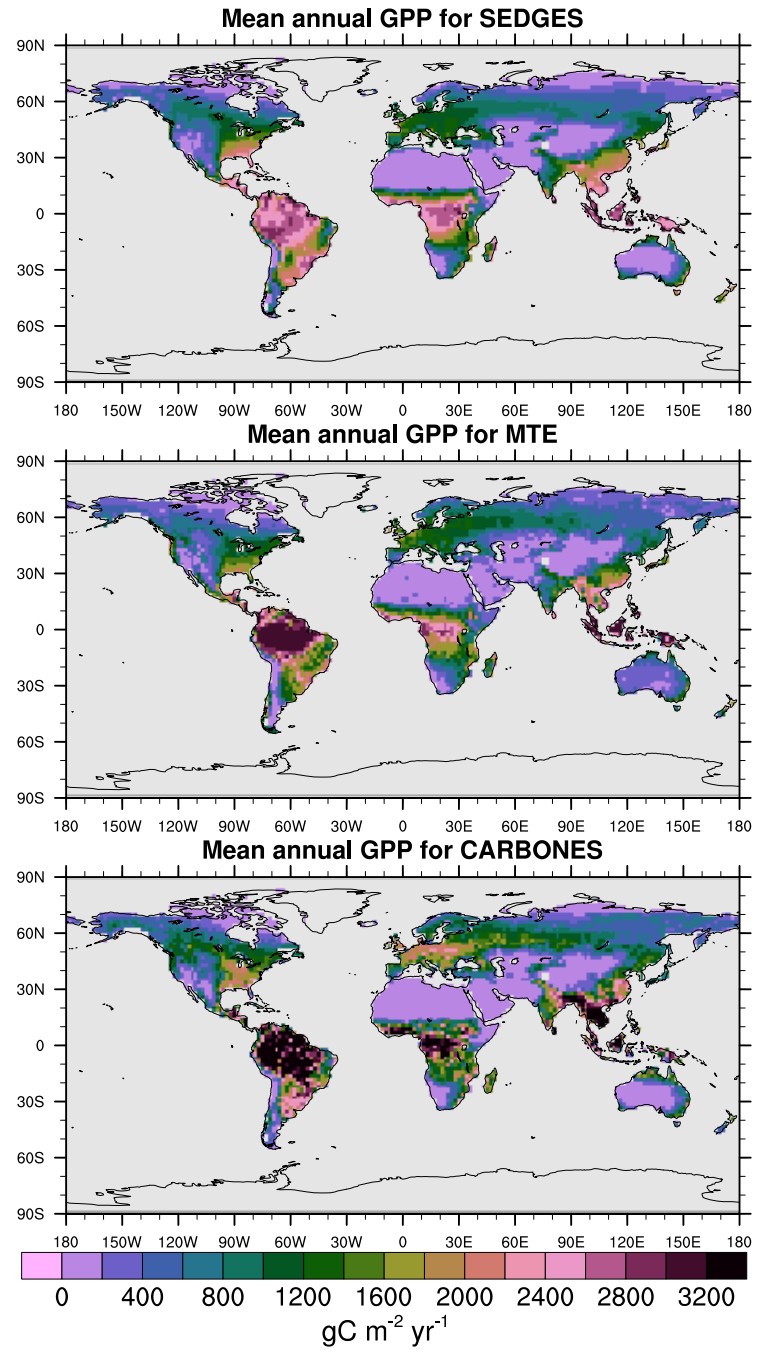

**Figure A7.** Multi-year annual mean of gross primary productivity (GPP) for SEDGES and the two reference datasets, MTE and CARBONES, for 1990-2009 over non-glaciated land.

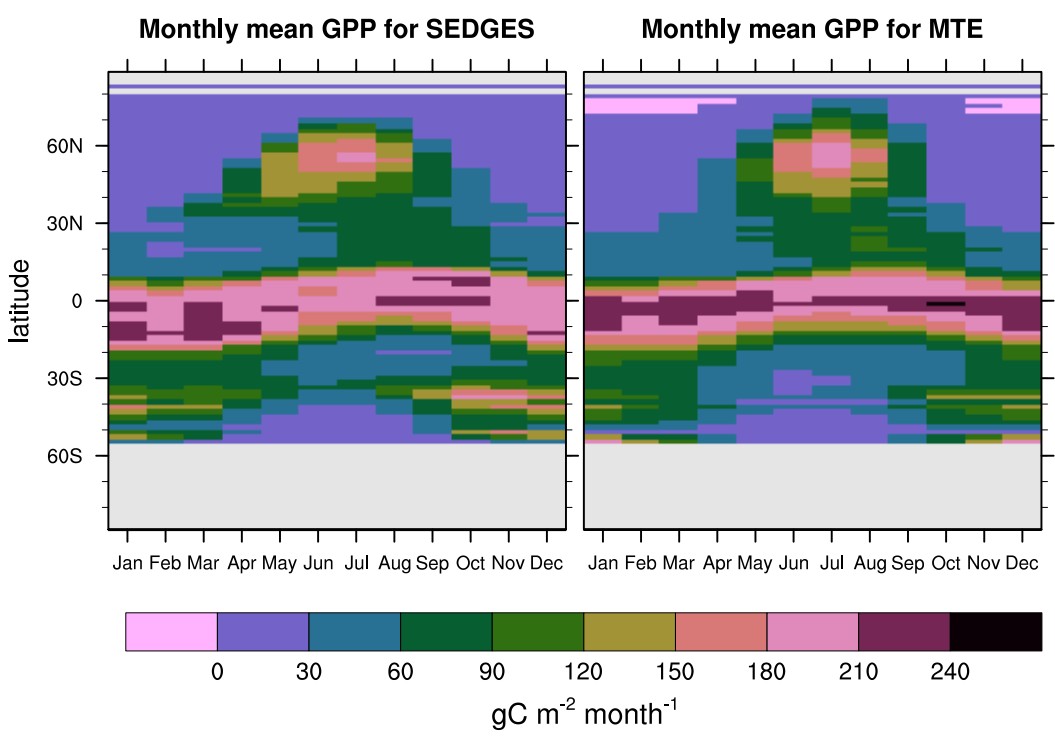

**Figure A8.** Zonal multi-year monthly means of gross primary productivity (GPP) for SEDGES and the MTE reference dataset for 1982-2010 over non-glaciated land.

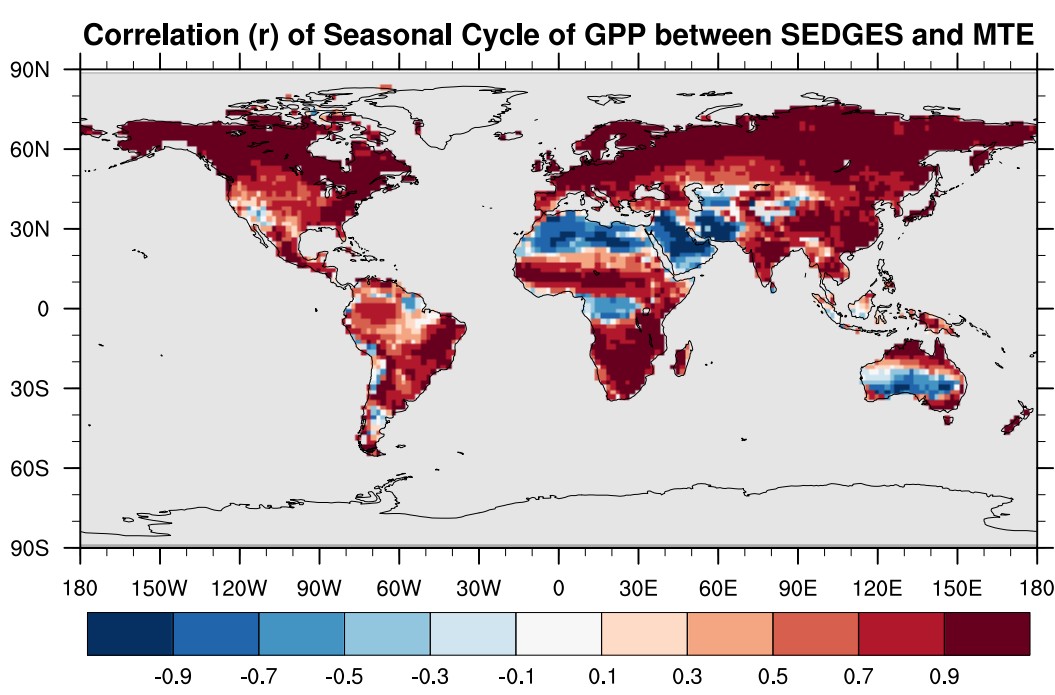

**Figure A9.** Pearson correlation coefficients of the seasonal cycle (i.e. of the multi-year monthly means) between SEDGES and the MTE reference dataset for the 1986-2005 period.

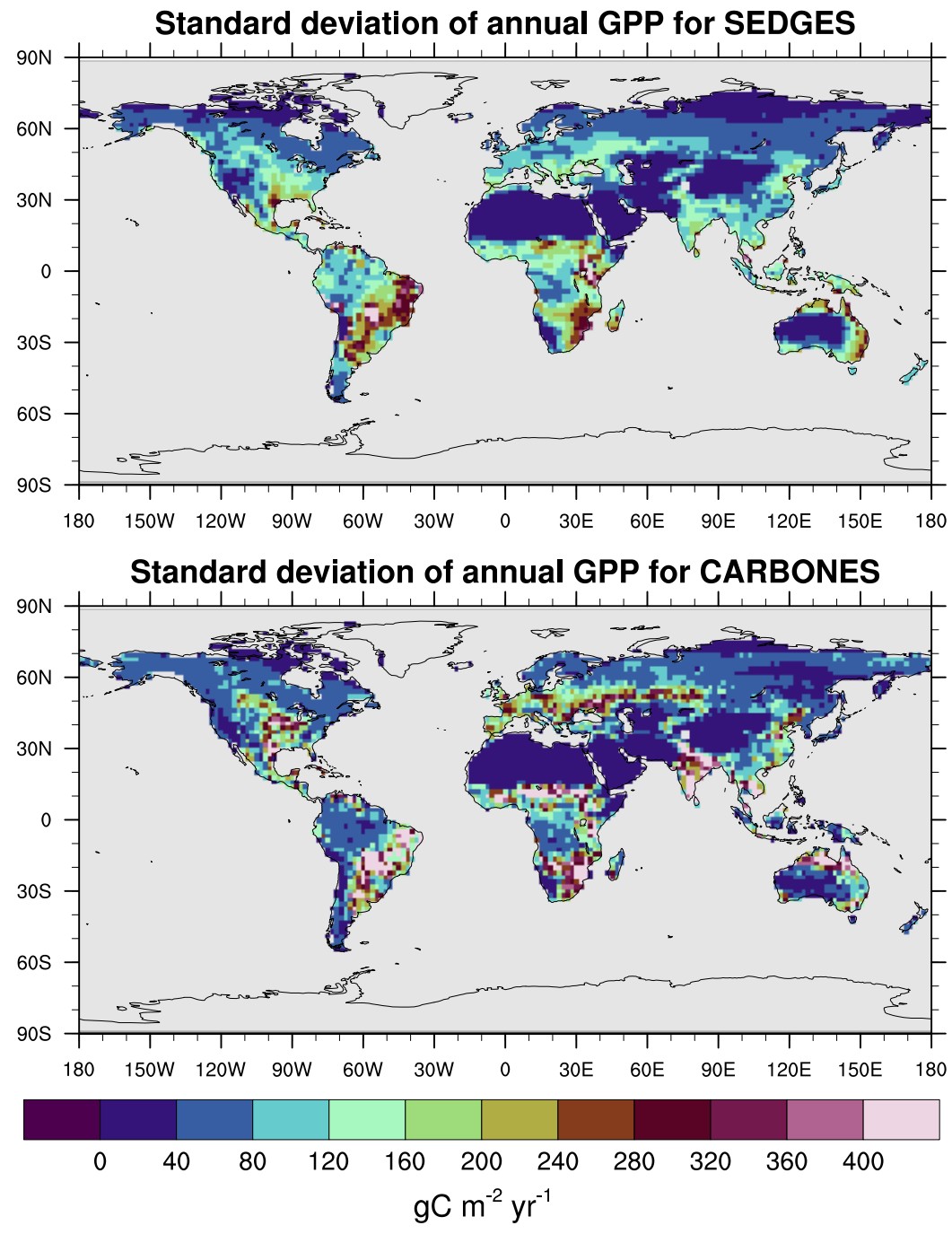

**Figure A10.** Interannual variability of GPP for SEDGES and the CARBONES reference dataset for the 1990-2009 period.

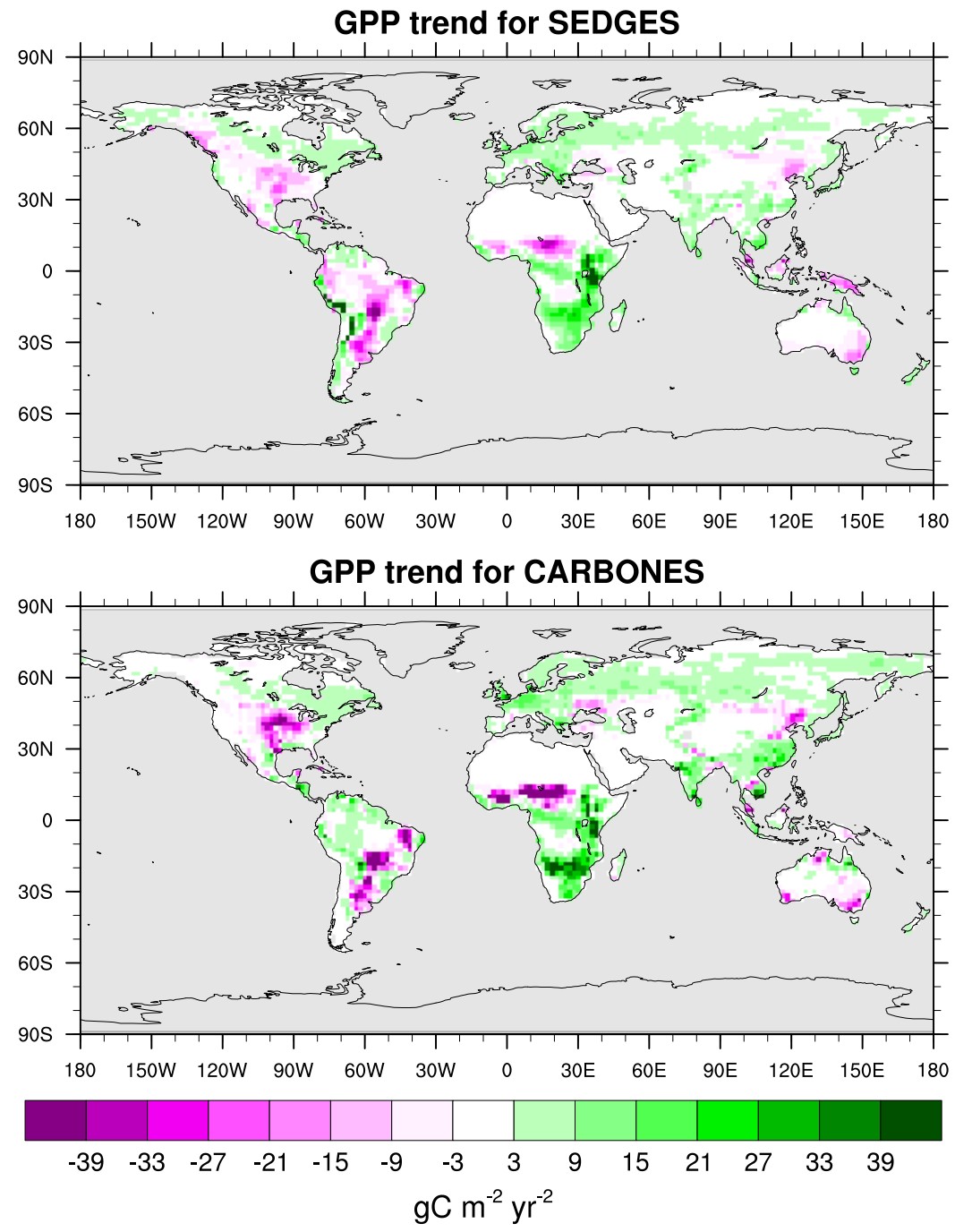

**Figure A11.** Linear regression trend in annual GPP for SEDGES and the CARBONES reference dataset for 1990-2009.

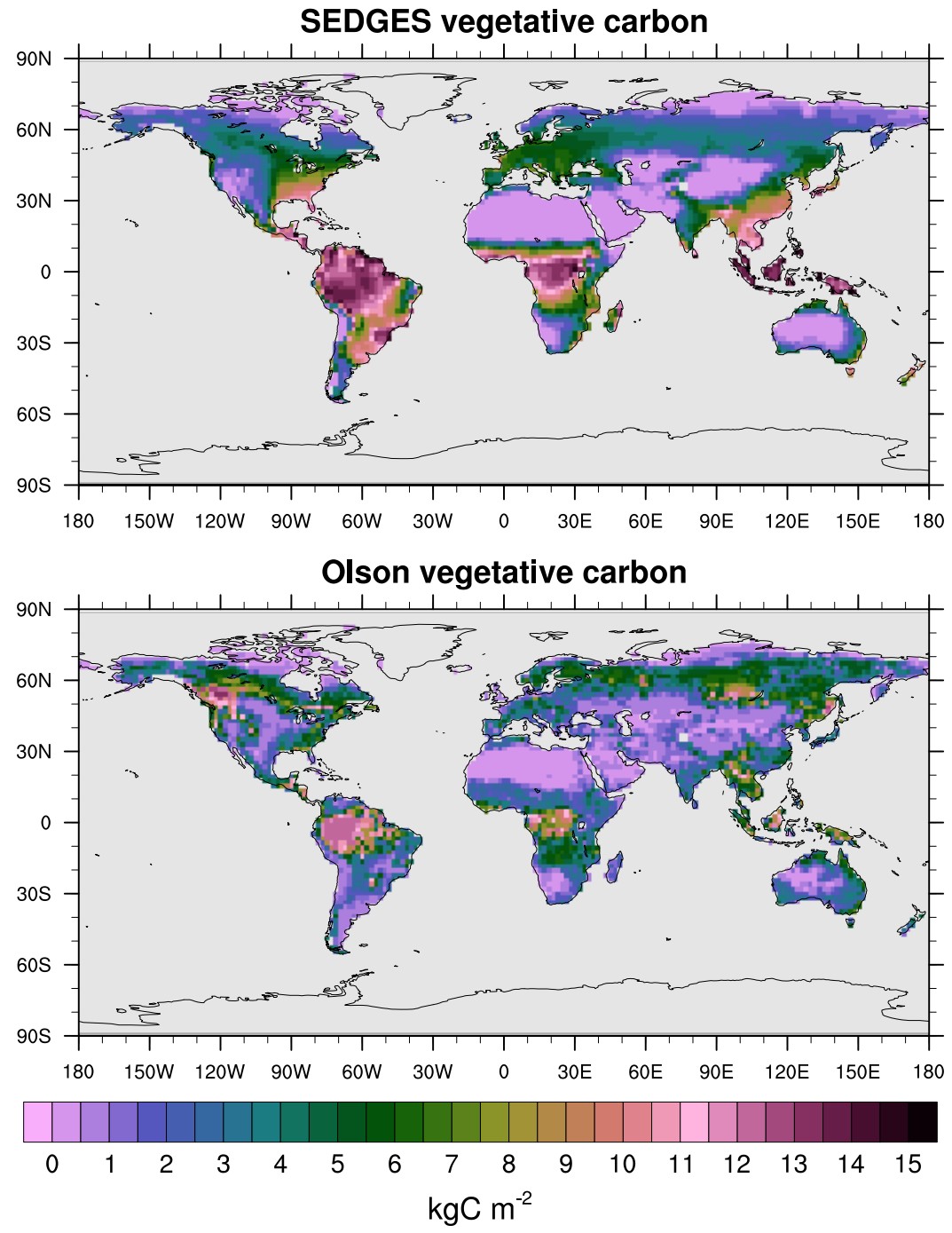

**Figure A12.** Vegetative carbon for SEDGES: mean over 1981-2010 in the transient $CO_2$ simulation; vegetative carbon from the Olson et al. (1985) reference dataset, which represents pre-Iron Age vegetation save for the most extreme anthropogenic land cover changes.

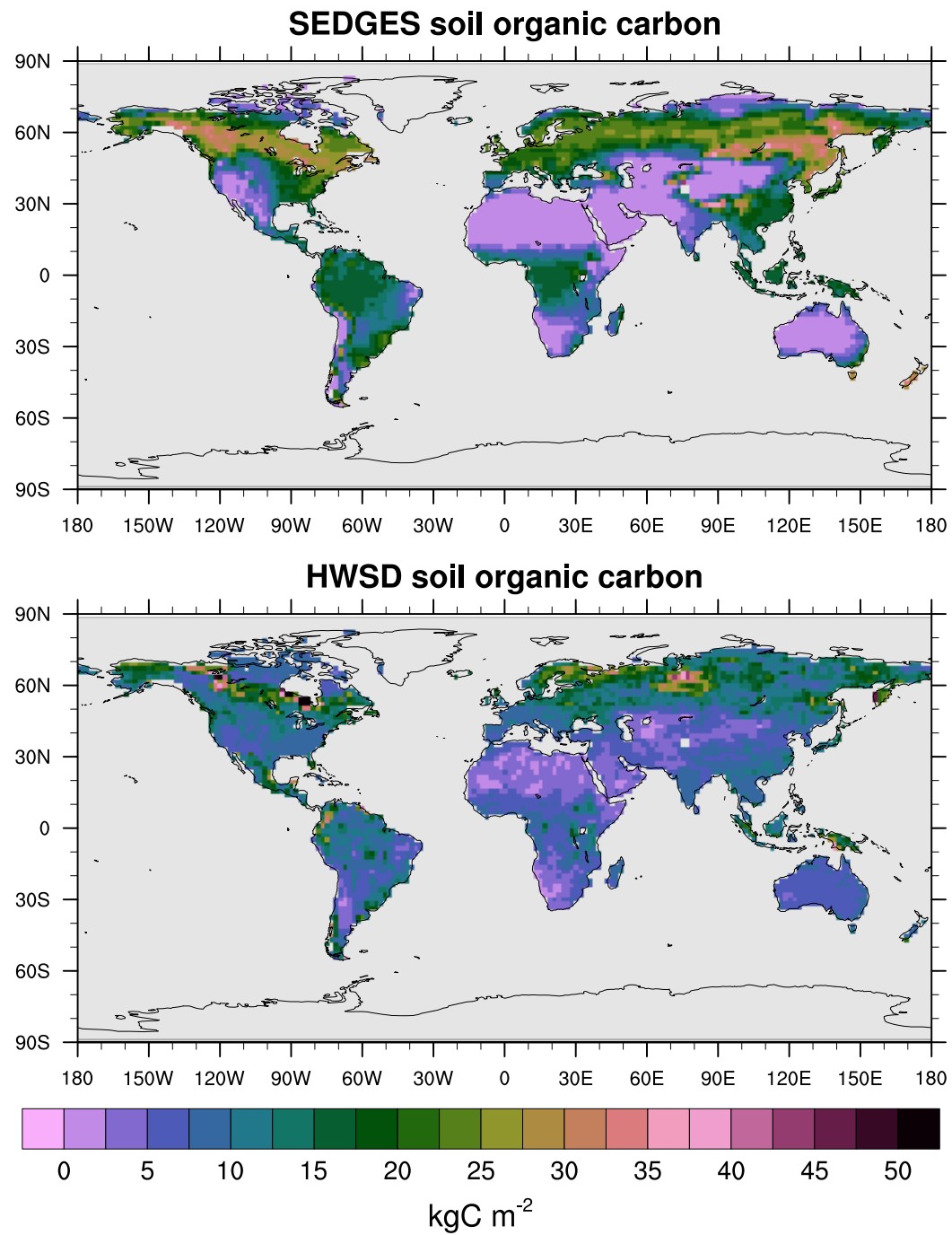

**Figure A13.** Soil organic carbon for SEDGES: mean over 1981-2010 of the transient $CO_2$ simulation; soil organic carbon from the Harmonized World Soil Database (HWSD) reference dataset (Wieder et al., 2011). The HWSD dataset has values for the top meter of the profile, only.

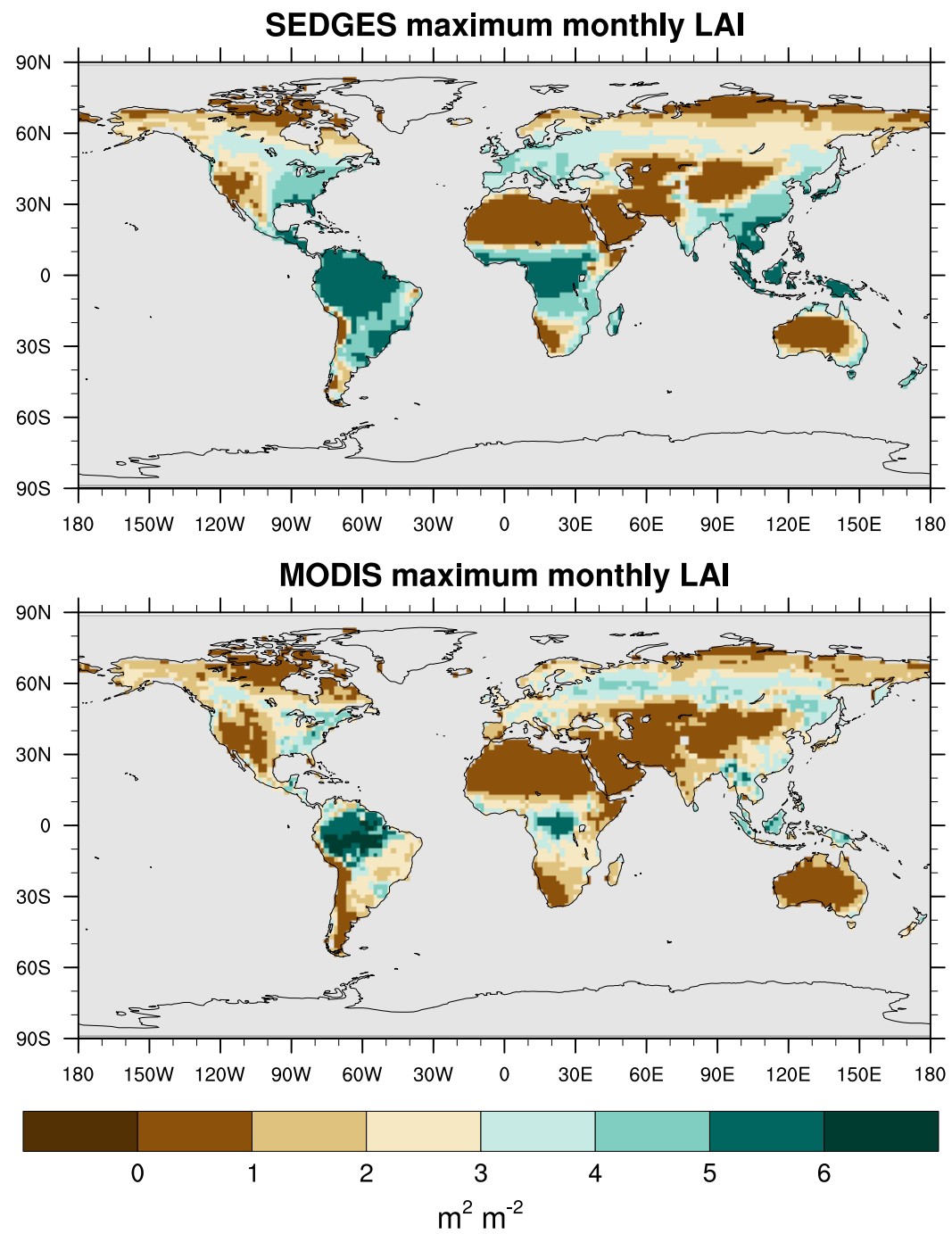

**Figure A14.** Maximum leaf area index (LAI) of the multi-year monthly means for SEDGES and for the BNU MODIS-based reference dataset for the 2001-2010 period.

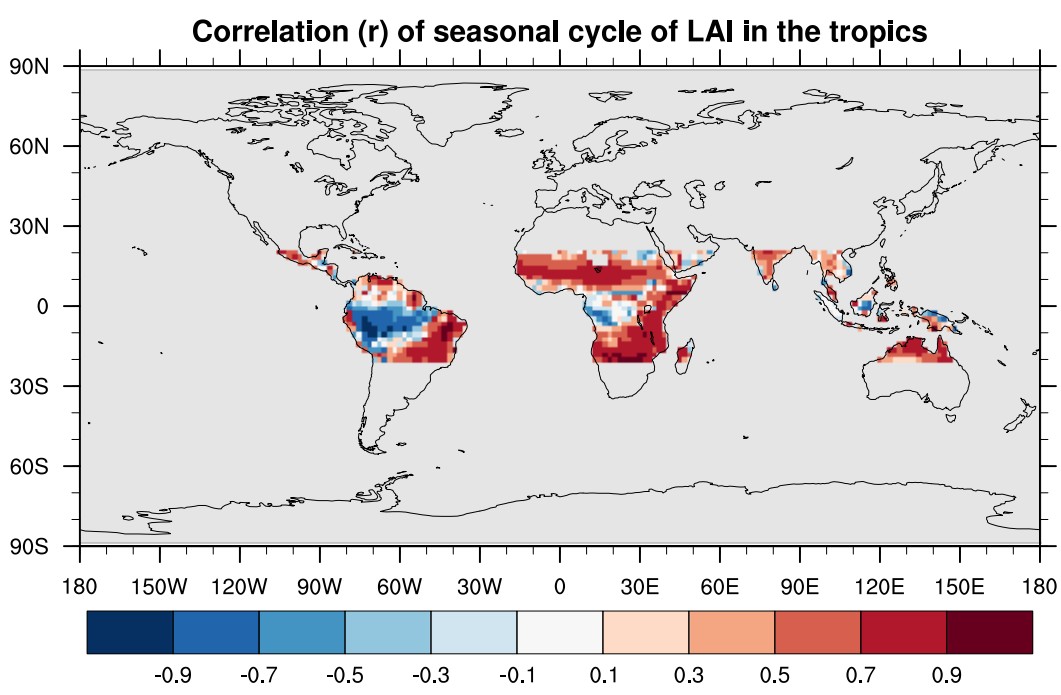

**Figure A15.** Temporal correlation at each grid point in the tropics (-20° to 20°) between the multi-year monthly means of LAI for 2001-2010 of SEDGES and of the BNU MODIS-based reference dataset. Some desert grid points do not have values because they have LAI's of 0 in one or both sets of data for all 12 months.

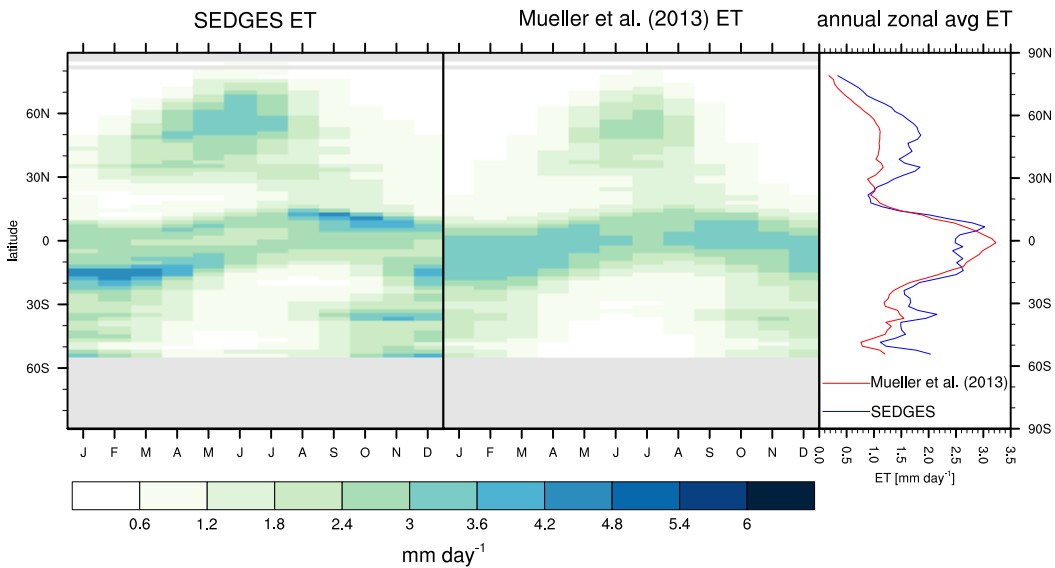

**Figure A16.** Zonal multi-year monthly means and zonal multi-year annual means of evapotranspiration (ET) for SEDGES and the Mueller et al. (2013) reference dataset for the 1989-2005 period over non-glaciated land.

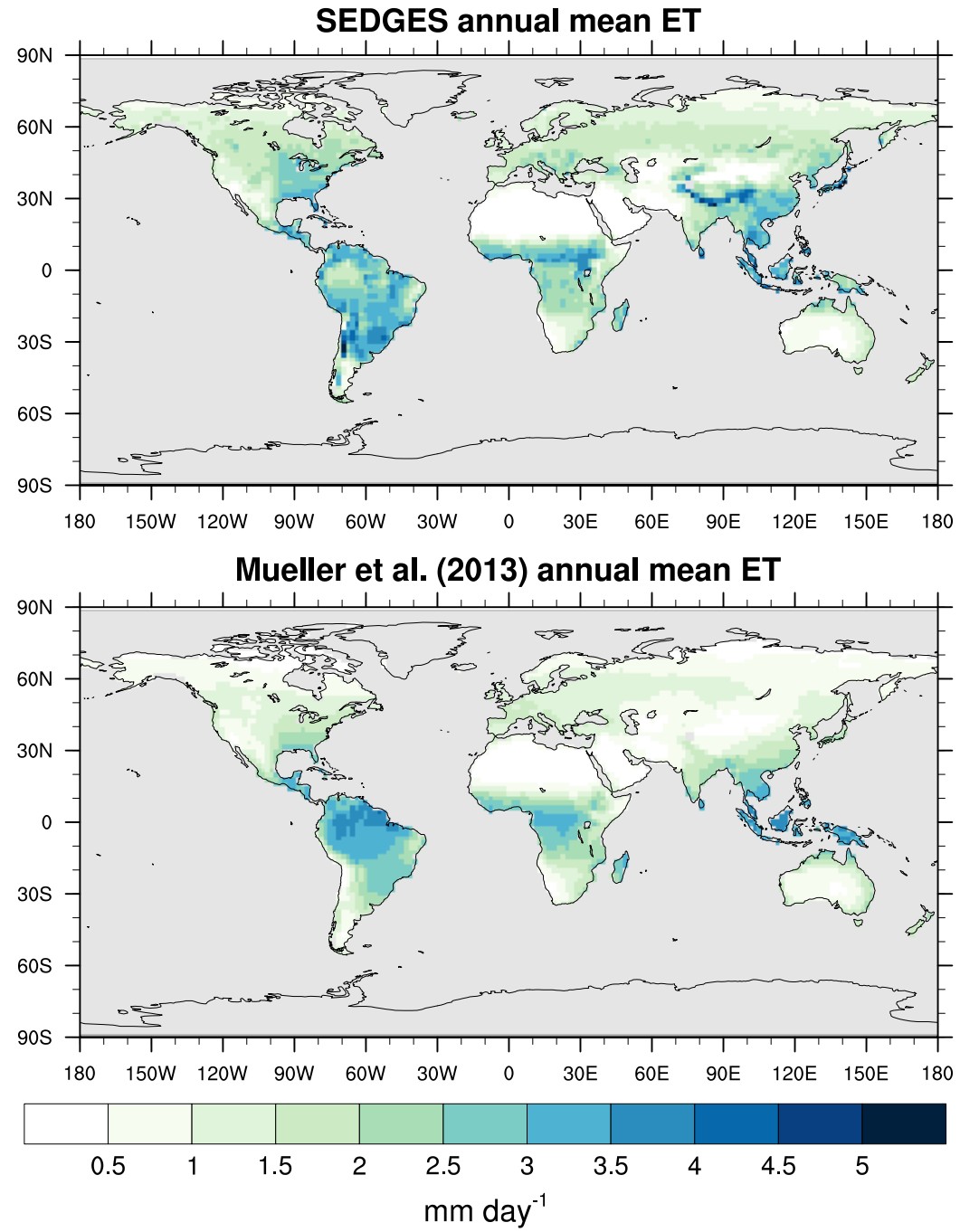

**Figure A17.** Multi-year annual means of evapotranspiration (ET) for SEDGES and the Mueller et al. (2013) reference dataset for the 1989-2005 period over non-glaciated land.

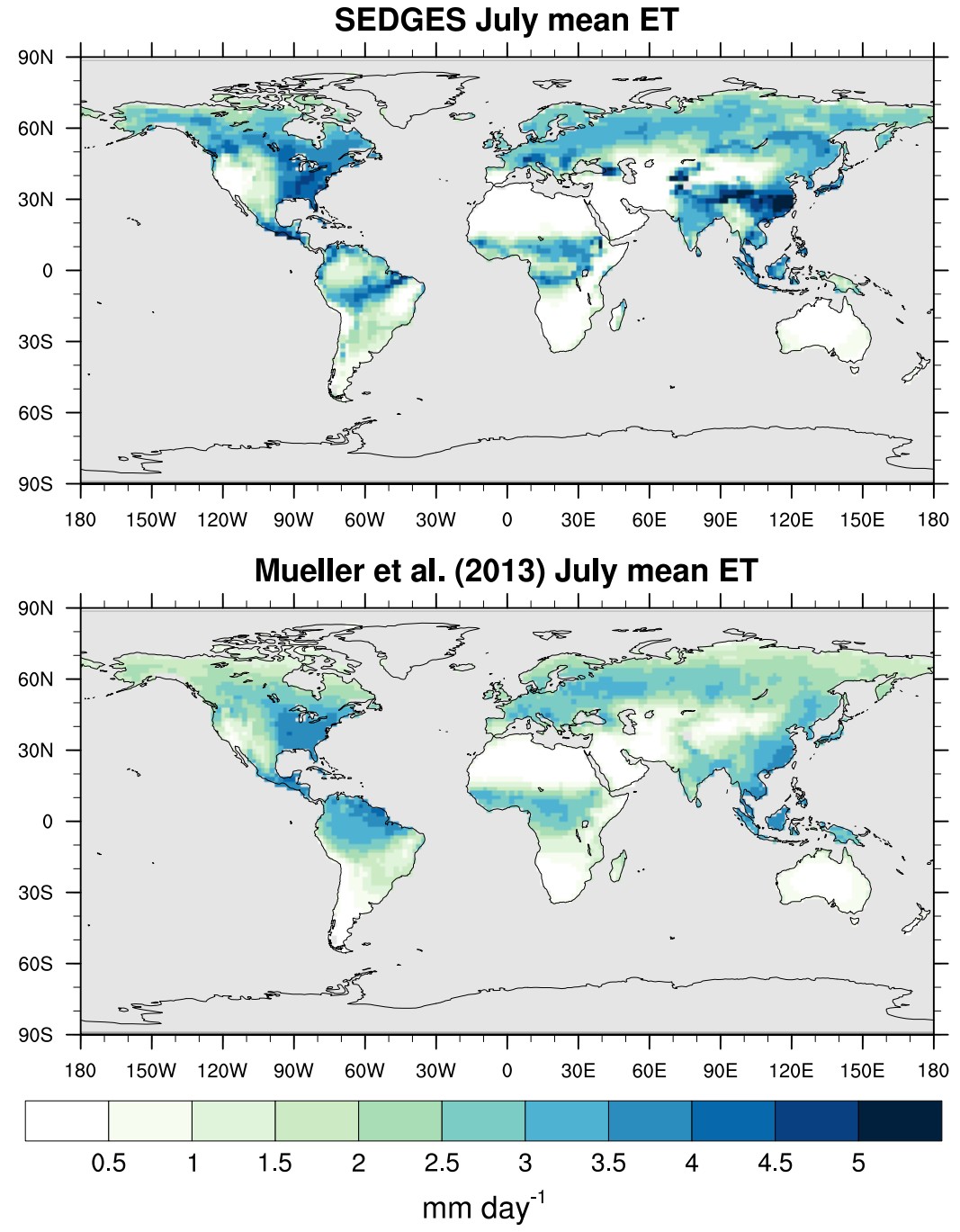

**Figure A18.** July mean evapotranspiration (ET) for SEDGES and the Mueller et al. (2013) reference dataset for the 1989-2005 period over non-glaciated land.

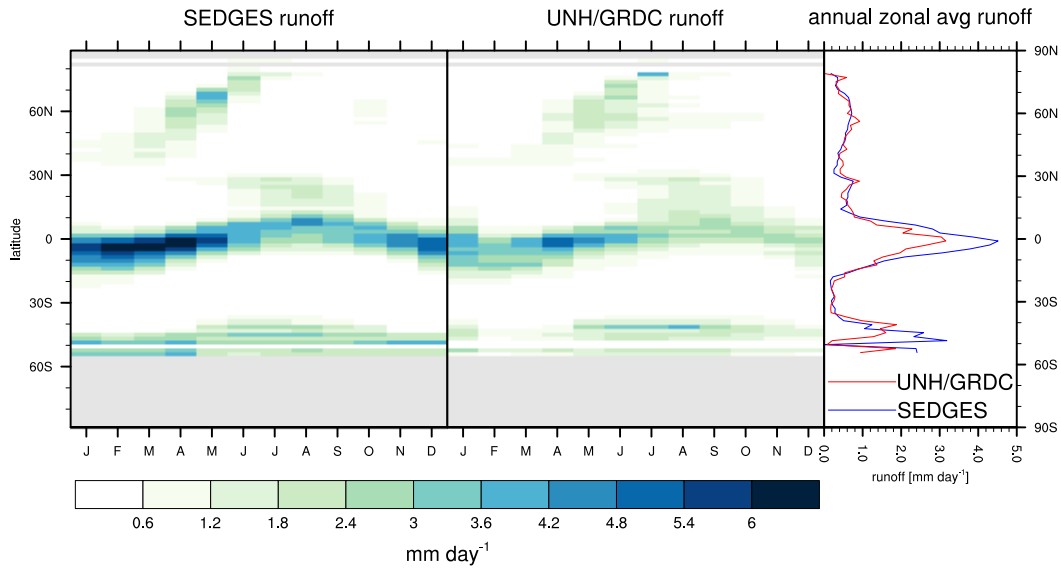

**Figure A19.** Zonal monthly mean and zonal annual mean climatologies of runoff for SEDGES and the UNH-GRDC reference dataset (Fekete et al., 2002) over non-glaciated land.

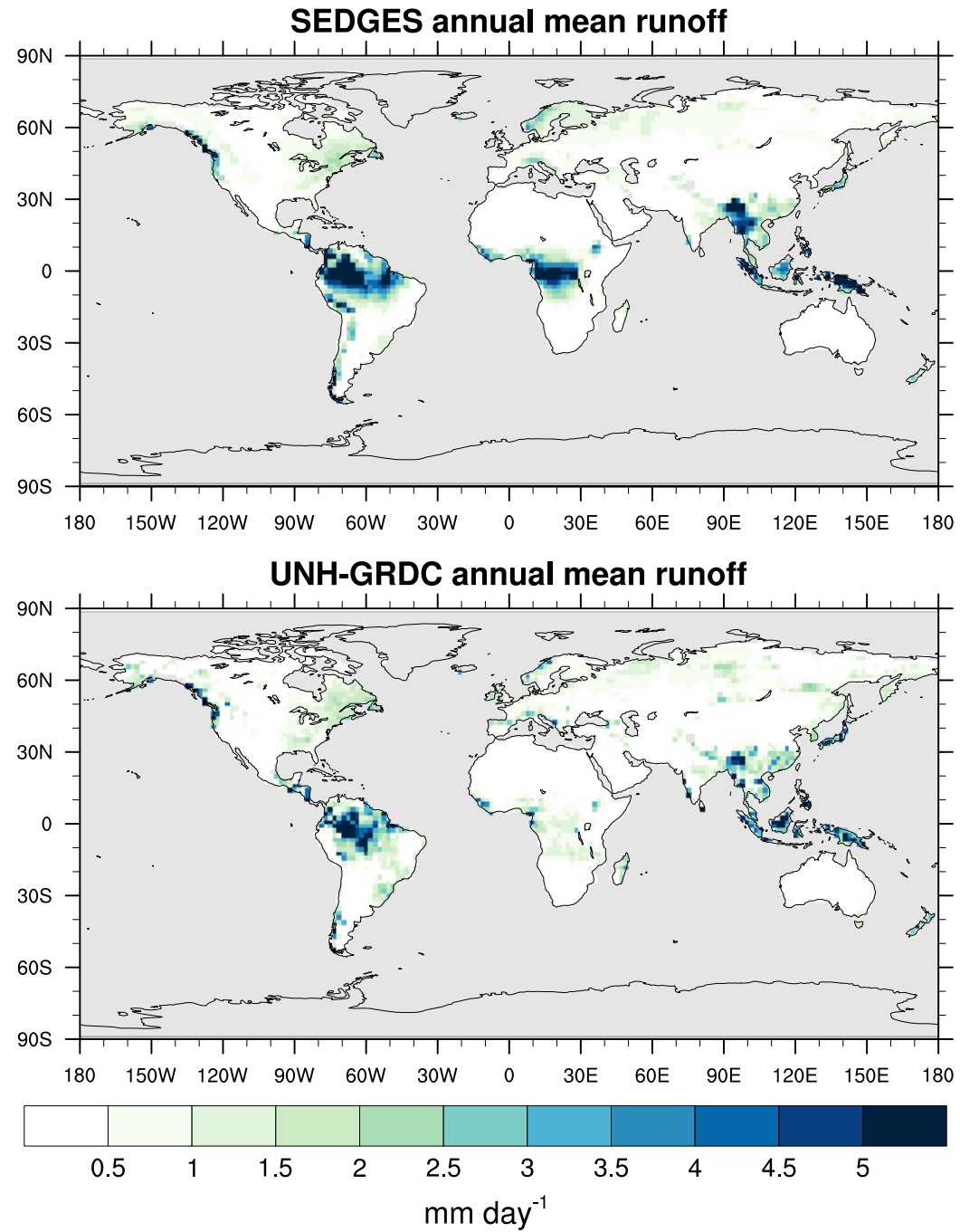

**Figure A20.** Climatological annual mean of runoff for SEDGES and the UNH-GRDC reference dataset (Fekete et al., 2002) over non-glaciated land.

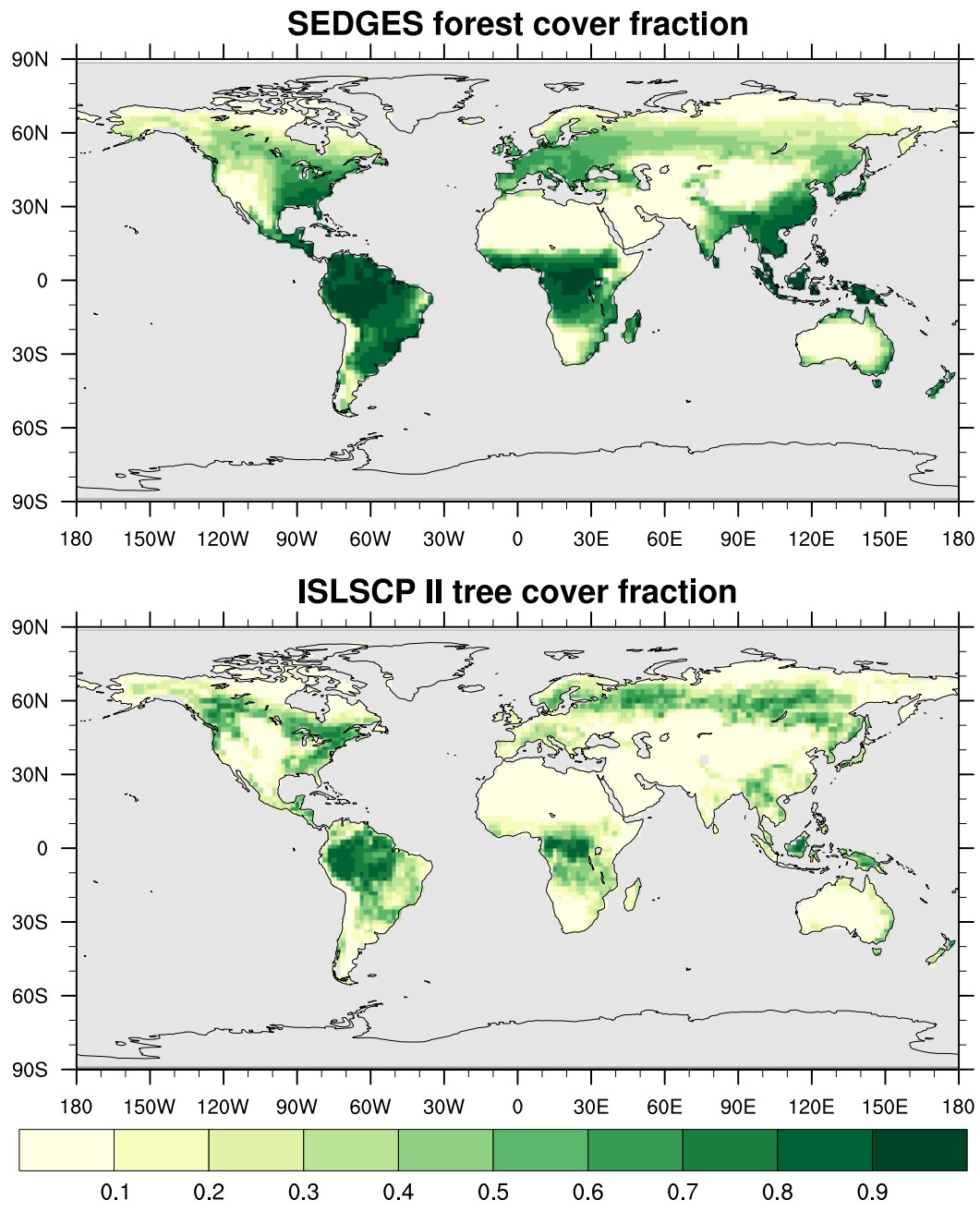

**Figure A21.** Mean forest cover fraction for SEDGES and tree cover fraction from the reference ISLSCP II dataset (DeFries and Hansen, 2009).

**List of Tables**

**Table 1.** SEDGES variables in the paper. Notation for the rightmost column is as follows: "I", "O", and "M" indicate variables that are, respectively, input to, output by, or modified by the SEDGES model. A blank field in that column indicates that the variable *could* be output if desired, but some code changes would be needed. "EI" denotes a variable from the ERA-Interim Reanalysis that is used by the simple land surface scheme that we use to drive SEDGES with in section 4. We denote by "M" only those output variables that are output by SEDGES that are expected to be already existing in the land surface scheme in which SEDGES is embedded.

| symbol | units | description | relationship with SEDGES |
|---|---|---|---|
| $C_{veg}$ | $kgC\,m^{-2}$ | vegetative carbon | O |
| $L$ | $kgC\,m^{-2}\,s^{-1}$ | litterfall | O |
| $NPP$ | $kgC\,m^{-2}\,s^{-1}$ | net primary productivity | O |
| $GPP$ | $kgC\,m^{-2}\,s^{-1}$ | gross primary productivity | O |
| $GPP_L$ | $kgC\,m^{-2}\,s^{-1}$ | light-limited gross primary productivity | O |
| $GPP_W$ | $kgC\,m^{-2}\,s^{-1}$ | water-limited gross primary productivity | O |
| $f_1(CO_2)$ | - | $CO_2$ fertilization function | |
| $f_2(T_{sfc})$ | - | temperature limitation function | O |
| $T_{sfc}$ | K | surface temperature | I |
| $f_{APAR}$ | - | fraction of photosynthetically active radiation (PAR) that is absorbed by green vegetation | |
| $SW\downarrow$ | $W\,m^{-2}$ | surface downwelling short wave radiation | I |
| $LAI$ | $m^2$ leaf area $(m^2$ ground area$)^{-1}$ | leaf area index | O |
| $f_{leaf}$ | - | vegetative leaf cover fraction | O |
| $g_a$ | $m\,s^{-1}$ | aerodynamic conductance | I |
| $r_a$ | $s\,m^{-1}$ | aerodynamic resistance | |
| $r_c$ | $s\,m^{-1}$ | canopy resistance | O |
| $\rho$ | $kg\,m^{-3}$ | surface air density | I |
| $p_{sfc}$ | Pa | surface pressure | I |
| $ET$ | $m^3\,m^{-2}\,s^{-1}$ | evapotranspiration | calculated outside of SEDGES |
| $qsat_{sfc}$ | $kgH_2O\,kgair^{-1}$ | surface saturation specific humidity | calculated outside of SEDGES |
| $q$ | $kgH_2O\,kgair^{-1}$ | specific humidity at the lowest atmospheric level | EI |
| $C_w$ | - | surface wetness factor | M |
| $\beta_{ss}$ | - | soil surface water stress factor | |
| $r_{ss}$ | $s\,m^{-1}$ | soil surface resistance | |
| $W_{frac}$ | - | soil wetness fraction | |
| $W_{soil}$ | m | soil water content | I |
| $W_{max}$ | m | soil "bucket" depth | M |
| $T$ | $m^3\,m^{-2}\,s^{-1}$ | transpiration | O |
| $r_{cu}*$ | $s\,m^{-1}$ | case-specific unconstrained canopy resistance | |
| $r_{cu}$ | $s\,m^{-1}$ | unconstrained canopy resistance | |
| $\beta_{tr}$ | - | water stress factor for transpiration | |
| $r_c min$ | $s\,m^{-1}$ | minimum canopy resistance | O |
| $C_{soil}$ | $kgC\,m^{-2}$ | soil organic carbon | O |
| $R_{soil}$ | $kgC\,m^{-2}\,s^{-1}$ | soil respiration rate | O |
| $T_{soil}$ | K | soil temperature at 0.20m depth | I |
| $LAI_m$ | $m^2$ leaf area $(m^2$ ground area$)^{-1}$ | leaf area index without soil moisture stress | |
| $f_{leaf_m}$ | - | (green) leaf cover fraction in absence of soil moisture stress | |
| $f_{leaf_{dry}}$ | - | max vegetative leaf cover fraction under soil moisture stress | |
| $f_{for}$ | - | forest cover fraction | O |
| $\alpha_0$ | - | snow-free surface albedo | |
| $\alpha_{soil}$ | - | albedo of bare soil | |
| $\alpha$ | - | albedo | M |
| $\alpha_{snowflat}$ | - | snow-covered albedo of flat portion of grid cell | |
| $\alpha_{snowfor}$ | - | snow-covered albedo of forested portion of the grid cell | |
| $f_{snowflat}$ | - | fraction of "flat" portion of grid cell that is snow-covered | |
| $swe$ | $m^3\,m^{-2}$ | snow depth in liquid water equivalent | I |
| $\alpha_{deepsnowflat}$ | - | albedo of deep and pure snow | |
| $z_0$ | m | surface roughness | M |
| $z_{0oro}$ | m | surface roughness due to orography | I |
| $z_{0veg}$ | m | surface roughness due to vegetation | |
| $P$ | $m^3\,m^{-2}\,s^{-1}$ | precipitation in liquid water equivalent | EI |
| $S$ | $m^3\,m^{-2}\,s^{-1}$ | snowfall in liquid water equivalent | EI |
| $M$ | $m^3\,m^{-2}\,s^{-1}$ | snowmelt in liquid water equivalent | EI |
| $ET_{soil}$ | $m^3\,m^{-2}\,s^{-1}$ | bare soil evaporation plus transpiration | see section 4 |
| $E_{soil}$ | $m^3\,m^{-2}\,s^{-1}$ | bare soil evaporation | O (when snow present) |
| $PET$ | $m^3\,m^{-2}\,s^{-1}$ | potential evapotranspiration | I |

**Table 2.** SEDGES parameters in the paper.

| symbol | value | units | description | source(s) |
|---|---|---|---|---|
| $\tau_{veg}$ | 10 | years (converted into seconds) | biomass residence time | SimBA (all versions) |
| $R_d$ | 287.0 | J K$^{-1}$ kg$^{-1}$ | gas constant for dry air on Earth | - |
| $\epsilon_{max}$ | $5.0 \times 10^{-10}$ | kgC J$^{-1}$ | max. light use efficiency | model calibration |
| $CO_{2comp}$ | 40 | ppmv | $CO_2$ light compensation point | Franks et al. (2013) |
| $T_{crit}$ | 20 | $^\circ$C | temperature at which productivity limitation begins | see section 2.2.3 |
| $k_{veg}$ | 1 | - | light extinction coefficient | see section 2.2.3 |
| $\Omega_c$ | 0.7 | - | clumping index | Pisek et al. (2010); He et al. (2012) |
| $co2conv$ | $4.15 \times 10^{-7}$ | kgC kgair$^{-1}$ ppmv$^{-1}$ | unit conversion factors | manipulation of equation B7 from Raupach (1998) |
| $\frac{c_i}{c_a}$ | 0.80 | - | ratio of intercellular to atmospheric $CO_2$ | somewhat common daytime value for C3 plants |
| $r_{ssmin}$ | 10 | s m$^{-1}$ | minimum soil surface resistance | van de Griend and Owe (1994) |
| $r_{ssmax}$ | $10^{30}$ | s m$^{-1}$ | maximum soil surface resistance | - |
| $\rho_w$ | 1000 | kg m$^{-3}$ | density of liquid water | - |
| $f_{snowfor}$ | 0.12 | - | snow-covered fraction of the forest cover | see section 2.2.5 |
| $trmax$ | $2.78 \times 10^{-7}$ | m s$^{-1}$ | max. transpiration rate | Knorr (2000) |
| $r_cmin_{min}$ | 0 | s m$^{-1}$ | absolute min. canopy resistance | - |
| $r_cmax$ | $10^{30}$ | s m$^{-1}$ | max. canopy resistance | Sitch et al. (2003) |
| $c_8$ | $\approx 43.3$ (see section 2.2.7) | - | for normalizing 10 $^\circ$C soil respiration to that of SimBA | |
| $c_9$ | 106 | K | for soil respiration | Jenkinson et al. (1990) |
| $LAI_{min}$ | 0.05 | - | min. leaf area index in wet soils | - |
| $LAI_{max}$ | 7 | - | max. leaf area index in wet soils | model calibration |
| $c_6$ | 0.195 | kgC$^{-1}$ m$^2$ | biomass to $LAI$ conversion | model calibration |
| $W_{frac_{crit,lai}}$ | 0.05 | - | critical soil wetness fraction for commencement of leaf fall | model calibration |
| $c_1$ | 0.2 | kgC$^{-1}$ m$^2$ | biomass-forest cover relationship | see section 2.2.9 |
| $c_2$ | 1.0 | kgC m$^{-2}$ | biomass threshold for forest cover commencement | see section 2.2.9 |
| $c_7$ | 9 | kgC m$^{-2}$ | soil organic carbon saturation value with respect to soil albedo | see section 2.3.1 |
| $\alpha_{sand}$ | 0.32 | - | sandy soil albedo | see section 2.3.1 |
| $\alpha_{peat}$ | 0.12 | - | albedo of organic matter-rich soil | see section 2.3.1 |
| $c_4$ | 1.5 | kgC$^{-1}$ m$^2$ | shape parameter for snow-covered albedo | model calibration |
| $c_5$ | 1.5 | kgC m$^{-2}$ | biomass threshold for snow masking | model calibration |
| $\alpha_{mindeepsnowflat}$ | 0.40 | - | albedo of warm, deep, pure snow | Roesch et al. (2001) |
| $\alpha_{maxdeepsnowflat}$ | 0.80 | - | albedo of cold, deep, pure snow | Roeckner et al. (2003) |
| $\alpha_{maxsnowfor}$ | 0.30 | - | maximum albedo of snow-covered forest | Moody et al. (2007) |
| $c_{12}$ | 0.10 | kgC$^{\frac{1}{2}}$ | conversion of biomass into soil bucket depth | model calibration |
| $W_{maxmin}$ | 0.05 | m | minimum soil bucket depth | see section 2.3.2 |
| $z_{0min}$ | 0.01 | m | surface roughness for bare soil | Oke (1987) |
| $z_{0const}$ | $\approx 0.035$ | m | biomass-roughness relationship | see section 2.3.3 |
| $c_{15}$ | 8 | kgC m$^{-2}$ | biomass-roughness relationship | model calibration |
| $c_{16}$ | 0.5 | kgC$^{-1}$ m$^2$ | biomass-roughness relationship | model calibration |
| $c_{17}$ | 2.5 | m | $\approx$ surface roughness for fully-forested land | typical value for tropical rainforests (Sellers et al., 1996b) |

**Table 3.** RMSE's and correlations between multi-year annual means of SEDGES variables and those for reference datasets

| variable | correlation | RMSE | reference dataset | analyzed years |
|---|---|---|---|---|
| $ET$ | 0.778549 | 0.72505 mm d$^{-1}$ | Mueller et al. (2013) | 1989-2005 |
| $GPP$ | 0.924035 | - | MTE (Jung et al., 2011) | 1990-2009 |
| $GPP$ | 0.861378 | - | CARBONES | 1990-2009 |
| vegetative carbon | 0.570484 | 3.91639kgC m$^{-2}$ | Olson et al. (1985) | see text |
| soil organic carbon | 0.579095 | 7.93438kgC m$^{-2}$ | HWSD v.1.2 (Wieder et al., 2011) | see text |