# Peer review of "Description and Validation of the Simple, Efficient, Dynamic, Global, Ecological Simulator (SEDGES v.1.0)"

_Geoscientific Model Development, 2017_

## Referee Comment (RC1) · Anonymous Referee #1 · 12 Jun 2017

Review of paper: "Description and Validation of the Simple, Efficient, Dynamic, Global, Ecological Simulator (SEDGES v1.0). By Paiewonsky and Timm.

Abstract: The wording could be tightened in the Abstract to explain how SEDGES is "auxiliary" to the land-atmosphere coupling scheme. This will make it easier to understand better the sentence "evaluate….using a simple land surface scheme that is driven by reanalysis data". If SEDGES is compared extensively to another existing land surface model as ground truth, then from the Abstract alone, the reader will wonder what it does in addition.

Extending further the comment above, maybe get in the Abstract a list of ecological

variables and quantities that SEDGES generates, but are non-standard for most land surface models?

Introduction. Around line 20. JULES land surface model is also one that now has ecological components in it, for instance land fraction changes and terrestrial carbon stores. These are derived as a function of land-atmosphere water and $CO_2$ fluxes.

Sentence starting "In such a framework" gives two reasons for needing a more sophisticated model. But the third, is potentially a reason for keeping a simple model i.e. "computational burden...for increased complexity".

P2, lines 1-9 reads as if unduly critical of existing analyses – potentially of the original model developers. Was there really no validation of SimBA?

P2, lines 10-15. This really is the point at which more information should be provided on what SEDGES actually does, at an over-arching level, before leading in to detailed Model Description. What are the core additional quantities that SEDGES generates, and that are in addition to existing land surface models? It is also still vague about "model evaluation". Both SEDGES and another trusted land surface model are forced simultaneously with reanalysis data, and certain diagnostics compared?

P2,3 Overview. Again, please present what is different about SEDGES compared to existing models. In my view, it is that the model links more tightly land-atmosphere flux exchange components with other land surface elements that more traditionally would go in to Dynamic Global Vegetation Models (DGVMs). Hence p5, line 3 "Forest cover and leaf cover fractions", these are dynamic vegetation components that would not always be found within a standard land surface model.

Page 3. Assumption that NPP/GPP=0.5. This does feel like a very large assumption, and particularly as thermal responses in respiration might behave differently to gross photosynthesis. The authors themselves appear cautious, with the caveat that this might be accurate on very long time scales. However, this does mean that in compar-

ison with other land surface models, then only long-term averages of NPP and GPP should be compared.

Page 4,5 The Tables are excellent, and highly appropriate for a model development journal. It is also appreciated that all units are presented, and where justification of parameters is linked to existing literature.

Page 6. However, the caption for Table 3 could be clearer? I'd also avoid the word "climatologies" for things like ET, GPP. For a moment, I thought momentarily this might be referring to climatological drivers (so temperature, humidity etc).

Page 6 lines 5-9, and a few other places. The lines have been wrapped incorrectly, with a new line starting after each ";" symbol (or ",", e.g. p9, lines 6-9).

Page 7. There is some evidence now that splitting SW radiation in to direct and diffuse can have an influence on PAR. Is this something the authors considered, or maybe for the next model version?

The paper makes quite a lot of use of footnotes. Some of these feel more natural simply being in the main text itself?

P11-14. These variables are the more novel parts of this land surface model, and it might be appropriate to re-iterate this point? That is, components more associated with carbon stores than the fluxes.

P12, 13. The terminology could be made clearer between leaf cover fraction and forest cover fraction. In some DGVMs, these could potentially be the same thing. I guess from Equation (18) this is to do with wilting of leaves and that does not appear in forest cover fraction. It also includes seasonal phenology?

P17, Section "How to Couple SEDGES". Unfortunately at this point in the paper, I am again confused as to exactly what SEDGES is, given that it needs the variables listed lines 24,27. The issue here is that some of these components do not uncouple? For instance, if SEDGES predicts LAI, then altered LAI will adjust transpiration, in turn

affecting soil moisture content. So soil moisture content cannot be regarded as a pure input? I am happy to accept that I might not have fully understood the direction the paper is taking, but this could be made clearer. I can see that there are hints of this discussion around the middle of page 18.

P18. Related to the point above, in Equation (29), is ETsoil derived from SEDGES? If so, then Wsoil becomes a diagnostic, rather than an independent forcing.

P20. Model Evaluation. The comparison against observation-based datasets of things like GPP is an important and novel part of this paper, and indeed this should be more routine for all land surface modellers. However I am less convinced by the need to compare against other land surface models. Whilst any model developer may want to check if their simulations are outliers, there is a risk with fitting to other models. This creates a circularity and gives overly small uncertainty bounds that captures uncertainty expressed across land surface models.

The exception to the point above is if wishing to check components of SEDGES which are not considered as the new novel components – instead just verify such "drivers" are reasonable. In these circumstances, then the paper really needs to make a very clear statement. Something along the lines of "Many parts of SEDGES are common to other land surface models. We adopt similar approaches to other models for these parts. This then allows us to make projections of new terrestrial attributes that are important but not routinely in other models. These include. . . . . ."

---

## Short Comment (SC1) · 12 Jun 2017

Overview:

P. Paiewonsky and O.E. Timm are developed a Land surface model (LSM) call SEDGES and test it against global observation of GPP, carbon stock, LAI etc... They attempted to create a simple LSM in order to better understand which processes impact atmospheric change in coupled simulations with a global circulation model (GCM) calls PlaSim. This effort of simplification is welcome in global modeling science since the last generation of LSM are sometimes too complex to understand which are the most important processes that link the biosphere and the atmosphere. The description of SEDGES is precise and permit me to understand how SEDGES calculates different processes like carbon sequestration, evapotranpiration ...etc. Each simplifications/modifications are well documented and relevant arguments are proposed to support their choices. But in the validation process, the authors are more focus on showing how better SEDGES simulate GPP and compared their results to the state-of-the-art LSM (ORCHIDEE, JULES, and CLM4CN). I think it will be better to pay more attention to understand how far SEDGES can be simplified before losing efficiency in coupled simulation. A lot of models of this level of complexity was developed in the past (see review of Pitman et al. 2005) and SEDGES need to be replaced in this context. Authors must answer these questions : "Why SEDGES are different from others second generation LSM ?", and "How they deal with the trade-off between simplification, precision, robustness ?". I think this study can be published with deep changes in the aim and the structure by adding relevant tests and also being more honest on results they will decide to show or not.

In detail:

To clarify the aim of the study and increase the impact of this paper, I have some comments :

- First, I do not understand if SEDGES is a new model built from scratch by picking some ideas from other models like, SimBA, VECODE, ENTS, or if this is an improvement of SimBA. Especially when I read this sentence in introduction : « SEDGES is based on the original SimBA model (Kleidon, 2006b), which was coupled to the Planet Simulator (PlaSim) [...] even more strongly based on a later version of SimBA (Lunkeit et al., 2011), also coupled to PlaSim. » This sentence are in contradiction with the sentence in the conclusion : « A new simplified model for the representation of dynamic ecological processes for use in conjunction with climate models has been developed. » For me an improved version of SimBA are more suitable than another second new generation LSM. But if the authors will add features to SimBA, they increase the model complexity, they do not create simple model. The authors need to clearer this part !
[Figure]

- The authors argue that the GPP was better simulated by SEDGES than other state-of-the-art LSM models but strangely, authors only show this comparison for GPP. Is there something they don't want to show for ET, LAI, soil carbon, albedo ...etc ? In addition, a better comparison was to test their results with models of the same level of complexity like second generation LSM, SimBA, VECODE, ENTS. A comparison with far more complex LSM like ORCHIDEE, JULES, and CLM4CN are not relevant in this context because these LSMs are not developed just to provide information to GCM but also to understand global dynamics of vegetation across centuries that SEDGES are not able to simulates.

- To convince others that SEDGES is a good LSM for coupled simulations, the authors must test SEDGES with a coupled simulation and check if the simplifications/modifications they made, have an impact on the GCM outputs. Otherwise, a least, authors must write a couple of sentence to explain why this test was not done and when they plan to realize this essential step. When I read the conclusion part, I have the feeling that the authors are convinced that SEDGES are already validated for coupled simulations and no more tests are needed : « In conclusion, we feel that SEDGES provides a new viable and computationally efficient alternative to currently-implemented terrestrial vegetation/ecological models, [...] »

- In part dedicated to LAI. The authors explain that LAIm is only dependent to vegetation carbon stock (Cveg) and constraint by LAImin and LAImax (from calibration). In fact, LAI is not really use in SEDGES because LAIm is directly translated into Forest cover (fveg). Why do you need to pass by LAI ? Why not just calculate fveg with Cveg. I have the impression that the authors just put LAI in SEDGES just to say, "we have LAI". It becomes even dangerous when the authors try to validate SEDGES against observed LAI the same data used for calibration !

- Discussion part must be developed. For each matchless outputs, the authors must explain why they obtain this result and how this lack of precision can affect the outputs of coupled simulations.

- Figures are in general in a good shape but too much effort was made for spatial variations and less for temporal variations.

---

## Referee Comment (RC3) · Anonymous Referee #3 · 23 Jun 2017

The paper entitled "Description and Validation of the Simple, Efficient, Dynamic, Global, Ecological Simulator (SEDGES v1.0)" by Pablo Paiewonsky and Oliver Elison Timm describes a simple vegetation model SEDGES that is suitable for coupling to climate models with intermediate complexity. This model builds on SimBA model, but with modifications on representing gross primary productivity (GPP) mainly by including plant regulation of canopy resistance via coupling of light-dependent photosynthesis and transpiration. The SEDGES model also includes the dependency of bare soil albedo on soil organic carbon. Model predictions on the productivities (i.e. GPP and NPP) and properties (i.e. LAI, vegetation carbon) of vegetation, the properties of soil (i.e. soil albedo and carbon storage), and variables relevant to hydrological cycle (i.e.

ET and runoff) have been evaluated against various datasets. Generally, I found that the model description is clear. Their idea on developing simple Land Surface Models (LSMs) is welcome and should be encouraged, given that the current complex LSMs introduce large computational burden and untraceable to the key processes underlying experimental results.

My major concern is that the improvement of model performances between SimBA and SEDGES is not clearly demonstrated. Since SEDGES builds on SimBA, to emphasize the value of this work, the advance of SEDGES needs to be well manifested. For example, the authors show particularly well-simulated GPP, but it is not clear how much SEDGES improves the representations and simulations of GPP in SimBA. In other words, whether this well-simulated GPP is due to the incorporated processes in SEDGES, or due to the original framework set up in SimBA model? The authors need to prove that SEDGES indeed improves the GPP simulation compared to that from SimBA. Maybe adding the SimBA simulations in those relevant figures is the easiest way to illustrate. It will be even better if the authors could provide a short summary of how GPP is modeled in SimBA.

Similarly, the authors state that SEDGES improves most of the parameterization of SimBA, but again it is not clear whether or not those modifications of parameterization indeed improve SimBA simulation. The lesson we learnt from current LSMs tells us that increasing the complexity not necessarily guarantees a better model performance.

Therefore, I am also concerned about the trade-off between realism and simplicity. To balance this trade-off with the purpose of improving the reliability and robustness of models, the added processes or modified parameterizations need to be proved as necessary for improving the reliability of the model. Otherwise, those modifications on parameterizations follow the same routine as the current complex LSMs being developed. Moreover, the simplifications also need to prove as reasonable. For example, the ratio of $c_i/c_a$ in Equation (6) is considered as a constant, but has been shown that the optimal stomatal behavior allows $c_i/c_a$ decreases with VPD, increase with temper-

Interactive
comment

ature (e.g. Prentice et al. 2014, Medlyn et al. 2011, Lin et al. 2015). The variation in ci/ca seems quite important in terms of capturing the spatial pattern of GPP (Wang et al. 2014). This simplification needs a justification.

Some minor comments on equations of GPP:

I think it is ci (the intercellular CO2 concentration) that really matters in CO2 fertilization. If you consider ci for water-limited GPP in Equation (6), why not here for light-limited GPP?

Typically, modeled LUE (not only fAPAR) is represented with a dependency on vapor pressure deficit, why the equation here does not include such an effect? Is it implicitly considered via the coupling with water-limited GPP?

References

Lin, Yan-Shih, et al. (2015), 'Optimal stomatal behaviour around the world', Nature Climate Change. Medlyn, Belinda E, et al. (2011), 'Reconciling the optimal and empirical approaches to modelling stomatal conductance', Global Change Biology, 17 (6), 2134-44. Prentice, I Colin, et al. (2014), 'Balancing the costs of carbon gain and water transport: testing a new theoretical framework for plant functional ecology', Ecology letters, 17 (1), 82-91. Wang, H, Prentice, IC, and Davis, TW (2014), 'Biophsyical constraints on gross primary production by the terrestrial biosphere', Biogeosciences, 11 (20), 5987-6001.

---

## Author Comment (AC1) · 15 Jul 2017

**Interactive comment**

**Reply to Referee #2**

July 15, 2017

To being with, we thank the reviewer for taking the time to comment on our paper and provide us with feedback.

```
I think it will be better to pay more attention to understand
how far SEDGES can be simplified before losing efficiency in
coupled simulation.
```

We disagree that this is necessary.

The reviewer then suggests that we compare and contrast SEDGES and its assumed framework (i.e. SEDGES and the type of land surface model that it presupposes that it forms a part of) with other second generation land surface models. The intention behind this suggestion is reasonable and this sort of comparison might be useful to some readers, so we will include a couple of sentences on this comparison. Note that the SEDGES model framework is not a second generation land surface model. Rather, it has attributes that are specific to both the first and third generations of land surface models, as described by Pitman (2003) (note: we assume that the reviewer is referring to the review by Pitman (2003), instead of Pitman et al. (2005), which we could not locate).

The reviewer says that we must answer the question of how we deal with the trade-off between simplification, precision, and robustness. Prentice et al. (2015), in fact, suggest that robustness tends to be lost as land surface models increase in complexity. As we mention in our response to reviewer #3, most of the parameterization changes that are made to SimBA (on which SEDGES is based) do not significantly increase the complexity. However, when introducing the changes that do appreciably increase the complexity, we will provide some discussion on the aforementioned trade-off in the revised manuscript.

In response to the reviewer's comments on the lack of clarity as to what exactly SEDGES is, i.e. to what extent it is a model that is built from scratch as opposed to an improved version of SimBA, we will make this clearer in the revised manuscript.

```
But if the authors will add features to SimBA, they increase
the model complexity, they do not create simple model.  The
authors need to clearer this part !
```

In response, even though it is more complex than SimBA, SEDGES is still a simple model. One cannot say that it qualifies as "intermediate complexity". We will make this clear in the revised manuscript.

```
The authors argue that the GPP was better simulated by SEDGES
than other state- of-the-art LSM models but strangely, authors
only show this comparison for GPP. Is there something they
don't want to show for ET, LAI, soil carbon, albedo ...etc ?
```

Two things here:

We did not go as far as saying that GPP was overall better simulated by SEDGES than by state-of-the-art land surface models, except for the 20 year trend in GPP, in which SEDGES does surprisingly well.

It is not true that we only showed this comparison for GPP. We did some comparison

with the state-of-the-art LSM models for vegetative and soil carbon in sections 5.2 and 5.3, respectively. While it is true that we could make more comparisons between SEDGES and those state-of-the-art models with some of the other variables (ET and LAI), reviewer #1 questions the need for such comparisons, and reviewer #2 also states the following:

```
In addition, a better comparison was to test their results with
models of the same level of complexity like second generation
LSM, SimBA, VECODE, ENTS. A comparison with far more complex
LSM like ORCHIDEE, JULES, and CLM4CN are not relevant in this
context because these LSMs are not developed just to provide
information to GCM but also to understand global dynamics
of vegetation across centuries that SEDGES are not able to
simulates.
```

As a compromise, perhaps it would suffice to include an extra two figures showing the spatial annual averages of ET and runoff in SEDGES and how these compare to their reference datasets. We feel that more comparison with LAI is not needed, but would agree to it if the reviewer insists that we do so.

With regards to comparisons with other models of similar complexity, the third reviewer has already requested that comparisons with SimBA be made, and we did in our response. We can show such results for (the 2007 version of) SimBA in an appendix, perhaps for surface albedo, GPP, and ET. It is not worthwhile to put these results in the main text. SimBA has very severe problems when forced offline, i.e. when not coupled to Planet Simulator. See our response to reviewer #3 for ET and GPP figures. In the revised manuscript, we will add some qualitative comparison with ENTS (Williamson et al., 2006) for soil carbon, vegetative carbon, annual mean evaporation. Some comparison will be made with (offline-forced) VECODE (global mean NPP in Cramer et al. (2001) and tree fraction in Brovkin et al. (1997)), which are apparently the only published offline evaluations. We are hesitant to compare other (spatially-varying) ecological variables (maximum LAI, NPP, biomass, and soil carbon), as shown in Brovkin et al. (2002), when VECODE is coupled to CLIMBER-2, because the simulation of those variables depends not only on VECODE, but also on the simulation of climate variables by CLIMBER-2, which is an Earth system model of intermediate complexity, and may thus deviate too far from the actual climate for a fair comparison with reanalysis-forced SEDGES.

To clarify, SEDGES is a dynamic vegetation model, so it certainly can be used to understand some important dynamical aspects of vegetation across centuries, in contrast with what the reviewer states. We will make this clear in the revised manuscript.

```
To convince others that SEDGES is a good LSM for coupled
simulations, the authors must test SEDGES with a coupled
simulation and check if the simplifications/modifications they
made, have an impact on the GCM outputs.  Otherwise, a least,
authors must write a couple of sentence to explain why this
test was not done and when they plan to realize this essential
step.  When I read the conclusion part, I have the feeling that
the authors are convinced that SEDGES are already validated
for coupled simulations and no more tests are needed :  "
In conclusion, we feel that SEDGES provides a new viable and
computationally efficient alternative to currently implemented
terrestrial vegetation/ecological models, [...]  "
```

The reviewer raised a major concern that our first paper on the newly developed SEDGES model should also include an evaluation within a coupled climate model (or Earth System Model). We share the reviewer's concern that a stand-alone evaluation against present-day observations or in comparison with other models of similar complexity is not sufficient to make firm statements on how SEDGES performs in a coupled mode. Pablo Paiewonsky has been working with PlaSim-SEDGES and carefully analyzed the behavior of SEDGES inside that model for present-day climate (pre-industrial

climate conditions). However, we feel that an extension of the evaluation of SEDGES within ONE coupled climate model (namely PlaSim) already would add substantial amount of additional results (and discussions). Therefore, it would be better placed in a separate paper, along with an application of the coupled model for a specific research objective.

We are working on a second paper that will provide a detailed analysis of the PlaSim-SEDGES model. The results are part of Pablo Paiewonsky's PhD dissertation, which is going to become publicly available in August 2017.

As the reviewer knows very well, too, in a coupled system the cause-effect relations more often than not are difficult to assess. It could be therefore misleading if a poor evaluation of some variables that are numerically part of the SEDGES code is presented here, without going into an equally detailed evaluation of the physical part of the coupled system. For brevity we just highlight here, that the coupled model simulation, indeed shows some critically important model biases, for example, the equivalent of a grassland/shrub vegetation zone within the boreal forest zone of Siberia. Without careful study of the climatic conditions, and potential vegetation-climate feedbacks, there is no substantial value to adding the coupled-model evaluation into this paper, which has the ultimate purpose of introducing the model to the community at large.

[Moreover, we have identified a number of potential sources of coupling-issues that are intrinsic to PlaSim. For example, not all coupled models will have the same biases arising from low resolution (unresolved mountain topography), deficits in regional vegetation-climate feedbacks due to the use of a slab ocean model, and other physical parameterizations related to radiative fluxes and cloud processes, all of which can impact the results of the SEDGES component in PlaSim.]

In conclusion with regards to this concern, we hope that this paper with its stand-alone model description/evaluation of SEDGES can raise interest in the SEDGES model, and motivate modelers to couple our model with other GCMs or EMICs in near future. We

will revise our discussion and conclusions section accordingly and add a sentence that makes clear that the performance of SEDGES in a coupled system will need careful evaluation.

Moving on to the reviewer's comments on LAI, since the reviewer shows some confusion with respect to the variables and processes, it is possible that other readers will, too, and so we will revise the description to clarify the connections between (and roles of) LAIm, LAI, and fveg. fveg is leaf cover fraction, not forest cover fraction. LAIm is not directly translated into fveg, but rather fvegm (leaf cover fraction for moist soils). The reviewer questions the need to compute LAI at all. The final LAI does not need to be computed, as is said in the text in section 2.2.8. LAIm *does* need to be computed because it is directly used to compute fvegm, and thus, in conjunction with soil moisture limitation, fveg. We thank the reviewer for his or her point that it is dangerous to validate/evaluate SEDGES against observed LAI, when the observed LAI (from MODIS) was also used for calibration. In the revised manuscript, we will include some discussion on which datasets were used both for calibration and evaluation.

The reviewer further requests that we explain, for each lack of match between simulated output and reference dataset, why the mismatch occurs and what problems the lack of model precision could create in coupled simulations. We feel that we very often explain the reasons (or possible reasons) behind the mismatches between SEDGES and the reference datasets (used for evaluation) in our paper and that this is done adequately. To address the reviewer's other concerns, in the revised manuscript we will gladly try to anticipate conditions in which the model parameterizations would be expected to yield very inaccurate and/or untrustworthy results (particularly for non-modern conditions) and delve a bit into the ramifications of such model imprecision for coupled simulations, when we feel that they could be important. Such limitations are often not foreseeable from model performance under modern conditions (which is what is examined in the paper), but could be relevant for paleoclimate studies or if feedbacks in the coupled system amplify model imprecision in the offline mode.

References

Brovkin, V., Ganopolski, A., and Svirezhev, Y.: A continuous climate-vegetation classification for use in climate-biosphere studies, Ecological Modelling, 101, 251-261, 1997.

Brovkin, V., Bendtsen, J., Claussen, M., Ganopolski, A., Kubatzki, C., Petoukhov, V., and Andreev, A.: Carbon cycle, vegetation, and climate dynamics in the Holocene: Experiments with the CLIMBER-2 model, Global Biogeochemical Cycles, 16, 2002.

Cramer, W., Bondeau, A., Woodward, F.I., Prentice, I.C., Betts, R.A., Brovkin, V., Cox, P.M., Fisher, V., Foley, J.A., Friend, A.D. and Kucharik, C.: Global response of terrestrial ecosystem structure and function to CO2 and climate change: results from six dynamic global vegetation models, Global change biology, 7, 357-373, 2001.

Pitman, A. J.: The evolution of, and revolution in, land surface schemes designed for climate models, International Journal of Climatology, 23, 479-510, 2003.

Williamson, M., Lenton, T., Shepherd, J., and Edwards, N.: An efficient numerical terrestrial scheme (ENTS) for Earth system modelling, Ecological Modelling, 198, 362-374, 2006.

---

## Author Comment (AC2) · 15 Jul 2017

**Reply to Anonymous Referee #3**

July 15, 2017

To begin with, we thank the reviewer for raising numerous important issues in his or her critique of our paper. In particular, we hope that in our response, our discussion on the constant ci/ca in our model is useful to others in the land surface modeling community, too.

```
My major concern is that the improvement of model performances
between SimBA and SEDGES is not clearly demonstrated.  Since
SEDGES builds on SimBA, to emphasize the value of this work,
the advance of SEDGES needs to be well manifested.
```

We agree with the reviewer that the value of the development of SEDGES would be highlighted by demonstrating its improved performance as compared to SimBA. It is important to remember that there are basically two main versions of SimBA. The older version (Kleidon et al., 2006) has been used in several studies (which are cited in the main paper) and is included in version 15 of the Planet Simulator (Lunkeit et al., 2007). The newer version of SimBA was developed by Pablo Paiewonsky from this older version and is included in version 16 of Planet Simulator (Lunkeit et al., 2011). Pablo Paiewonsky prefers to not evaluate the newer version of SimBA that he developed, because that version had no formal evaluation and, to his knowledge, has not been

used in any published study. His personal opinion is that SEDGES is a much better model. However, if the reviewer insists, we would be willing to compare SEDGES with that newer version of SimBA.

```
For example, the authors show particularly well-simulated
GPP, but it is not clear how much SEDGES improves the
representations and simulations of GPP in SimBA. In other
words, whether this well-simulated GPP is due to the
incorporated processes in SEDGES, or due to the original
framework set up in SimBA model?  The authors need to prove
that SEDGES indeed improves the GPP simulation compared to
that from SimBA. Maybe adding the SimBA simulations in those
relevant figures is the easiest way to illustrate.  It will be
even better if the authors could provide a short summary of how
GPP is modeled in SimBA.
```

The older version of SimBA has a serious problem of multiple steady states, when used in a coupled model (Dekker et al., 2010). When we recently forced the old SimBA offline using the ERA-Interim reanalysis forcing as described in our paper, we also found multiple equilibria (at the end of the 280ppm $CO_2$ spin-up period). One simulation started from bare soil, and the other one was initialized with a "tropical forest" level of biomass. The forest-initialized simulation is only slightly more realistic than the bare ground-initialized simulation.

Results (GPP and ET) from only the forest-initialized simulation are shown below in the two figures for the years indicated (i.e. near the end of the increasing $CO_2$ phase). The simulated GPP is very unrealistic (fig. 1 below); it is much further away from the observational datasets than is the GPP from SEDGES (figure 2 in the main text). The simulated ET (fig. 2 below) is generally too large due to the Manabe bucket "beta" formulation, which gives the potential rate of ET for soil wetness fractions $\geq 0.25$. This figure can be compared with the SEDGES simulation of ET, given in figure 12 of the

Interactive
comment

main paper, which is much more realistic. The GPP figure, especially, shows that this older version of SimBA has very severe deficiencies when forced offline with the ERA-Interim reanalysis data.
```
Similarly, the authors state that SEDGES improves most of the
parameterization of SimBA, but again it is not clear whether
or not those modifications of parameterization indeed improve
SimBA simulation.  The lesson we learnt from current LSMs tells
us that increasing the complexity not necessarily guarantees
a better model performance.  Therefore, I am also concerned
about the trade-off between realism and simplicity.  To balance
this trade-off with the purpose of improving the reliability
and robustness of models, the added processes or modified
parameterizations need to be proved as necessary for improving
the reliability of the model.
```

First, we disagree that the modifications need to be formally shown to be necessary for improving model reliability. Informal evaluations occurred as the model was developed, and changes were made along the way as were deemed necessary. Second, for the majority of cases in which the SimBA parameterizations were changed, the resulting increase in model complexity was negligible. As is partially stated in the introduction, there are four main increases in complexity in SEDGES relative to SimBA: separation of ET into soil and vegetative components, inclusion of aerodynamic conductance in the formulation for carbon uptake by vegetation, full coupling of photosynthesis and transpiration through interactive canopy conductance, and soil organic carbon-dependent soil albedo. Of these three, we justify the need for only the fourth (in section 2.3.1). In the revised manuscript, we will list all four increases in complexity and explain what advantages are gained by incorporating them into SEDGES.

Below, we include a comparison between the old version of SimBA and SEDGES with respect to evapotranspiration and GPP.

[Figure]

> Moreover, the simplifications also need to prove as reasonable.
> For example, the ratio of ci/ca in Equation (6) is considered
> as a constant, but has been shown that the optimal stomatal
> behavior allows ci/ca decreases with VPD, increase with
> temperature (e.g. Prentice et al. 2014, Medlyn et al.
> 2011, Lin et al. 2015). The variation in ci/ca seems
> quite important in terms of capturing the spatial pattern
> of GPP (Wang et al. 2014). This simplification needs a
> justification.

We agree with the reviewer that it would be preferable to have variable ci/ca in our model. In a future version of SEDGES, we hope to include this feature. Its incorporation into SEDGES is actually quite difficult because the simple relationships between ci/ca and VPD that are derived by Medlyn et al. (2011) and Prentice et al. (2014) are incompatible with the SEDGES framework. Medlyn et al. (2011) and Prentice et al. (2014) assume a Fick's Law of Diffusion transmission between the leaf and the outside air. In order to have consistency in moisture and $CO_2$ fluxes between the land surface and atmosphere, the land surface scheme also needs to use a diffusive exchange between outside air and leaves, or at the least, reasonably approximate such diffusive exchange. Diffusive exchange of moisture and $CO_2$ is not used in the SEDGES framework. Instead, transfer occurs from surface to atmosphere through a bulk aerodynamic formulation. This formulation only approximates the diffusive scheme when canopy resistance ($r_c$) greatly exceeds aerodynamic resistance ($r_a$). As is said in the main text, in early versions of SEDGES (coupled to Planet Simulator), it was found that this condition ($r_c >> r_a$) was frequently violated.

It should be mentioned that Pablo Paiewonsky has been extremely interested in incorporating variable ci/ca into SEDGES and did in fact include its dependency on VPD in an early, unpublished version of SEDGES, but later realized that it was not theoretically justifiable and thus removed it.

It should also be noted that even though ci/ca decreases with VPD (Morison and Gifford, 1983), it is nevertheless conserved in the real world under many conditions, including a wide range of different light levels, $CO_2$ levels, and nutrient levels (Wong et al., 1979).

While we view this constant ci/ca as a limitation of the model, we think that the reviewer overstates its importance with regards to capturing the spatial pattern of GPP. Our GPP results with SEDGES suggest that a fixed ci/ca is adequate for this purpose. Also, the paper that the reviewer cites to justify the importance of variable ci/ca on GPP, Wang et al. (2014), shows that the effect of including the RuBP-regeneration limited rate (which has the ci dependency) has a much weaker impact on the spatial pattern of annual GPP than do the two key variables: incoming light levels and foliage cover, whose variation are indeed incorporated into SEDGES.

Moreover, LUE (light-use efficiency) models using remotely sensed data have been successfully used for many years, but typically lack explicit ci dependence (e.g., Yuan et al., 2007). (Granted, though, ci dependence may be implicit in productivity's dependence on VPD in some of those models). More specifically, with regards to the VPD dependency of ci/ca and its subsequent effects on GPP, as we describe below, dependency of GPP on VPD in SEDGES does occur by way of the water-limited rate, GPPW, i.e. by hydraulic transport limitation. In this sense, productivity limitation by VPD still occurs in SEDGES, albeit through a different mechanism than by ci/ca reduction. With regards to the temperature dependency of ci/ca and its subsequent effects on GPP, colder temperatures do result in lower ci (Prentice et al., 2014; Lin et al., 2015), which, in isolation, reduces GPP in the Farquhar model of photosynthesis (Farquhar et al., 1980). However, this effect is substantially offset by the decrease in the $CO_2$ compensation point with lower temperatures (Prentice et al., 2017). On the other hand, we think it would be beneficial to discuss in the manuscript what the ramifications are of using a fixed ci/ca on GPP, for the situations in which it is significant, and also the ramifications for simulating transpiration.

[Figure]

Having a constant ci/ca is likely to play a more important role in transpiration (by way of affecting the water use efficiency) than (at least directly) in GPP. This is because, when $r_c >> r_a$ (i.e. using the diffusive approximation for heuristic purposes), transpiration is (unless it is occurring at the maximum rate in our model) proportional to VPD*GPPL/(1 - ci/ca), where GPPL is the light-limited rate of GPP. As such, changes in ci/ca (whose values are typically closer to 1 than 0 for C3 plants) have greater relative impact on transpiration than they do in the equations for either RuBP regeneration-limited or Rubisco-limited photosynthesis (at least for non-extreme conditions) in the Farquhar model (Farquhar et al., 1980).

```
I think it is ci (the intercellular CO2 concentration) that
really matters in CO2 fertilization.  If you consider ci
for water-limited GPP in Equation (6), why not here for
light-limited GPP?"
```

As stated in the paper, the parameterization for $CO_2$ fertilization (Equation 3) comes from Franks et al. (2013). Pablo Paiewonsky has had a similar concern as the reviewer and agrees that a dependency on ci would make more sense, from a theoretical standpoint. The above fertilization parameterization is for the RuBP regeneration-limited rate of photosynthesis, but it makes the approximation that ci $\approx$ ca. In Pablo Paiewonsky's current interpretation, the Franks et al. (2013) parameterization is with respect to ca and not ci because it seems that ca and not always ci was measured in the $CO_2$ sensitivity experiments that they used to validate their parameterization. So, strictly speaking, the empirical support of that fertilization parameterization is with respect to ca, rather than ci. Regardless, the impact of using ca instead of ci in that equation is not large. For instance, if one uses ci in Equation 3 and ci/ca = 0.8, then it can be shown that there is only an $\approx$ 6% increase in the light-limited rate of GPP at $CO_2$ of 2000ppm as compared to using ca in Equation 3. In fact, a potentially greater source of inaccuracy in the fertilization parameterization than the above is the assumption of a constant $CO_2$ compensation point.

```
Typically, modeled LUE (not only fAPAR) is represented with
a dependency on vapor pressure deficit, why the equation here
does not include such an effect?  Is it implicitly considered
via the coupling with water-limited GPP?
```

Yes, it is correct that the effect of VPD is implicit within the water-limited GPP. Canopy conductance is limited by a maximum rate of transpiration, which, in turn, depends on the specific humidity deficit ($\Delta q$), which, in turn, is approximately proportional to VPD. Note that the effect of VPD has nothing to do with optimality, though.

References

Dekker, S., De Boer, H., Brovkin, V., Fraedrich, K., Wassen, M., and Rietkerk, M.: Biogeophysical feedbacks trigger shifts in the modelled vegetation-atmosphere system at multiple scales, Biogeosciences, 7, 1237-1245, 2010.

Dewar, R. C.: A simple model of light and water use evaluated for Pinus radiata, Tree Physiology, 17, 259-265, 1997.

Farquhar, G., von Caemmerer, S., and Berry, J.: A Biochemical Model of Photosynthetic CO2 Assimilation in Leaves of C3 Species, Planta, 149, 78-90, 1980.

Franks, P. J., Adams, M. A., Amthor, J. S., Barbour, M. M., Berry, J. A., Ellsworth, D. S., Farquhar, G. D., Ghannoum, O., Lloyd, J., McDowell, N., Norby, R. J., Tissue, D. T., and von Caemmerer, S.: Sensitivity of plants to changing atmospheric CO2 concentration: from the geological past to the next century, New Phytologist, 197, 1077-1094, 2013.

Kleidon, A.: The climate sensitivity to human appropriation of vegetation productivity and its thermodynamic characterization, Global and Planetary Change, 54, 109-127, 2006.

Lunkeit, F., Bottinger, M., Fraedrich, K., Jansen, H., Kirk, E., Kleidon, A., and Luksch, U.: Planet Simulator-reference manual, version 15.0, Tech. rep., Meteorologisches Institut, Universität Hamburg, 2007.

Lunkeit, F., Borth, H., Bottinger, M., Fraedrich, K., Jansen, H., Kirk, E., Kleidon, A., Luksch, U., Paiewonsky, P., S, S., Sielmann, S., and Wan, H.: Planet Simulator-reference manual, version 16, Tech. rep., Meteorologisches Institut, Universität Hamburg, 2011.

Medlyn, B. E., Duursma, R. A., Eamus, D., Ellsworth, D. S., Prentice, I. C., Barton, C. V., Crous, K. Y., de Angelis, P., Freeman, M., and Wingate, L.: Reconciling the optimal and empirical approaches to modelling stomatal conductance, Global Change Biology, 17, 2134-2144, 2011.

Morison, J.I. and Gifford, R.M.: Stomatal sensitivity to carbon dioxide and humidity, Plant physiology, 71, 789-796, 1983.

Mueller, B., Hirschi, M., Jimenez, C., Ciais, P., Dirmeyer, P., Dolman, A., Fisher, J., Jung, M., Ludwig, F., Maignan, F., et al.: Benchmark products for land evapotranspiration: LandFlux-EVAL multi-data set synthesis, Hydrology and Earth System Sciences, 17, 3707-3720, 2013.

Prentice, I. C., Dong, N., Gleason, S. M., Maire, V., and Wright, I. J.: Balancing the costs of carbon gain and water transport: testing a new theoretical framework for plant functional ecology, Ecology letters, 17, 82-91, 2014.

Prentice, I.C., Cleator, S.F., Huang, Y.H., Harrison, S.P., and Roulstone, I.: Reconstructing ice-age palaeoclimates: Quantifying low-CO 2 effects on plants, Global and Planetary Change, 149, 166-176, 2017.

Wang, H., Prentice, I. C., and Davis, T. W.: Biophsyical constraints on gross primary production by the terrestrial biosphere, Biogeosciences, 11, 5987-6001, 2014.

Wong, S. C., Cowan, I. R., and Farquhar, G. D.: Stomatal conductance correlates with photosynthetic capacity, Nature, 282, 424-426, 1979.

Yuan, W., Liu, S., Zhou, G., Zhou, G., Tieszen, L. L., Baldocchi, D., Bernhofer, C., Gholz, H., Goldstein, A. H., Goulden, M. L., et al.: Deriving a light use efficiency model

from eddy covariance flux data for predicting daily gross primary production across biomes, Agricultural and Forest Meteorology, 143, 189-207, 2007.

[Figure]

**Mean annual GPP for SimBA**

**Mean annual GPP for MTE**

**Mean annual GPP for CARBONES**

0    400    800    1200    1600    2000    2400    2800    3200

gC m$^{-2}$ yr$^{-1}$

**Fig. 1.** Multi-year annual mean of gross primary productivity (GPP) for "old" SimBA (version 15 of Planet Simulator) and two reference datasets, MTE and CARBONES, for 1990-2009 over non-glaciated land.

[Figure]

**Fig. 2.** Zonal monthly mean and annual mean climatologies of evapotranspiration for "old" SimBA (version 15 of Planet Simulator) and Mueller et al. (2013) reference dataset for 1989-2005 (non-glaciated land).

---

## Author Comment (AC3) · 18 Jul 2017

**Reply to Anonymous Referee #1**

July 18, 2017

The reviewer brings to our attention the lack of clarity in the manuscript with respect to exactly what the SEDGES model is and what it does. We agree that this is a very important issue and will most certainly address it in the revised manuscript. Briefly, SEDGES is not a land surface model. It simulates *some* aspects of the land surface, e.g. surface albedo. SEDGES is a dynamic vegetation model. It needs to be used in conjunction with (i.e. is "'auxiliary' to") a full-fledge land surface model.

We respond now in more detail to specific comments made by the reviewer:

```
Sentence starting 'In such a framework' gives two reasons for
needing a more sophisticated model.
```

We will clarify our logic better in the revised manuscript. The idea here is that a model is only as good as its weakest link(s). In general, the benefit of having one component of an Earth System Model be sophisticated will be small when there are bigger inaccuracies coming from other components of the model.

```
P2, lines 1-9 reads as if unduly critical of existing analyses
?  potentially of the original model developers.  Was there
really no validation of SimBA?
```

[Figure]

As far as we know, the only sort of validation of SimBA is in Kleidon (2006) in which the simulation by SimBA coupled to Planet Simulator of annual means of GPP, surface temperature, and precipitation are shown. However, no comparison is made with observational-based datasets.

```
P2, lines 10-15.  This really is the point at which more
information should be provided on what SEDGES actually does,
at an over-arching level, before leading in to detailed
Model Description.  What are the core additional quantities
that SEDGES generates, and that are in addition to existing
land surface models?  It is also still vague about ?model
evaluation?.  Both SEDGES and another trusted land surface
model are forced simultaneously with reanalysis data, and
certain diagnostics compared?
```

In the revised paper, we will rework lines 10-15 on the top of page 2 to address the reviewer's concerns.

P2,3 Overview: Again, we will clarify better what SEDGES is and does in the revised manuscript. SEDGES *is* a Dynamic Global Vegetation Model (DGVM).

```
Page 3.  Assumption that NPP/GPP=0.5.  This does feel like a
very large assumption, and particularly as thermal responses in
respiration might behave differently to gross photosynthesis.
The authors themselves appear cautious, with the caveat that
this might be accurate on very long time scales.  However, this
does mean that in comparison with other land surface models,
then only long-term averages of NPP and GPP should be compared.
```

The reviewer is correct in saying that only longer-term (i.e. weeks or longer as is mentioned in the text) averages of NPP and GPP should be compared. We will add more discussion on this limitation. This point is relevant for those who are interested in *short-*

Interactive
comment

*term* changes in carbon fluxes from the land surface. SEDGES might be unsuitable for such purposes if high precision is required.

```
Page 4,5 The Tables are excellent, and highly appropriate for
a model development journal.  It is also appreciated that all
units are presented, and where justification of parameters is
linked to existing literature.
```

We appreciate the reviewer's compliment on our tables!

In the revised manuscript, the references of "climatologies" will be changed for ecological variables (e.g. GPP) to something like "multi-year monthly means". The reviewer is right in pointing this out.

Regarding the line wrapping, we will consult with the journal editors to match what they want, etc.

```
Page 7.  There is some evidence now that splitting SW radiation
in to direct and diffuse can have an influence on PAR. Is this
something the authors considered, or maybe for the next model
version?
```

We believe that the reviewer here means to refer to the effect of direct/diffuse partitioning on light-use efficiency (LUE), rather than on PAR. Direct and diffuse radiation separation was considered at one point during model development, but it was deemed to be not worth the additional complexity at the time. In doing research with regards to this concern, we found two observation-based studies on the relationship between the diffuse fraction of SW radiation at top-of-canopy and LUE that control for the negative correlation between VPD (vapor pressure deficit) and diffuse SW fraction in their results (Alton et al., 2007; Williams et al., 2014). When going from conditions of predominantly direct solar radiation to predominantly diffuse solar radiation, Alton et al. (2007) finds an observed 6% to 33% increase in LUE in three forests, whereas Williams et al. (2014)

finds an ≈17% increase in LUE in shrub tundra. The increase in LUE is apparently due to a more even distribution of PAR among the leaves, which reduces light saturation among the sunlit leaves. The distinction between sunlit and shade leaves is missing in our model's single big leaf approach to canopy radiation, which tacitly assumes a spatially-averaged light profile at each level of the canopy (de Pury and Farquhar, 1997; Monson and Baldocchi, 2014, p. 355). In a future version of SEDGES, we hope to incorporate the sunlit/shade leaf distinction. Not including it implies that, in the absence of water limitation, our model underpredicts GPP at low sun angles and under cloudy conditions (and overpredicts it for opposite conditions). We will discuss this in the revised manuscript.

If, on the other hand, the reviewer *is* referring to how the PAR fraction of incoming SW radiation varies with its diffuse fraction, then, from what we have seen in the literature, the effect is quite small: ≤0.03 variation in PAR fraction over all observed diffuse fractions (Jacovides et al., 2004; Li et al., 2010), which can be neglected for a simple model such as SEDGES.

Regarding the extensive use of footnotes: we will try to reduce these in the revised manuscript by putting them back into the main text and/or omitting them as appropriate.

```
P11-14.  These variables are the more novel parts of this land
surface model, and it might be appropriate to re-iterate this
point?  That is, components more associated with carbon stores
than the fluxes.
```

Again, in the revised manuscript, we will make clear that SEDGES is a DGVM that simulates only some aspects of the land surface, etc.

```
P12, 13.  The terminology could be made clearer between leaf
cover fraction and forest cover fraction.  In some DGVMs, these
could potentially be the same thing.  I guess from Equation
(18) this is to do with wilting of leaves and that does not
```

```
appear in forest cover fraction.  It also includes seasonal
phenology?
```

In the revised manuscript, we will add a short definition of forest cover fraction in section 2.2.9 to make clearer what it is. There is no seasonal phenology with regards to forest cover fraction.

```
P17, Section ?How to Couple SEDGES?.  Unfortunately at this
point in the paper, I am again confused as to exactly what
SEDGES is, given that it needs the variables listed lines
24,27.  The issue here is that some of these components do
not uncouple?  For instance, if SEDGES predicts LAI, then
altered LAI will adjust transpiration, in turn affecting soil
moisture content.  So soil moisture content cannot be regarded
as a pure input?  I am happy to accept that I might not have
fully understood the direction the paper is taking, but this
could be made clearer.  I can see that there are hints of this
discussion around the middle of page 18.
```

Some variables such as soil moisture content ($W_{soil}$) and (total) ET are to be simulated outside of SEDGES, i.e. by the land surface model that SEDGES is coupled to. However, the simulation of such variables will use SEDGES output, which in this case would be the surface wetness factor, $C_w$ (section 2.2.5) and also (possibly) the soil water holding capacity, $W_{max}$, for the case of $W_{soil}$. The important thing is that there be compatibility between how these variables are simulated by the land surface scheme and what SEDGES presupposes for a land surface scheme (e.g. the simplified mosaic approach and neglect of canopy interception). We will make these things clearer in the revised manuscript. ET is not input into SEDGES, but rather PET. A source of the confusion here has probably been what the reviewer indicated above: that in the current manuscript, it is not sufficiently clear what SEDGES is and does.

[Figure]

P18. Related to the point above, in Equation (29), is ETsoil derived from SEDGES? If so, then Wsoil becomes a diagnostic, rather than an independent forcing.

First of all, there is a typographical error on line 22 of page 19. Instead of "ET", it should be "PET". We will correct this in the revised manuscript. Secondly, what is done with ETsoil is admittedly confusing at the moment. ETsoil is only calculated explicitly when there is snow cover present, because, in this scenario, a distinction needs to be made between sublimation from snow and sublimation from the soil. ETsoil is therefore SEDGES model output (and thus soil water content, $W_{soil}$, is diminished by it in the land surface model that is coupled to SEDGES). The revised manuscript will explicitly include ETsoil as an output variable, and we will introduce it earlier, probably in section 2.2.5, around equation 13, rather than in section 4.

What might also be confusing the reviewer is that equation 29 in section 4 only describes the simple hydrological model that we used as part of our forcing of SEDGES offline with the reanalysis data, and the calculation of $W_{soil}$ in that equation is not part of the actual SEDGES model. Perhaps we should make this more clear in the revised manuscript?

$W_{soil}$ is not a diagnostic. As we say on line 14 of page 19, $W_{soil}$ is a prognostic variable. It needs to be simulated outside of SEDGES and inputted into SEDGES. Regardless of how it is formulated to change, $W_{soil}$ will depend on SEDGES output (as we describe above). However, $W_{soil}$ also depends on the hydrological scheme of the land surface model that SEDGES is coupled to (and thus includes whatever processes are involved in runoff generation in that scheme).

Note that what is input and output for SEDGES versus what is input and output for the whole scheme of forcing SEDGES with reanalysis data differ. Again, maybe we should make this more clear in the revised manuscript...

On line 20 in section 4, there is a typographical error. $W_{frac}$ should instead be $W_{max}$.

We will correct this in the revised manuscript.

p. 20: `However I am less convinced by the need to compare against other land surface models.`

We understand the concern that a comparison of SEDGES to other models is not in all instances valuable or at least not necessary. However, we believe that it is important to give the reader some ideas how SEDGES compares with other model present-day simulations. In particular, one could argue that such information is important in any case: if SEDGES were an outlier model for some specific simulated variables or processes then the reader should be made aware of this situation. Or when SEDGES is similar to other model results, this could be useful in cases when the observational database is known to have large uncertainties.

We are open to suggestions regarding this issue. Reviewer #2 has indicated a preference for not comparing SEDGES with state-of-the-art land surface models, but reviewers #2 and #3 would like to see SEDGES compared with SimBA, if not also other low-complexity models. It seems very reasonable to us to at least reduce the number of comparisons with state-of-the-art models in the revised manuscript, i.e. restrict the comparisons to just the most essential variables and/or eliminate the comparisons to vegetative and soil carbon simulated by Earth System Models (Todd-Brown et al., 2013, Jiang et al., 2015), since these carbon values depend on the simulated climate in those models.

References

Alton, P.B., North, P.R. and Los, S.O.: The impact of diffuse sunlight on canopy light-use efficiency, gross photosynthetic product and net ecosystem exchange in three forest biomes, Global Change Biology, 13, 776-787, 2007.

de Pury, D.G.G. and Farquhar, G.D.: Simple scaling of photosynthesis from leaves to canopies without the errors of big-leaf models, Plant, Cell and Environment, 20, 537-

557, 1997.

Jacovides, C.P., Timvios, F.S., Papaioannou, G., Asimakopoulos, D.N. and Theofilou, C.M., Ratio of PAR to broadband solar radiation measured in Cyprus, Agricultural and Forest Meteorology, 121, 135-140, 2004.

Jiang, L., Yan, Y., Hararuk, O., Mikle, N., Xia, J., Shi, Z., Tjiputra, J., Wu, T., and Luo, Y.: Scale-Dependent Performance of CMIP5 Earth System Models in Simulating Terrestrial Vegetation Carbon, Journal of Climate, 28, 5217-5232, 2015.

Kleidon, A.: The climate sensitivity to human appropriation of vegetation productivity and its thermodynamic characterization, Global and Planetary Change, 54, 109-127, 2006.

Li, R., Zhao, L., Ding, Y., Wang, S., Ji, G., Xiao, Y., Liu, G. and Sun, L., Monthly ratios of PAR to global solar radiation measured at northern Tibetan Plateau, China, Solar Energy, 84, 964-973, 2010.

Monson, R. and Baldocchi, D.: Terrestrial biosphere-atmosphere fluxes, Cambridge University Press, 2014.

Todd-Brown, K., Randerson, J., Post, W., Hoffman, F., Tarnocai, C., Schuur, E., and Allison, S.: Causes of variation in soil carbon simulations from CMIP5 Earth system models and comparison with observations, Biogeosciences, 10, 1717-1736, 2013.

Williams, M., Rastetter, E.B., Van der Pol, L. and Shaver, G.R.: Arctic canopy photosynthetic efficiency enhanced under diffuse light, linked to a reduction in the fraction of the canopy in deep shade, New Phytologist, 202, 1267-1276, 2014.
* * *

---

## Author Response (AR1)

**Description and Validation of the Simple, Efficient, Dynamic, Global, Ecological Simulator (SEDGES v.1.0)**

Pablo Paiewonsky[1] and Oliver Elison Timm[1]

[1]Department of Atmospheric and Environmental Sciences, State University of New York at Albany, 1400 Washington Ave., Albany, NY 12222

*Correspondence to:* Pablo Paiewonsky (ppaiewonsky@albany.edu)

Reviewer comments that were not addressed in our previous responses are addressed now. Also, some more specific comments on the changes are included.

**With respect to reviewer #1's comments:**

The wording could be tightened in the Abstract to explain how SEDGES is "auxiliary" to the land-atmosphere coupling scheme. This will make it easier to understand better the sentence "evaluate.
. ..using a simple land surface scheme that is driven by reanalysis data". If SEDGES is compared extensively to another existing land surface model as ground truth, then from the Abstract alone, the reader will wonder what it does in addition.

We think that the original abstract was ambiguous in its meaning. We have updated our abstract to hopefully reduce such confusion, although we have not explicitly stated which variables are simulated by SEDGES because we feel that this is too specific for an abstract. Regardless, we thank reviewer #1 for signaling the need to revise the abstract.

P2, lines 1-9 reads as if unduly critical of existing analyses– potentially of the original model developers. Was there really no validation of SimBA?

We added a short clarification of this in the revised paper. For more clarification, see our first response to reviewer #1.

P12, 13. The terminology could be made clearer between leaf cover fraction and forest cover fraction. In some DGVMs, these could potentially be the same thing. I guess from Equation (18) this is to do with wilting of leaves and that does not appear in forest cover fraction. It also includes seasonal phenology?

In addition to adding a definition of forest cover fraction (as stated in our first response to reviewer #1), we changed the notation for leaf cover fraction throughout the paper from $f_{veg}$ to $f_{leaf}$, which should hopefully emphasize that it is *leaf* cover fraction and not *vegetative* cover fraction. This also required some revisions to appendix B, in which we had, in fact, interchanged the use of leaf cover fraction and vegetative cover fraction. We also added text to indicate that LAI has a one-to-one relationship with the leaf cover fraction, and we added an additional equation to make explicit how the LAI of moist soils relates to the leaf cover fraction with moist soils. All these changes should also clear up the second reviewer's confusion with regards to LAI and leaf cover fraction.

p. 20: However I am less convinced by the need to compare against other land surface models.

With respect to this issue, that was also raised by reviewer #2, we have now eliminated the comparisons of vegetative and soil carbon simulated by SEDGES with that simulated by Earth System Models (Todd-Brown et al., 2013; Jiang et al., 2015) in section , since these carbon values depend on the simulated climate in those models (see section 5.3). The remaining comparisons have been left in.

**With respect to reviewer #2's comments:**

Specifically in response to the reviewers's criticism that we should show more non-GPP results, we have expanded the section on ET and runoff, and now include a figure for each showing the full spatial climatological annual mean values for SEDGES and the reference datasets.

We decided not to make any comparisons of SEDGES with VECODE because there is not that much information to be gained from comparing global NPP and forest cover fraction. The latter is not that important of a variable, at least in SEDGES.

Also as promised in our first response to the reviewer, we clarified leaf cover fraction and made notational changes, as indicated above in our addressing of reviewer #1's comments.

Discussion part must be developed. For each matchless outputs, the authors must explain why they obtain this result and how this lack of precision can affect the outputs of coupled simulations.

As per the reviewer's wishes, we have expanded our discussion and conclusions section. Overall we discuss more critically some outstanding problems in the parameterizations on NPP- and ET-relevant processes. In so doing, we address the fixed $c_i/c_a$ ratio that reviewer #3 was concerned about, and suggest potential candidates for future improvements. Furthermore, we discuss limits of the current model scheme for paleoclimate applications.

As promised in our first response to the reviewer, we included a brief discussion in the discussion and conclusions section on the expected effect of SEDGES biases on coupled simulations. Part of doing this entailed expanding the discussion in the albedo validation section.

**With respect to reviewer #3's comments:**

Although we had said that "we think it would be beneficial to discuss in the manuscript what the ramifications are of using a fixed $c_i/c_a$ on GPP, for the situations in which it is significant", we decided, at least for now, to minimize discussion of the negative ramifications of using a fixed $c_i/c_a$ on GPP, and instead mention (in the discussion and conclusions section) that the light use efficiency approach without an explicit formulation for $c_i$ is quite common and has not been found to be less accurate in simulating GPP. However, as suggested in our first response to reviewer #3, we have included discussion on the effects of using a fixed $c_i/c_a$ on transpiration (also in the discussion and conclusions section).

**Other changes that were made include the following:**

Throughout the paper, to be more specific, "soil carbon" was replaced with "soil organic carbon". The URL commands were used for the URL's, which are mostly in the data availability section.

[revised manuscript text omitted]
}$ kgC m$^{-2}$ vegetative carbon $L$ kgC m$^{-2}$ s$^{-1}$ litterfall $NPP$ kgC m$^{-2}$ s$^{-1}$ net primary productivity $GPP$ kgC m$^{-2}$ s$^{-1}$ gross primary productivity $GPP_L$ kgC m$^{-2}$ s$^{-1}$ light-limited gross primary productivity $GPP_W$ kgC m$^{-2}$ s$^{-1}$ water-limited gross primary productivity $f_1(CO_2)$ - $CO_2$ fertilization function $f_2(T_{sfc})$ - temperature limitation function $T_{sfc}$ K surface temperature $f_{APAR}$ - fraction of photosynthetically active radiation (PAR) that is absorbed by green vegetation $SW\downarrow$ W m$^{-2}$ surface downwelling short wave radiation $LAI$ m$^2$ leaf area (m$^2$ ground area)$^{-1}$ leaf area index $f_{veg}$ - fractional green vegetation (leaf) cover $g_a$ m s$^{-1}$ aerodynamic conductance $r_a$ s m$^{-1}$ aerodynamic resistance $r_c$ s m$^{-1}$ canopy resistance $\rho$ kg m$^{-3}$ surface air density $p_{sfc}$ Pa surface pressure $ET$ m$^3$ m$^{-2}$s$^{-1}$ evapotranspiration $qsat_{sfc}$ kgH$_2$O kgair$^{-1}$ surface saturation specific humidity $q$ kgH$_2$O 
[revised manuscript text omitted]

| $L$ | $kgC\,m^{-2}\,s^{-1}$ | litterfall |
| $NPP$ | $kgC\,m^{-2}\,s^{-1}$ | net primary productivity |
| $GPP$ | $kgC\,m^{-2}\,s^{-1}$ | gross primary productivity |
| $GPP_L$ | $kgC\,m^{-2}\,s^{-1}$ | light-limited gross primary productivity |
| $GPP_W$ | $kgC\,m^{-2}\,s^{-1}$ | water-limited gross primary productivity |
| $f_1(CO_2)$ | ~ | $CO_2$ fertilization function |
| $f_2(T_{sfc})$ | ~ | temperature limitation function |
| $T_{sfc}$ | K | surface temperature |
| $f_{APAR}$ | ~ | fraction of photosynthetically active radiation (PAR) that is absorbed by green vegetation |
| $SW\downarrow$ | $W\,m^{-2}$ | surface downwelling short wave radiation |
| $LAI$ | $m^2$ leaf area $(m^2$ ground area$)^{-1}$ | leaf area index |
| $f_{leaf}$ | ~ | vegetative leaf cover fraction |
| $g_a$ | $m\,s^{-1}$ | aerodynamic conductance |
| $r_a$ | $s\,m^{-1}$ | aerodynamic resistance |
| $r_c$ | $s\,m^{-1}$ | canopy resistance |
| $\rho$ | $kg\,m^{-3}$ | surface air density |
| $P_{sfc}$ | Pa | surface pressure |
| $ET$ | $m^3\,m^{-2}s^{-1}$ | evapotranspiration |
| $q_{sat,sfc}$ | $kgH_2O\,kgair^{-1}$ | surface saturation specific humidity |
| $q$ | $kgH_2O\,kgair^{-1}$ | specific humidity at the lowest atmospheric level |
| $C_w$ | ~ | surface wetness factor |
| $\beta_{ss}$ | ~ | soil surface water stress factor |
| $r_{ss}$ | | soil surface resistance |
| $W_{frac}$ | ~ | soil wetness fraction |
| $W_{soil}$ | m | soil water content |
| $W_{max}$ | m | soil "bucket" depth |
| $T$ | $m^3\,m^{-2}s^{-1}$ | transpiration |
| $r_{cu}*$ | $s\,m^{-1}$ | case-specific unconstrained canopy resistance |
| $r_{cu}$ | $s\,m^{-1}$ | unconstrained canopy resistance |
| $\beta_{tr}$ | ~ | water stress factor for transpiration |
| $r_{c,min}$ | $s\,m^{-1}$ | minimum canopy resistance |
| $C_{soil}$ | $kgC\,m^{-2}$ | soil organic carbon |
| $R_{soil}$ | $kgC\,m^{-2}\,s^{-1}$ | soil respiration rate |
| $T_{soil}$ | K | soil temperature at 0.25m depth |
| $LAI_m$ | $m^2$ leaf area $(m^2$ ground area$)^{-1}$ | leaf area index without soil moisture stress |
| $f_{leaf,m}$ | ~ | (green) leaf cover fraction in absence of soil moisture stress |
| $f_{leaf,dry}$ | ~ | max vegetative leaf cover fraction under soil moisture stress |
| $f_{for}$ | ~ | forest cover fraction |
| $\alpha_0$ | ~ | snow-free surface albedo |
| $\alpha_{soil}$ | ~ | albedo of bare soil |
| $\alpha$ | ~ | albedo |
| $\alpha_{snow,flat}$ | ~ | snow-covered albedo of flat portion of grid cell |
| $\alpha_{snow,for}$ | ~ | snow-covered albedo of forested portion of the grid cell |
| $f_{snow,flat}$ | ~ | fraction of "flat" portion of grid cell that is snow-covered |
| $swe$ | $m^3\,m^{-2}$ | snow depth in liquid water equivalent |
| $\alpha_{deep,snow,flat}$ | ~ | albedo of deep and pure snow |
| $z_0$ | m | surface roughness |
| $z_{0,oro}$ | m | surface roughness due to orography |
| $z_{0,veg}$ | m | surface roughness due to vegetation |
| $P$ | $m^3\,m^{-2}\,s^{-1}$ | precipitation in liquid water equivalent |
| $S$ | $m^3\,m^{-2}\,s^{-1}$ | snowfall in liquid water equivalent |
| $M$ | $m^3\,m^{-2}\,s^{-1}$ | snowmelt in liquid water equivalent |
| $ET_{soil}$ | $m^3\,m^{-2}\,s^{-1}$ | bare soil evaporation plus transpiration |
| $E_{soil}$ | $m^3\,m^{-2}\,s^{-1}$ | bare soil evaporation |
| $PET$ | $m^3\,m^{-2}\,s^{-1}$ | potential evapotranspiration |

**Table 2.** SEDGES parameters in the paper.

| symbol | value | units | description | source(s) |
|---|---|---|---|---|
| $\tau_{veg}$ | 10 | years (converted into seconds) | biomass residence time | SimBA (all versions) |
| $R_d$ | 287.0 | J K$^{-1}$ kg$^{-1}$ | gas constant for dry air on Earth | - |
| $\epsilon_{max}$ | $5.0 \times 10^{-10}$ | kgC J$^{-1}$ | max. light use efficiency | model calibration |
| $CO_{2comp}$ | 40 | ppmv | $CO_2$ light compensation point | Franks et al. (2013) |
| $T_{crit}$ | 20 | $^\circ$C | temperature at which productivity limitation begins | see section 2.1.3 |
| $k_{veg}$ | 1 | - | light extinction coefficient | see section 2.1.3 |
| $\Omega_c$ | 0.7 | - | clumping index | Pisek et al. (2010); He et al. (2012) |
| $co2conv$ | $4.15 \times 10^{-7}$ | kgC kgair$^{-1}$ ppmv$^{-1}$ | unit conversion factors | manipulation of equation B7 from Raupach (1998) |
| $\frac{c_i}{c_a}$ | 0.80 | - | ratio of intercellular to atmospheric $CO_2$ | somewhat common daytime value for C3 plants |
| $r_{ssmin}$ | 10 | s m$^{-1}$ | minimum soil surface resistance | van de Griend and Owe (1994) |
| $r_{ssmax}$ | $10^{30}$ | s m$^{-1}$ | maximum soil surface resistance | - |
| $\rho_w$ | 1000 | kg m$^{-3}$ | density of liquid water | - |
| $f_{snowfor}$ | 0.12 | - | snow-covered fraction of the forest cover | see section 2.1.5 |
| $trmax$ | $2.78 \times 10^{-7}$ | m s$^{-1}$ | max. transpiration rate | Knorr (2000) |
| $r_{cmin_{min}}$ | 0 | s m$^{-1}$ | absolute min. canopy resistance | - |
| $r_{cmax}$ | $10^{30}$ | s m$^{-1}$ | max. canopy resistance | Sitch et al. (2003) |
| $c_8$ | $\approx 43.3$ (see section 2.1.7) | - | for normalizing 10 $^\circ$C soil respiration to that of SimBA | |
| $c_9$ | 106 | K | for soil respiration | Jenkinson et al. (1990) |
| $LAI_{min}$ | 0.05 | - | min. leaf area index in wet soils | - |
| $LAI_{max}$ | 7 | - | max. leaf area index in wet soils | model calibration |
| $c_6$ | 0.195 | kgC$^{-1}$ m$^2$ | biomass to LAI conversion | model calibration |
| $W_{frac_{crit,lai}}$ | 0.05 | - | critical soil wetness fraction for commencement of leaf fall | model calibration |
| $c_1$ | 0.2 | kgC$^{-1}$ m$^2$ | biomass-forest cover relationship | see section 2.1.9 |
| $c_2$ | 1.0 | kgC m$^{-2}$ | biomass threshold for forest cover commencement | see section 2.1.9 |
| $c_7$ | 9 | kgC m$^{-2}$ | soil organic carbon saturation value with respect to soil albedo | see section 2.2.1 |
| $\alpha_{sand}$ | 0.32 | - | sandy soil albedo | see section 2.2.1 |
| $\alpha_{peat}$ | 0.12 | - | albedo of organic matter-rich soil | see section 2.2.1 |
| $c_4$ | 1.5 | kgC$^{-1}$ m$^2$ | shape parameter for snow-covered albedo | model calibration |
| $c_5$ | 1.5 | kgC m$^{-2}$ | biomass threshold for snow masking | model calibration |
| $\alpha_{mindeepsnowflat}$ | 0.40 | - | albedo of warm, deep, pure snow | Roesch et al. (2001) |
| $\alpha_{maxdeepsnowflat}$ | 0.80 | - | albedo of cold, deep, pure snow | Roeckner et al. (2003) |
| $\alpha_{maxsnowfor}$ | 0.30 | - | maximum albedo of snow-covered forest | Moody et al. (2007) |
| $c_{12}$ | 0.10 | kgC$^{\frac{1}{2}}$ | conversion of biomass into soil bucket depth | model calibration |
| $W_{max_{min}}$ | 0.05 | m | minimum soil bucket depth | see section 2.2.2 |
| $z_{0min}$ | 0.01 | m | surface roughness for bare soil | Oke (1987) |
| $z_{0const}$ | $\approx 0.035$ | m | biomass-roughness relationship | see section 2.2.3 |
| $c_{15}$ | 8 | kgC m$^{-2}$ | biomass-roughness relationship | model calibration |
| $c_{16}$ | 0.5 | kgC$^{-1}$ m$^2$ | biomass-roughness relationship | model calibration |
| $c_{17}$ | 2.5 | m | $\approx$ surface roughness for fully-forested land | typical value for tropical rainforests (Sellers et al., 1996b) |

**Table 3.** RMSE's and correlations between multi-year annual means of SEDGES variables and those for reference datasets

| variable | correlation | RMSE | reference dataset | analyzed years |
|----------|-------------|------|-------------------|----------------|
| $ET$ | 0.778549 | $0.72505 \, \text{mm d}^{-1}$ | Mueller et al. (2013) | 1989-2005 |
| $GPP$ | 0.924035 | - | MTE (Jung et al., 2011) | 1990-2009 |
| $GPP$ | 0.861378 | - | CARBONES | 1990-2009 |
| vegetative carbon | 0.570484 | $3.91639 \text{kgC m}^{-2}$ | Olson et al. (1985) | see text |
| soil organic carbon | 0.579095 | $7.93438 \text{kgC m}^{-2}$ | HWSD v.1.2 (Wieder et al., 2011) | see text |

---

## Referee Report (RR1)

**Review comments**

This study present a simple dynamic global vegetation model (SEDGES) to simulate ecological, hydrological and surface energy variables. The results showed that gross primary production is well simulated, and this model is useful to simulate large scale vegetation and land surface characteristics. However, there are several concerns should be taken into consideration.

1. The SEDGES is based on the SimBA model, and the SEDGES builds upon SimBA by improving most of its parameterizations. Compared to the SimBA, SEDGES has four major increases in complexity. However, the authors should explain more about how they improve most of SimBA parameterizations and what's different between them.

2. The SEDGES uses "big leaf" formulation for vegetation $CO_2$ uptake, but why don't SEDGES use the individual plants and trees formulation to capture the outcome of competition for environmental resources?

3. SEDGES uses a constant NPP/GPP = 0.5 approximation, and only impacts biomass changes and the latter occur on very long time scales. The constant value might be difficult to capture the dynamic of ecosystem, especially on short term temporal dynamics.

4. In general, SEDGES has well spatial correlations but weak temporal correlation with reference datasets in GPP, LAI etc. For the model simulation, temporal dynamic is more challenged to capture, but it is important to understand the temporal variation of ecosystems. The authors should explain more about weak temporal correlation and how it affects the accuracy and uncertainty of the model.

5. In page 22 L11-15, the interannual variability of global GPP for 1990-2009 in SEDGES is 1.79 PgC yr$^{-1}$, whereas it is 2.50 PgC yr$^{-1}$ for referenced dataset. This result means SEDGES underestimated the seasonal variations and phenology, and might be also limited to capture extreme climate events or disturbance.

---

## Author Response (AR2)

[revised manuscript text omitted]

**With respect to reviewer #1's comments:**

The reviewer's suggestion to highlight the model variables "purpose/function" into input and output variables and "modified" in the table is a great idea. The table was updated accordingly, using extra categories for variables that did not fit into the originally suggested three. In section 3 (top of p.19), we added to the list of outputs by SEDGES the temperature limitation function ($f_2(T_{sfc})$), litterfall ($L$), and the minimum canopy resistance $r_c min$. (We had inadvertently left these out in the previous paper version). When first introducing the table in section 2.1 (the top of page 4), we now mention the variable categories, too.

In modifying table 1, we discovered that the calculation locations and input/output natures of evapotranspiration (ET) and soil evaporation (Esoil) could be confusing to a prospective user of the model, especially because ET occurs outside of SEDGES, but Esoil occurs *in* SEDGES

when there is snow cover. To remedy such possible confusion, we made changes to the text that better highlight the special case of soil evaporation as follows:

We redid the beginning of section 2.2.5 (Evapotranspiration and Transpiration): almost all the text in lines 7 to 22 of p.8.

We also added a sentence that explains how Esoil and ET relate to ETsoil at the very bottom of p.20.

**With respect to reviewer #4's comments:**

The SEDGES is based on the SimBA model, and the SEDGES builds upon SimBA by improving most of its parameterizations. Compared to the SimBA, SEDGES has four major increases in complexity. However, the authors should explain more about how they improve most of SimBA parameterizations and what's different between them.

Overall, we modified the paper to include more specific differences between SEDGES and SimBA and to be more comprehensive in our discussion of these differences. Where previously missing, we now include discussion on why the new SEDGES parameterizations are better and/or why we choose to use them. Below, we explain these paper changes in more detail.

At the end of section 2.1 (p.4, lines 9-14), we added, "In sections 2.2 and 2.3, we describe in detail the equations of the ecological and physical variables in the SEDGES model. Because SEDGES uses the original SimBA model (Kleidon, 2006b) (and its code) as a basis, we explicitly mention in the text when significant changes are introduced in SEDGES compared with SimBA. We provide extra detail on the original SimBA for the cases in which Kleidon (2006b) provides insufficient information for comparison with SEDGES. The broadest structural increases in complexity in SEDGES have already been identified in the introduction; our focuses have finer scope in these coming sections."

How new parameterizations in SEDGES differ from SimBA were scarcely or not at all discussed in four cases before: soil respiration, forest cover fraction, the snow cover fraction of flat land, and soil bucket depth. Now, the original SimBA equations of these are provided for comparison purposes in the paper in (lines 13-14 of p.12; lines 13-16 of p.14; footnote 6 on p.16; lines 14-17 on p.17; respectively).

In the "Forest Cover Fraction" subsection (p.14, lines 16-17), the following sentence was added: "Neither SimBA parameterization yields a good match with the aforementioned reference data in the absence of using additional fitting parameters."

Footnote 6 on p.16 now includes the SimBA parameterization for snow cover fraction of flat land and also explains how it was found to be worse than the ECHAM5 one used in SEDGES.

We include a new brief argument in favor of the new SEDGES parameterization for soil bucket depth (Wmax) in section 2.32 (lines 16-17 on p.17).

A new brief argument in favor of the new SEDGES parameterization for CO2 fertilization is now included in section 2.2.3 (line 3 of p.6).

We updated the GPPL section (2.2.3) to explicitly indicate that GPPL's "overall structure is very similar to that of the original version of SimBA (Kleidon, 2006b)" (lines 11-12 on p.5). We drew attention to the difference of short wave radiation term in the GPPL formulation of SEDGES as opposed to the original SimBA (lines 19-20 on p.5).

We added the following to Appendix A: "(While the critical temperature for the Kleidon (2006b) SimBA model is 5 $^\circ$C, it otherwise uses the same ramp function formulation as SEDGES.)" (lines 4-5 on p.38)

Other changes that are relevant to the issue of differences between SimBA and SEDGES are our removal of "direct" from "simple and direct determination" in the part of the introduction (line 15 of p.2) that states how LAI depends on biomass. Some readers could argue that LAI is not "directly" determined by biomass in the original SimBA, although it is in SEDGES. To improve clarity, we also added "(which, in turn, depended directly on biomass)" right after "this moist soil LAI was a linear function of forest cover fraction" in section 2.2.8 (line 22 of p.12). We also changed some wording regarding the LAI parameterizations to avoid confusion: "For SimBA in version 16 of Planet

Simulator, the LAI functional dependency on biomass (Lunkeit et al., 2011) was updated to reduce the high number of multiple equilibria" to "This parameterization was discarded in the subsequent SimBA in version 16 of Planet Simulator, in favor of a simpler LAI functional dependency on biomass (Lunkeit et al., 2011) to reduce the high number of multiple equilibria" (lines 22-24 of p.12).

The SEDGES uses "big leaf" formulation for vegetation $CO_2$ uptake, but why don't SEDGES use the individual plants and trees formulation to capture the outcome of competition for environmental resources?

In the Discussion and Conclusions section (bottom line of p.31 and top two lines of p.32), we now mention that the use of one plant functional type "carries over from the SimBA model, on which SEDGES is based. Even an expansion to just two plant functional types (e.g., tree and herbaceous) and simulation of competition between them would entail a significant increase in model complexity."

SEDGES uses a constant NPP/GPP = 0.5 approximation, and only impacts biomass changes and the latter occur on very long time scales. The constant value might be difficult to capture the dynamic of ecosystem, especially on short term temporal dynamics.

The approximation of constant NPP/GPP and its possible inaccuracy on short time scales is already addressed in the main paper in section 2.2.2 and in the Discussions and Conclusions section (lines 16-20 of p.34).

In general, SEDGES has well spatial correlations but weak temporal correlation with reference datasets in GPP, LAI etc. For the model simulation, temporal dynamic is more challenged to capture, but it is important to understand the temporal variation of ecosystems. The authors should explain more about weak temporal correlation and how it affects the accuracy and uncertainty of the model.

In the main paper, we already discuss the regions of weak temporal correlations of GPP, LAI, and ET in sections 5.1, 5.4, and 5.6 respectively, including the reasons for them. We also reiterate here that the annual trend in GPP is captured extremely well by SEDGES and is a model strength.

In page 22 L11-15, the interannual variability of global GPP for 1990-2009 in SEDGES is 1.79 PgC yr-1, whereas it is 2.50 PgC yr-1 for referenced dataset. This result means SEDGES underestimated the seasonal variations and phenology, and might be also limited to capture extreme climate events or disturbance.

We agree with the reviewer that the above result with GPP deserves some discussion. However, underestimating the interannual variability of global GPP is not necessarily caused by an underestimate of the seasonal variations and phenology. Rather, we feel that this general underestimate of GPP variability by SEDGES is caused by several factors. Plausible factors behind this underestimate are now discussed in the text in section 5.1 (lines 22-30 on p.23). We agree with the reviewer that SEDGES is probably somewhat limited in capturing the effects of extreme weather events on vegetation.

Reviewer #4 generally points out that SEDGES may have difficulty in accurately representing some large scale ecological dynamics, especially on short time scales. We rearranged pp. 33-35 of the Discussion and Conclusions section slightly and added a caution with regards to these drawbacks: "a group of model deficiencies in simulating some ecological dynamics. Of these simulation deficiencies, the most severe are probably of the phenological changes (GPP, NPP, transpiration, and especially LAI) associated with green-up in the Northern Hemisphere. More generally, however, it is likely that simplifying assumptions made by SEDGES (namely, constant NPP/GPP as well as its universal temperature limitation function (appendix C)) lead the model to overestimate the capacity of vegetation to adapt to rapidly-changing conditions, especially on daily time scales. As such, SEDGES may underestimate the less positive impacts of extreme weather events on vegetation." (line 33 of p.34 to line 2 of p.35)

**Other changes that were made include the following:**

We made some minor edits to make the use of past and present tenses consistent throughout the paper (e.g., when referring to the SimBA model). In the main text, we added a couple of references to other paper sections for increased clarity. We slightly moved the location of the 4th footnote so that it would not be interpreted as an exponent. Finally, we added missing citations and equation references and corrected numerous small errors. The latter include term omissions in equations 18 and 26 and in the definition of $z_{0const}$ (line 6 of p.18). There were

5    inconsistencies in the values for $T_{soil}$ (previously 0.10 m and 0.25 m). We changed this to 0.20 m and added some explanation for this value when first defining $T_{soil}$ (line 4 of p.12). In table 1, we had inadvertently left out the units of soil surface resistance, so we added them. In the first paragraph of the section on water-limited gross primary productivity (lines 10-11 of p.7), we made a minor edit to change a phrase to passive voice. In section 2.2.5 (lines 15-16 of p.10), we made a minor wording change for referring to the portion of the grid cell that is considered to be snow-free.

10   A typographical error that is worth mentioning is in the equation for soil evaporation (Esoil) (Eq. 15) from partially snow-covered land. It should be "PET" (potential evapotranspiration) on the right hand side instead of "ET" (evapotranspiration). After correcting this error, we made further small changes to section 2.2.5 so that PET would be more clearly presented and integrated, and also so it could be referred to with an equation (Eq. 8). This was needed so that the corrected equation for Esoil would be understandable.

We also corrected numerous even smaller typographical errors in the manuscript (such as incorrect punctuation).

15   In section 2.1., to improve clarity for the reader, we moved and edited a sentence on sub-grid scale heterogeneity from the end of the 3rd paragraph to near the beginning of that paragraph.

Also, in section 2.1, there was essentially a duplicate phrase stating that SEDGES is designed to couple with the Planet Simulator near the end of the 2nd and beginning of the 3rd paragraphs. This was reworked (lines 15-21 of p.3).

**List of Figures**

[revised manuscript text omitted]